# Bogong moths use a stellar compass for long-distance navigation at night

David Dreyer[1,13] ✉, Andrea Adden[1,2,13] ✉, Hui Chen[1,3], Barrie Frost[4,14], Henrik Mouritsen[5], Jingjing Xu[5,6], Ken Green[7], Mary Whitehouse[8], Javaan Chahl[9], Jesse Wallace[1,10,11], Gao Hu[3], James Foster[1,12], Stanley Heinze[1] & Eric Warrant[1,9,10] ✉

Each spring, billions of Bogong moths escape hot conditions across southeast Australia by migrating up to 1,000 km to a place that they have never previously visited—a limited number of cool caves in the Australian Alps, historically used for aestivating over summer[1,2]. At the beginning of autumn, the same individuals make a return migration to their breeding grounds to reproduce and die. Here we show that Bogong moths use the starry night sky as a compass to distinguish between specific geographical directions, thereby navigating in their inherited migratory direction towards their distant goal. By tethering spring and autumn migratory moths in a flight simulator[3–5], we found that, under naturalistic moonless night skies and in a nulled geomagnetic field (disabling the moth's known magnetic sense[4]), moths flew in their seasonally appropriate migratory directions. Visual interneurons in different regions of the moth's brain responded specifically to rotations of the night sky and were tuned to a common sky orientation, firing maximally when the moth was headed southwards. Our results suggest that Bogong moths use stellar cues and the Earth's magnetic field to create a robust compass system for long-distance nocturnal navigation towards a specific destination.

The Bogong moth (*Agrotis infusa*; Fig. 1a) is an iconic and threatened migratory insect endemic to Australia. During the austral spring, newly eclosed Bogong moths make a highly directed migration of up to 1,000 km to a geographically restricted assemblage of high alpine caves in the extreme southeast of the country (Fig. 1b), a place that they have never previously visited. After 3–4 months of dormancy (aestivation; Fig. 1c), the same individuals return to their breeding grounds during autumn to reproduce, after which they die[1,2]. We have recently shown[4] that these moths steer flight during the long-distance phase[6] of their migration by sensing the Earth's magnetic field and correlating its inherent directional information with one or more visual landmarks, potentially using these landmarks as temporary orientation beacons along the route. However, exactly which natural visual features are used by Bogong moths for navigation remains unclear. A dark feature on the nocturnal horizon, one or more stars, or the moon are all possible cues. In the austral night sky, the constellations of stars and the bright Milky Way are particularly prominent and have excellent potential as reliable navigational cues[7,8]. Their predictable celestial positions over the course of each night make stellar cues useful for determining an arbitrary compass direction for short-range orientation (for example, for dung beetles[9–11] and cricket frogs[12]) and as a geographically relevant compass for long-range navigation

(for example, for birds[13,14], humans[15,16] and possibly seals[17]). However, to our knowledge, no invertebrate is currently known to use the stars for discerning specific geographical directions (that is, a direction relative to north) for directed long-range navigation to a distant goal. Here we show that moths have this ability and use the stars as a compass during the long-distance phase[6] of their migration to determine and follow an inherited migratory bearing, reversing this bearing when the season changes. Moreover, we show that visual interneurons in the moth's brain are tuned to specific orientations of the night sky, suggesting their involvement in a stellar compass network.

## Moths navigate under natural night skies

Using a light trap located at Mount Selwyn, about 70 km north-northeast of the main alpine aestivation sites in southeast New South Wales, we captured migrating Bogong moths departing northwards during their autumn return migration (Fig. 1b). Captured moths were transferred to our nearby experimental site and tethered and flown within a modified Mouritsen–Frost flight simulator[3–5] that continuously measured the instantaneous orientations of vigorously flying moths that were free to turn in any azimuthal direction (details of the simulator, the tethering and the criteria used to select moths for analysis are provided in the

[1]Lund Vision Group, Department of Biology, University of Lund, Lund, Sweden. [2]Neural Circuits and Evolution Laboratory, The Francis Crick Institute, London, UK. [3]Department of Entomology, Nanjing Agricultural University, Nanjing, China. [4]Department of Psychology, Queens University, Kingston, Ontario, Canada. [5]Institute for Biology and Environmental Sciences, University of Oldenburg, Oldenburg, Germany. [6]Department of Biochemistry and Molecular Biology, University of Southern Denmark, Odense, Denmark. [7]College of Asia and the Pacific, Australian National University, Canberra, Australian Capital Territory, Australia. [8]Department of Applied Biosciences, Macquarie University, Sydney, New South Wales, Australia. [9]School of Engineering, University of South Australia, Adelaide, South Australia, Australia. [10]Research School of Biology, Australian National University, Canberra, Australian Capital Territory, Australia. [11]National Collections & Marine Infrastructure, CSIRO, Canberra, Australian Capital Territory, Australia. [12]Centre for the Advanced Study of Collective Behaviour, University of Konstanz, Konstanz, Germany. [13]These authors contributed equally: David Dreyer, Andrea Adden. [14]Deceased: Barrie Frost. ✉e-mail: David.Dreyer@biol.lu.se; Andrea.Adden@crick.ac.uk; Eric.Warrant@biol.lu.se

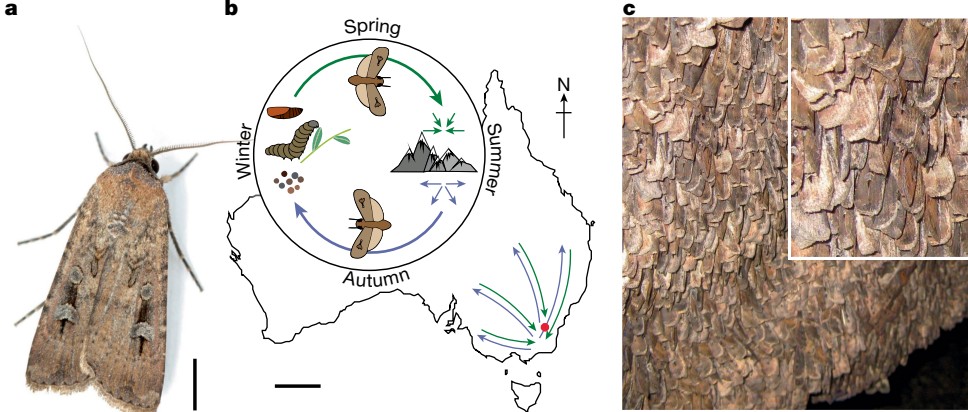

**Fig. 1 | The Bogong moth life history. a**, A male Bogong moth. Scale bar, 5 mm. Photo: A. Narendra, from ref. 2. **b**, Adult moths migrate from their breeding grounds in various regions of southeast Australia to the Australian Alps during spring (green arrows), where they aestivate in cool alpine caves over summer, and return to the breeding grounds in autumn (purple arrows). At the breeding grounds, they mate, lay eggs and die. Immature stages develop underground during the winter. The red dot indicates the experimental site at Adaminaby. Scale bar, 500 km. **c**, Around 16,000 moths per m² aestivate on the walls of specific caves in the Australian Alps for up to 4 months before making the return migration. Inset: close-up image of the moths.

Methods and Extended Data Fig. 1). To test whether Bogong moths fly in their inherited migratory directions under natural night skies, two transparent ultraviolet (UV)-transmissive non-magnetic cylindrical flight simulator arenas were placed on an open hilltop (Methods and Extended Data Fig. 1a,b). Moths tethered within the arena had a full view of the surrounding landscape (meadows, trees and hills) as well as of the entire dome of the night sky. They also experienced the natural geomagnetic field.

If Bogong moths use the stars for maintaining a geographically specific, inherited migratory heading, they need to deal with the seasonal differences in the appearance of the stars (Fig. 3a), as well as with the nightly movements of the stars across the sky that result from the rotation of the Earth. Such movements substantially shift and rotate the stripe of the Milky Way and the positions of the stars as the night progresses, therefore creating a potential problem for migrating animals that rely on them to navigate a straight course. Early during the night (20:49 ± 17 min), under a cloudless autumn starry sky with a waxing gibbous half-full moon, moths flown in the two arenas were significantly oriented northward, in their expected migratory direction (Fig. 2a; $n = 95$, $\alpha = 5°$). When tested 3 h later (23:42 ± 17 min), when the stars and the moon had shifted substantially (and the sky was still cloudless), the same cohort of moths (which had been swapped between arenas to eliminate landmarks as possible cues) retained their northward orientation (Fig. 2b; $n = 95$, $\alpha = 355°$), with their earlier and later mean orientation directions being indistinguishable (likelihood-ratio test on population means for all 95 moths, early versus late: $\chi^2 = 2.663$, d.f. = 1, $P = 0.103$). If one exclusively considers moths that were highly oriented both earlier and later in the evening ($r > 0.8$ in both trials: 35 out of 95 moths), this finding is only reinforced (Extended Data Fig. 2a). Thus, Bogong moths not only orient in their expected migratory direction under natural night skies, but they also do so despite the nightly movements of the stars and moon.

The dim pattern of polarized light[18] associated with the moon is also a potential compass cue. Lunar polarized light is used by nocturnal dung beetles for straight-line orientation[19] but, when the moon is absent, they instead revert to the Milky Way, and are disoriented when the sky is obscured by clouds[9]. However, substantial migrations of Bogong moths also occur around the new moon (E.W. and B.F., unpublished data), implying that, if polarized light is used as a compass cue, it is not essential.

Notably, on nights on which the moon and stars were obscured by heavy cloud, Bogong moths were still oriented in a northward direction (Fig. 2c; $n = 44$, $\alpha = 13°$), with a mean orientation direction that was not significantly different to either of the two mean directions (Fig. 2a,b) recorded on clear nights (likelihood-ratio tests comparing mean orientation direction in Fig. 2c with those in Fig. 2a,c: $\chi^2 = 0.007$ and 0.242, d.f. = 1, $P = 0.935$ and 0.886, respectively). Thus, even when the stars and moon are unavailable, Bogong moths can still navigate in their inherited migratory direction, strongly suggesting reliance on the only known remaining compass cue—the Earth's magnetic field.

The most parsimonious explanation for these results is that, in overcast conditions, Bogong moths rely on the geomagnetic field as their compass for navigation, which raises the critical question of whether Bogong moths use a stellar compass at all.

## Bogong moths use a stellar compass

To investigate whether Bogong moths use a stellar compass, we captured Bogong moths at Mount Selwyn that were migrating roughly southwards into the Australian Alps in spring and back northward during autumn (Fig. 1b). Moths were transferred to a purpose-built dark ferromagnetic-free laboratory at our field site and tethered and flown in a flight simulator (details of the laboratory are provided in the Methods and Extended Data Fig. 1c–i). The simulator arena was placed at the centre of a three-axis Helmholtz coil system that was used to null (that is, remove) the Earth's magnetic field and therefore disable the influence of the moth's magnetic sense. Moths were subjected to a natural clear moonless austral night sky (matching the date and time of the experiment) projected at natural intensity onto a flat screen placed on top of the dark-walled arena (Fig. 3a) and their flights were recorded for 5 min.

Under naturally oriented night skies (geographical north 0°; Fig. 3a) and in the absence of a magnetic field, moths flew in their seasonally appropriate migratory directions[4] during two springs and two autumns in 2018 and 2019 (Fig. 3c): roughly southwards in spring ($n = 70$, mean vector direction $\alpha = 168°$) and north-northwest in autumn ($n = 54$, $\alpha = 341°$). Under the natural sky stimulus rotated by 180° (arena-centric geographical north now at 180°), the moths flew in almost exactly the opposite direction (Fig. 3d), turning by 187° in spring ($n = 68$, $\alpha = 355°$) and by 174° in autumn ($n = 56$, $\alpha = 167°$). When the stars of the natural sky were randomly distributed (identical light intensity, no directional information: Fig. 3b), moths were disoriented in both seasons (spring: Fig. 3e, $n = 58$; autumn: Fig. 3f, $n = 49$). Full statistics are provided in Extended Data Fig. 2g and Supplementary Table 1.

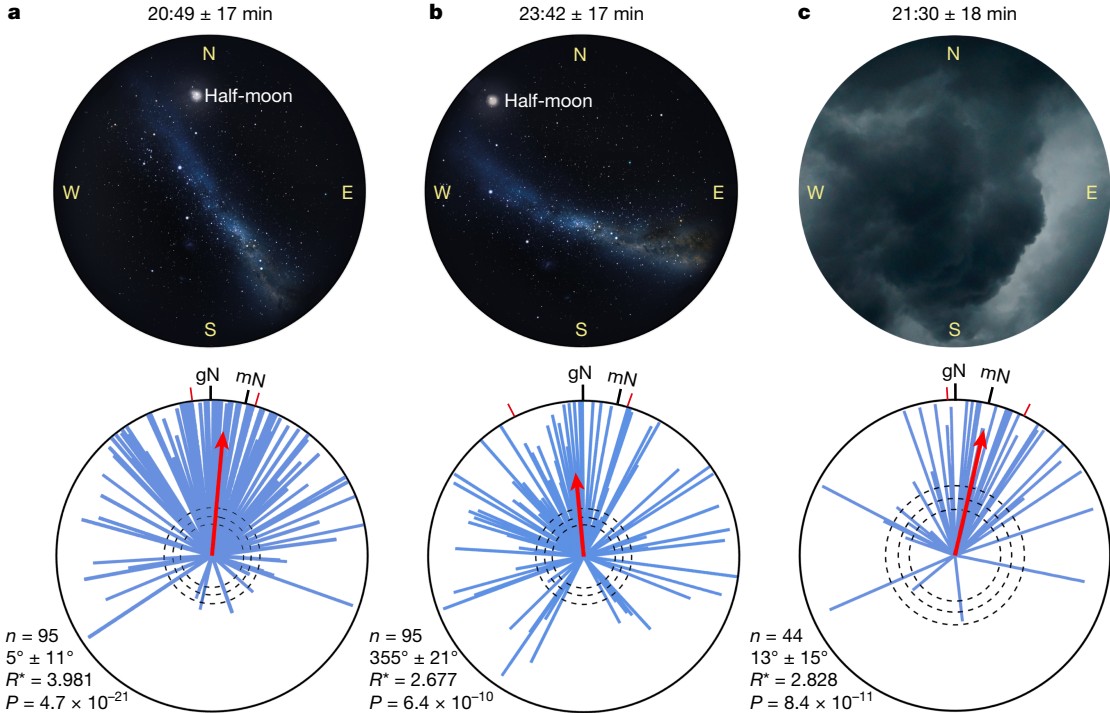

**Fig. 2 | Bogong moth navigation under natural night skies. a,b**, The orientation directions of 95 migratory moths (blue vectors) under a cloudless autumn night sky (Adaminaby, over 3 nights at the end of March 2023), early in the evening (**a**; 20:49 ± 17 min, $\alpha$ = 5°, 95% confidence interval = 22°, $R^*$ = 3.981, $P$ = 4.23 × 10⁻²⁰) and again 3 h later when the half-moon and stars had substantially shifted their positions (**b**; 23:42 ± 17 min, $\alpha$ = 355°, 95% confidence interval = 42°, $R^*$ = 2.677, $P$ = 5.76 × 10⁻⁹). Sunset occurred at 19:00 ± 4 min. Moths were tethered in a clear cylindrical UV-transmissive Perspex arena with a full view of the sky and the surrounding landscape. **c**, The orientation directions of 44 migratory moths (blue vectors) under a completely overcast autumn night sky (Adaminaby, end of March 2023) at 21.30 ± 18 min ($\alpha$ = 13°, 95% confidence interval = 30°,

$R^*$ = 2.828, $P$ = 7.56 × 10⁻¹⁰). Each red mean vector (MV) in **a**–**c** results from weighting the mean directions and mean directedness (vector lengths) of all of the individual moths. Statistical analysis was performed using one-sided Moore's modified Rayleigh test with Bonferroni correction for multiple comparisons; corrected $P$ = 4.23 × 10⁻²⁰ (**a**), 5.76 × 10⁻⁹ (**b**) and 7.56 × 10⁻¹⁰ (**c**) (Methods). The direction and directedness of mean vectors is given by $\alpha$ and $R^*$, respectively. The dashed circles show the required $R^*$ value for statistical significance: $P$ < 0.05, $P$ < 0.01 and $P$ < 0.001, respectively for increasing radius. The red radial dashes show the 95% confidence intervals. gN, geographical north; mN, magnetic north. Sky images in **a** and **b**, Stellarium.

Moth flight directions during spring and autumn, and under rotated and unrotated skies in either season, were significantly different (Mardia–Watson–Wheeler tests; Extended Data Fig. 2g). These results show that Bogong moths can use the distribution of stars within the austral night sky alone as a navigational compass to distinguish between specific geographical directions, thereby determining their required migratory direction, during both their southward spring migration and their northward autumn migration. When these stars were rearranged randomly, this ability disappears. Apart from humans[15,16] and several night-migratory songbirds[6,13,14], to our knowledge, no other animal is known to use the stars to navigate in a specific geographical direction. Birds use the centre of stellar rotation to determine north or south[6,13,14] and, if Bogong moths do the same, this might explain their ability to hold an inherited migratory course despite the nightly movements of the stars (Fig. 2a,b). Alternatively, Bogong moths might possess a time-compensated star compass that can account for the predictable movements of the stars, analogous to the time-compensated sun compass of Monarch butterflies[3]. However, exactly which aspects of the starry sky are used by Bogong moths remains to be determined.

In summary, our behavioural results demonstrate that Bogong moths, like night-migratory birds[20], use a stellar compass and a geomagnetic compass and, like birds[6,21], remain oriented in their inherited migratory direction when at least one is available, either the natural stars alone (Fig. 3c,d) or the Earth's magnetic field alone (Fig. 2c). When neither cue is available the moths are disoriented (Fig. 3e,f).

## The neural basis of the stellar compass

To explore the neural basis of the Bogong moth's stellar compass, we recorded intracellularly from visual neurons in three regions of the Bogong moth brain[22,23]: the optic lobe, the central complex (CX; the insect navigation centre[24]) and the lateral accessory lobes (LAL; the steering centre[25]). We measured neural responses to the same stimuli that induced behavioural responses in our laboratory flight experiments (dorsal starry sky, nulled magnetic field). Restrained moths were mounted in the apparatus with an eastward orientation relative to the initial starry sky. During experiments, we rotated the sky through 360° from this starting point. Impaled neurons ($n$ = 28) revealed a peak spiking frequency at a specific sky rotation angle $\varphi_{max}$ (that is, at a specific compass direction of the moth; Fig. 4a). Their responses fell into four physiological categories: (1) unimodally excited; (2) unimodally inhibited; (3) rotation-direction-selective unimodal; or (4) rotation-direction-selective bimodal (Fig. 4a,b and Extended Data Fig. 3). While unimodal neurons showed a single activity peak (maximal excitation or maximal inhibition), bimodal neurons showed two peaks separated by approximately 180°. Responses of neurons in categories 1 and 2 were largely insensitive to movement direction, with clockwise and anticlockwise sky rotations leading to similar responses. By contrast, category 3 and 4 neurons responded differently to clockwise and anticlockwise rotations. Despite the diversity of response types, $\varphi_{max}$ was consistent across repetitions, with the majority of cells having a tuning variability of 20° or less (Fig. 4c and Extended Data Figs. 4–6).

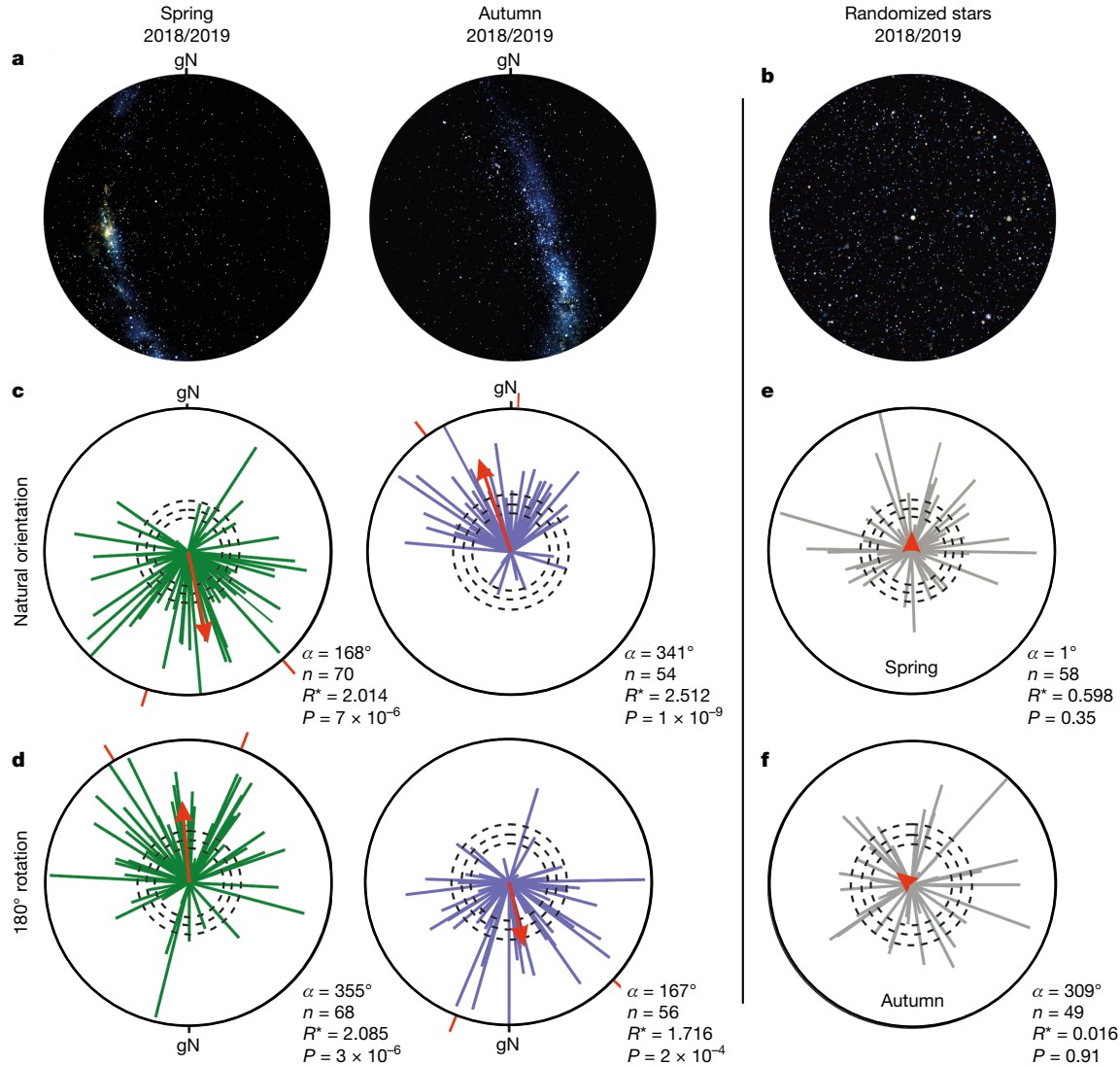

**Fig. 3 | A stellar compass in Bogong moths. a,b**, Images of a laboratory-projected natural night sky during spring and autumn (**a**), and an autumn sky with its stars randomly arranged (**b**). **c**, The orientation directions of migratory moths under naturally oriented night skies, and in a nulled magnetic field, during spring 2018 and 2019 (green vectors, left; $n = 70$, $\alpha = 168°$, 95% confidence interval = 58°, $R^* = 2.014$, $P = 6.3 \times 10^{-5}$) and autumn (purple vectors, right; $n = 54$, $\alpha = 341°$, 95% confidence interval = 41°, $R^* = 2.512$, $P = 9.0 \times 10^{-9}$). **d**, As for **c**, but for night skies rotated by 180° (spring: $n = 68$, $\alpha = 355°$, 95% confidence

interval = 53°, $R^* = 2.085$, $P = 2.7 \times 10^{-5}$; autumn: $n = 56$, $\alpha = 167°$, 95% confidence interval = 69°, $R^* = 1.716$, $P = 1.8 \times 10^{-3}$). **e,f**, The orientation directions of migratory moths under randomized starry skies during spring (**e**; $n = 58$, $R^* \approx 0.598$, $P = 0.35$) and autumn (**f**; $n = 49$, $R^* \approx 0.016$, $P = 0.91$) in 2018 and 2019. All other definitions and statistics are as described in Fig. 2. Bonferroni correction for multiple comparisons was used; corrected $P = 6.3 \times 10^{-5}$ (spring, **c**) and $9.0 \times 10^{-9}$ (autumn, **c**), $2.7 \times 10^{-5}$ (spring, **d**) and $1.8 \times 10^{-3}$ (autumn, **d**). Sky images in **a**, Stellarium.

However, when randomized stars (Fig. 3b) were rotated as a control, cells failed to show tuning (Figs. 4e and 5b). As the control shares local contrast and motion cues, its inability to drive the recorded neurons suggests that these neurons are not simply movement detectors but are tuned to features present in the natural starry sky.

To further test the selectivity of these responses, a subset of six cells was also stimulated with artificial compass cues (of natural intensity) that mimicked aspects of the Milky Way (Fig. 4d): its brightest region around the Carina nebula (revolving dot) and its stripe-like shape (rotating bar). All but one of the cells responded to both cues (dot, $n = 6$; bar, $n = 5$) and these qualitatively resembled their response to the starry sky, with similar azimuthal tuning (deviations from starry sky $\varphi_{max}$ were clustered around 0° across neurons; Fig. 4d). Whereas the maximum deviation from the starry sky $\varphi_{max}$ was larger for the dot (52°) than the bar (15°), the responses to the bar were weaker than those to the starry sky or the dot (Fig. 4e). Thus, the bar more closely reflected the cell's azimuthal tuning to the starry sky, while the dot reflected its response

strength. This suggests that these neurons are suited to encode at least two features of the starry sky: the extended shape of the Milky Way and its brightest region around the Carina nebula.

Notably, across the entire population of unimodal cells, $\varphi_{max}$ values consistently clustered in the Southern Hemisphere, independent of recording season or response type (Fig. 4b; $n = 18$, $\alpha = 171° \pm 35°$). Thus, in the reference frame of a Bogong moth flying under a natural (stationary) sky, unimodal cells fire maximally when moths head southwards (irrespective of season: a Mardia–Watson–Wheeler test indicates that the spring and autumn mean $\varphi_{max}$ values do not significantly differ: $P > 0.2$). Notably, head direction cells recorded in the medial pallium of fledgling shearwater chicks show a similar behaviour—all fire maximally when the chick's head is directed towards magnetic north (even though the chick will eventually fly due south on its inaugural migration)[26]. In fixed quiescent Monarch butterflies, CX neurons recorded extracellularly in response to a rotating 'sun' stimulus (a bright green LED at 30° elevation), also revealed narrow $\varphi_{max}$ tuning[27]. However, when allowed

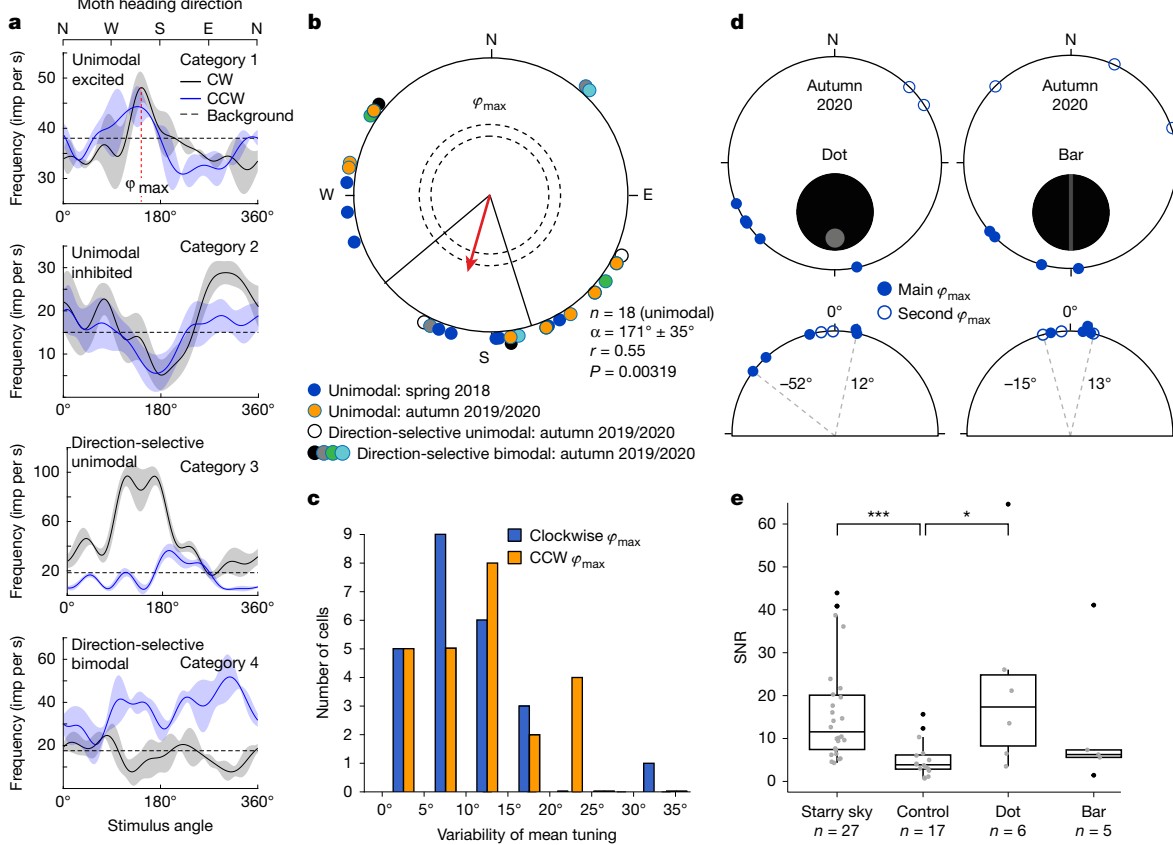

**Fig. 4 | Brain visual neurons respond to rotations of the night sky. a**, Examples of general response categories (1–4) for 360° clockwise (CW; grey curves) and anticlockwise (CCW; purple curves) night sky stimulus rotations (mean ± s.d. tuning curves show spike impulses (imp) per s). In the moth's reference frame, flying under a natural (stationary) sky, 0° = 360° indicates that it is heading north; 90° indicates that it is heading east; 180° indicates that it is heading south; and 270° indicates that it is heading west. $\varphi_{max}$ is defined at the top (red dashed line). The horizontal dashed lines show the pre-stimulation response level. **b**, The $\varphi_{max}$ for all cells (note that bimodal cells have two $\varphi_{max} \approx 180°$ apart; separate plots for all four cell categories are shown in Extended Data Fig. 3). The red vector gives the mean $\varphi_{max}$ ($\alpha$) and the vector length ($r$) for all 18 unimodal cells (one-sided Rayleigh test, spring and autumn pooled, $P = 0.00319$; 95% confidence limits (black straight lines), ±35°). The dashed circles show the required $r$ value for statistical significance: $P < 0.05$ and $P < 0.01$, respectively for increasing radius. Unimodal cells fire maximally when moths head

southwards (171° ± 35°) under natural skies (irrespective of season). **c**, The variability in $\varphi_{max}$ for successive clockwise (average 9°) or anticlockwise (average 10°) sky rotations in individual cells. **d**, $\varphi_{max}$ for dot and bar control stimuli in a subset of unimodal and bimodal cells (autumn 2020). The empty circles show the second $\varphi_{max}$ for bimodal cells. The bottom row shows the angular difference between the cell $\varphi_{max}$ for dot/bar stimuli and its starry sky $\varphi_{max}$ (quoted ± angular limits, dashed lines). **e**, The response SNR for different stimuli (grey dots, included data; black dots, outliers). For the box plots, the box limits show the 25th to 75th percentiles, the centre line shows the median value, and the whiskers extend to 1.5 × interquartile range. The SNR is the maximum response during sky rotation (that is at $\varphi = \varphi_{max}$) divided by the standard error before rotation. Statistical analysis was performed using two-sided Wilcoxon rank-sum tests; *$P = 0.052$, ***$P = 0.00011$. Sky/control data: 2018, 2019 and 2020; dot/bar data, 2020.

to fly and receive self-motion cues this tuning disappeared, revealing dependence on behavioural state[27]. As our Bogong moths were likewise fixed and quiescent, we cannot rule out that the narrow $\varphi_{max}$ tuning we report is also due to behavioural state.

The functional diversity of neurons also corresponded to a morphological diversity. Of the 28 neurons that responded to our stimuli, we anatomically identified 11 cells by intracellular tracer injections (Fig. 5a). The identified neurons were distributed across all three brain regions targeted in our recordings: the optic lobes, the CX and the LAL. While the optic lobe is responsible for primary visual processing, the CX is a higher-order integration centre for spatial orientation, navigation and action selection[28,29]. The LAL is the major output target of the CX and generates steering commands to downstream motor circuits[25]. For visually guided migration in Bogong moths, all three brain regions are therefore expected to have key roles in processing visual compass cues and transforming these signals into steering commands. While the optic lobe neurons showed sharply tuned responses with a high signal-to-noise ratio (SNR), the responses of the two CX neurons that we held long enough to record from were less

pronounced (Fig. 5 and Extended Data Figs. 4 and 7; cell 181122_003), indicating that visual stimulation alone is insufficient to fully drive these cells. The three anatomically identified category 4 (bimodal) cells localized to the LAL and the ventrolateral and ventromedial protocerebrum—their direction-selective responses are suggestive of a potential role in steering. The six identified category 1 and 2 (unimodal) cells localized to the optic lobe and CX, locations that are more concerned with early-stage sensory processing (optic lobe) and heading encoding (CX).

Although we obtained only a sparse sample of neurons, and despite the constraints of in vivo intracellular electrophysiology, our data nonetheless demonstrate that information from the starry sky is encoded in the moth's brain. While these neurons are located in brain regions that are known to analyse multiple navigational cues in other insects[24], surprisingly, our identified cell types have not previously been described in analyses of insect compass pathways[24] or head-direction circuits[30] (Fig. 5a and Extended Data Fig. 7). Further work is clearly required to understand these neural circuits in detail, and whether their properties vary with behavioural state.

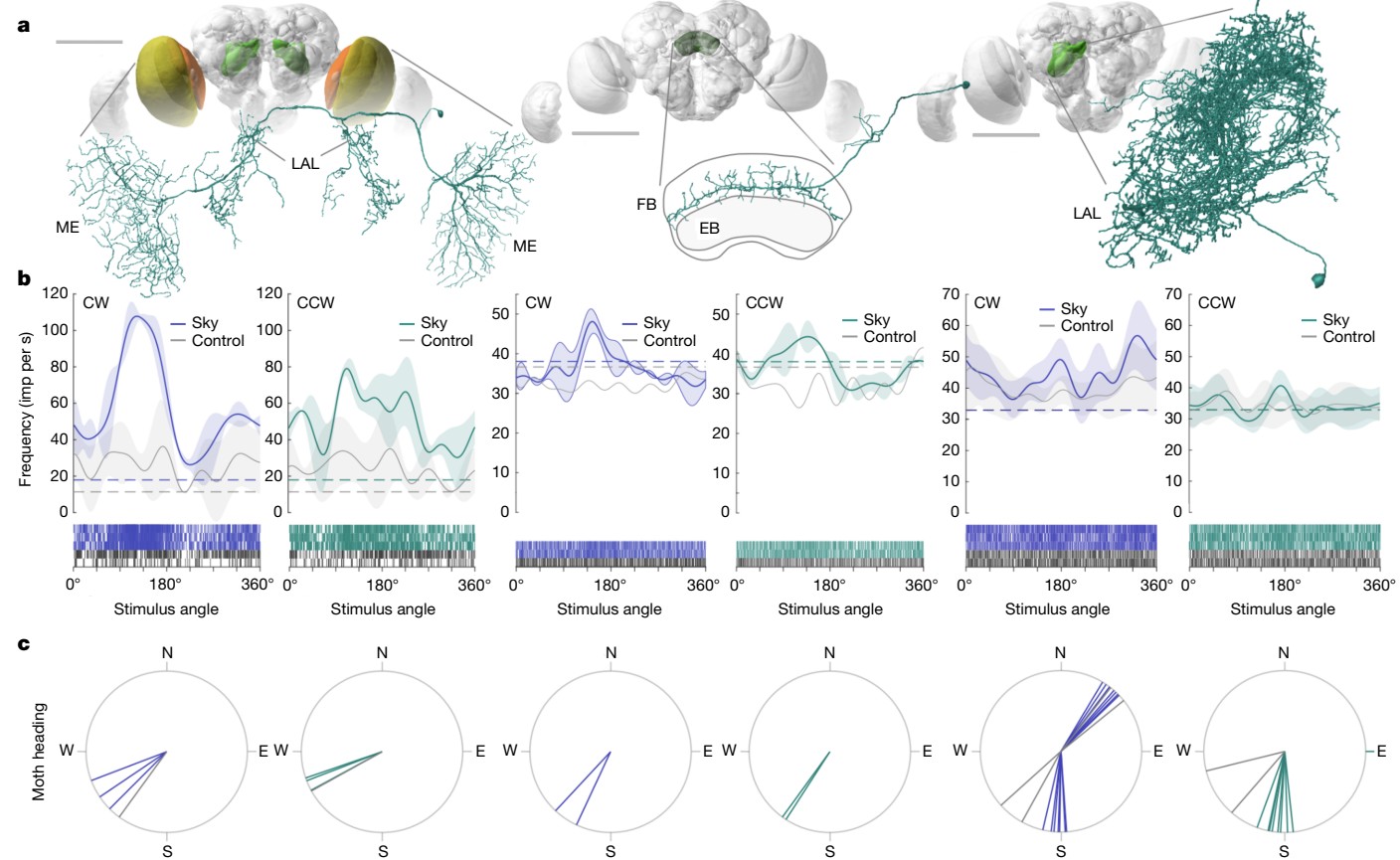

**Fig. 5 | Neurons responding to the starry sky reside in known compass-related regions of the insect brain. a**, Neuron morphologies for a neuron interconnecting both optic lobes and innervating the LAL of the central brain (medulla (ME) to medulla, left), a central-complex neuron innervating the fan-shaped body (FB; middle) and an LX neuron (right). EB, ellipsoid body. Scale bars, 500 μm. **b**, Raster plots of spike data shown with the mean tuning curves for clockwise and anticlockwise rotating starry skies (blue/green curves) and rotating randomized stars (control, grey curves). Other conventions are as described in Fig. 4a. **c**, Peak tuning directions ($\varphi_{max}$) of individual cells following clockwise and anticlockwise sky rotations (the rotation angle when cellular firing frequency was maximal, derived from spike train raster plots in **b**). $\varphi_{max}$ values correspond to moth headings relative to north. The blue/green lines show the starry sky $\varphi_{max}$ values, and the grey lines show the $\varphi_{max}$ for the dot stimulus if tested.

## The control of long-distance navigation

Our results indicate that a stellar compass is sufficient on its own to guide Bogong moths during their migration, and this may become useful if the geomagnetic field becomes perturbed by a magnetic storm or by local magnetic anomalies[31] (which are admittedly not strong in southeast Australia, and typically alter field strength by only a few per cent). Under natural conditions, a geomagnetic compass is also sufficient if the stars become obscured by cloud. When neither cue is available, moths are disoriented. Moreover, we have also shown that Bogong moths possess previously undescribed neurons in the navigational centres of the brain that respond to a specific orientation of the night sky. Notably, these neurons are all tuned to a common azimuth, firing maximally when the moth is heading southwards, irrespective of season (Fig. 4b).

Newly eclosed Bogong moths therefore rely on stellar and geomagnetic compass cues to target a distant and restricted geographical region they have never previously visited (during spring), relying on the same global cues for the reverse migration (during autumn). For Bogong moths, how these cues are interpreted to determine the desired migratory heading depends on three spatiotemporal factors: the moth's geographical origin (different breeding populations across southeast Australia migrate in different directions to reach the same target), the season (moths migrate towards the mountains in spring and return to the breeding grounds in autumn) and the time of night

(stellar positions change due to Earth's rotation). As in other insects[24], to move in a desired (migratory) heading direction, Bogong moths must first establish their current heading (for example, using global compass cues), compute the angular difference between their current and desired headings and, depending on geographical location, season and time of night, generate a steering command that turns the moth to its required migratory direction. While head direction is well understood to be encoded by the CX[24,30,32], our understanding of how desired heading is encoded is only now emerging[33,34]. However, consistent with data from flies[28,34–37], bees[38] and Monarch butterflies[33], the computation of desired migratory heading in Bogong moths could well take place in the fan-shaped body of the CX. Indeed, our identification of starry sky responses in a fan-shaped body input neuron (Fig. 5a) supports this notion. However, exactly how the fan-shaped body of Bogong moths accounts for geographical location, season and time of night to generate a desired migratory heading remains unclear.

## At least two compass mechanisms

The question remains of what aspects of a moonless austral night sky might be useful as compass cues for Bogong moths. Humans[15,16] and birds[6,13,14] can use individual stars, or constellations of stars, as compass cues for distinguishing specific geographical directions, but it remains unclear how many of the brightest stars are discernible by the moth's small compound eyes. However, the Milky Way—which, in the Southern

Hemisphere, is a bright extended stripe of light that becomes brightest somewhere in the southern half of the sky[7,10]—is very likely visible to Bogong moths[39], as also suggested by our physiological results (Fig. 4). Its shape and celestial position at any time of night are predictable from one night to the next throughout each migratory season, and its rotation around the South Celestial Pole over any single night retains the brightest parts of the Milky Way (including the Carina nebula) somewhere in the southern half of the sky, but not consistently due south. Exactly where the brightest parts are located depends on season and time of night and can vary from southeast (Figs. 2a and 3b) to almost due west (Fig. 3a). Thus, a simple phototactic mechanism, with Bogong moths orienting towards the brightest parts of the sky in spring and away from them in autumn, seems unlikely. Furthermore, the presence of a bright moon in different parts of the sky would severely disrupt such a mechanism. However, Bogong moths orient in their inherited migratory direction even when the moon is present (Fig. 2a,b).

Notably, the Southern Hemisphere Milky Way is used to determine an arbitrary compass direction for short-range orientation in ball-rolling dung beetles[7,9,10], not for distinguishing a specific geographical direction to travel towards a distant goal, but as a visual guide to keep a straight bearing for a few minutes in any random (goalless) direction. A straight trajectory is the safest for taking the beetle away from the furious competition of the dung heap. By contrast, Bogong moths, navigate long distances over many nights towards a distant destination they have never previously visited and therefore require a global compass to distinguish and hold their specific, inherited, geographical migratory heading. While the Earth's magnetic field certainly acts as one such compass[4] (Fig. 2c), the southern night sky clearly acts as another, with each of these compasses probably taking over from the other when the salience of either diminishes or fails (Figs. 2c and 3c,d). However, exactly which celestial features are used for the stellar compass, whether the common directional tuning of stellar compass neurons has any role in the computation of desired heading, and how stellar, magnetic and any other hitherto unknown sensory cues en route are behaviourally and neurally integrated for robust navigation (including how and whether they are calibrated against each other[40]), all remain enticing topics for future research.

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

## Methods

### Capture and care of moths

Bogong moths (*A. infusa*) of both sexes were caught in the wild during their autumn and spring migrations (2019 and 2018) using a LepiLED insect light (www.gunnarbrehm.de), or a vertical beam search light (model GT175, Ammon Luminaire Company), placed in front of a white sheet suspended between two trees. Almost all of the animals were caught near the Mount Selwyn Snowfields (southeast New South Wales, Australia: 35.914° S, 148.444° E; elevation, 1,600 m), which is approximately 70 km north-northeast of the nearest aestivation cave in the Main Range of the New South Wales Alps. Thus, to reach these caves in spring, these moths (a tiny subset of all moths travelling to the mountains in a multitude of directions from across southeast Australia) would be expected to fly south-southwest in spring, and returning moths might be expected to travel north-northeast in autumn (which agrees with our behavioural results). A few animals were also caught near Thredbo (Dead Horse Gap, southeast New South Wales, Australia: 36.524° S, 148.260° E, elevation 1,580 m). These moths were used for electrophysiology only. Each captured moth was transferred to its own plastic container to isolate it from influence by other moths. After capture, moths were transported to the testing site Glenhare, a rural property near Adaminaby New South Wales (36.040° S, 148.864° E; elevation, 1,250 m), fed with 20% honey solution (in water) and stored in a cool and sheltered place (exposed to the natural light cycle) to recover from stress induced by capture.

### Laboratory for controlled indoor experiments

A purpose-built ferromagnetic-free laboratory located at Glenhare, Adaminaby (built on a concrete slab reinforced with fibreglass and constructed entirely from non-magnetic materials) housed the indoor behavioural and electrophysiological experiments (Extended Data Fig. 1c). Each experimental apparatus (behaviour and electrophysiology) has its own dedicated earth separated from the mains earth (through a 6-mm thick, 30-mm wide and 12-m long copper strap dug into the ground below the concrete slab). Background levels of radio-frequency disturbances at this rural site are extraordinarily low[4]. All of the experiments were performed on dark-adapted moths in darkness at night (beginning at least 1 h after sunset). Darkness was achieved with black-out blinds (to remove residual starlight and moonlight from outside) and dark cloth around the experimental apparatus (to shield from the minimal stray light emitted from the equipment).

**Non-magnetic electrophysiological apparatus.** The non-magnetic electrophysiological apparatus (such as table and animal mounts; Extended Data Fig. 1f,g) was constructed from Thorlabs aluminium optomechanical components using high-grade stainless-steel fasteners. Vibration isolation between the aluminium pillar legs and the aluminium bread board table (on which the moth and manipulators were mounted) was provided by four high-grade stainless steel Stillpoints Ultra 6 (with Ultra base) isolators (Stillpoints). The moth was mounted (see below) onto a pillar attached to the bread board table, and a custom-built non-magnetic Sensapex piezo micromanipulator (Sensapex Oy, Oulu), also attached to the pillar, was used to move and advance a glass microelectrode. A removable circular UV-transmissive Perspex disc (diameter, 250 mm; thickness, 5 mm), covered in a layer of UV-transmissive diffusing paper (Lee Filters 251 1/4 white diffuser) and mounted 127 mm above the moth, was used for projection of celestial visual stimuli (see below). The electrophysiological apparatus was placed at the centre of a computer-controlled, double-wrapped[41] three-axis (3D) Helmholtz coil system custom built in aluminium and copper (University of Oldenburg workshop; outer coil diameters; *x*, 900 mm; *y*, 835 mm; *z*, 775 mm) to create a nulled magnetic field (Extended Data Fig. 8) around the experimental moth. These coils were mounted onto the experimental table holding the moth and manipulators. The coil systems were powered by constant-current power supplies (Kepco, BOP 50-2M) and the current running through the coil systems was controlled through High-Speed USB Carriers (USB-9162, National Instruments) and custom-written codes in MATLAB (v.2019a and 2022b, MathWorks). Further details were reported previously[3,4]. Before each experimental session, Meda FVM-400 magnetometer measurements ensured that the magnetic field was nulled within the apparatus (Extended Data Fig. 8b).

**Non-magnetic behavioural apparatus.** The non-magnetic behavioural apparatus (Extended Data Fig. 1h,i) consisted of a modified Mouritsen–Frost flight simulator[3–5,42] used to record the virtual flight path of tethered migratory Bogong moths. In brief, each flight simulator consisted of a cylindrical Perspex arena (diameter, 50 cm; height, 35 cm) placed vertically onto an aluminium table with a clear Perspex top within a 3D Helmholtz coil system (as described above, but with coil outer dimensions: *x*, 1,245 mm; *y*, 1,300 mm; *z*, 780 mm). Again, the nulled magnetic field conditions within the coils were carefully monitored using the Meda FVM-400 magnetometer (Extended Data Fig. 8a). The arena walls were covered with two layers of black felt. An optical encoder (E4T Miniature Optical Kit, US Digital) was mounted in the middle of a UV-transmissive Perspex disc (diameter, 50 cm; thickness, 0.5 cm), which was placed on top of the arena like a lid. A layer of UV-transmissive diffusing paper (Lee Filters 251 1/4 white diffuser) was placed on top of the disc (and served as a screen for dorsal projection of celestial stimuli; see below). A fine vertical tungsten rod (the encoder shaft: diameter, 0.5 mm; length, 153 mm), inserted into the axial centre of the optical encoder, extended downwards into the arena and allowed the attachment of tethered flying moths (see below). We used the encoder manufacturer's software (USB1 Digital Explorer 1.07, US Digital) to continuously record the moth's heading relative to geographical north (gN), therefore allowing us to reconstruct the moth's virtual flight path in the presence of celestial visual cues. An LED projector (ASUS S1 Mobile), neutral density (ND) filters (optical density between 4 and 5 log units) and a mirror placed at 45° under the Perspex tabletop were used to project a dim moving (10 mm s⁻¹) pattern of optic flow onto a screen (Lee Filters 251 1/4 white diffusing paper) placed beneath the arena and therefore below the tethered flying moth (Extended Data Fig. 1d). The direction of the optic flow was controlled by custom written software (M. York), which coupled the encoder system (USB1 or USB4 encoder data acquisition USB device, US Digital) through a feedback loop. Thus, the optic flow would always move from head to tail below the moth, instantaneously changing direction as the moth changed direction. The mean radiance of the optic flow at the location of the performing moth was $2.06 \pm 0.19 \times 10^9$ photons cm⁻² s⁻¹ sr⁻¹.

**Stimulation with natural starry skies.** A natural moonless starry austral night sky, in both the electrophysiological and the behavioural rigs, was projected using a downward pointing projector (behavioural rig: LED projector ASUS S1 Mobile; electrophysiological rig: Sony MP-CL1A laser projector; spectra of both projectors are shown in Extended Data Fig. 9d). Each projector was mounted sufficiently high above the moth in each rig to provide clear and correctly sized dorsal images of the night sky. To avoid unwanted stray light, each projector was housed in a custom-built 3D-printed plastic box featuring an opening in front of the lens and ventilation slits above the projector.

The free software Stellarium[43] was used to simulate the moonless starry night sky over Canberra (about 80 km from Adaminaby as the crow flies) at 21:30 on four respective dates: 1 October 2018 (spring 2018), 21 March 2018 (autumn 2018), 21 October 2019 (spring 2019) and 27 February 2019 (autumn 2019). Screenshots (screen resolution 7,480 × 720 pixels) of these simulated starry skies were taken, cut in a circular shape using CorelDRAW X5 and saved as PNG files (300 dpi) to create the stimulus images (Fig. 3a,b and Extended Data Fig. 9a).

Subtended at the moth, celestial images projected onto the circular screens in each rig (see above) provided 160° (behavioural rig) and 100° (electrophysiological rig) fields of view of the starry night sky centred on the zenith. Celestial images contained grey level values ranging from 4 (darkest) to 255 (brightest, on a scale of 0–255), with an average grey level of 62. The quality of the night sky provided by these images was comparable to that provided by the natural rural night sky at Glenhare (as measured with a Unihedron Sky Quality Metre; Extended Data Table 1). Before each experiment, the PNG files were opened using IrfanView64 on a PC using a screen resolution of 1,280 × 720 pixels. The PC was connected through an HDMI cable to the projector. The projected size of the circular sky matched the diameter of the circular screen in each rig.

Before each experiment, we ensured that the projection was centred accurately and that its light intensity was reduced to starlight levels using ND filters (the behavioural rig is shown in Extended Data Fig. 9b). In the case of the electrophysiological rig, two 1.2 log unit ND filters were inserted into the light path to dim the image to starlight levels (Extended Data Fig. 9c). The projection of the starry sky was initially set to its natural orientation relative to geographical north or flipped along its vertical and horizontal axes to test the moths under a 180° rotated sky.

As the projectors did not emit UV light, we installed a custom-made LED-ring (built by T. McIntyre; diameter, 120 mm; inner diameter, 50 mm) featuring eight UV LEDs (LED370E Ultra Bright Deep Violet LED, Thorlabs) in front of the projector. The brightness of the LED-ring was controlled using custom written software using MATLAB (v.2019a and 2022b, MathWorks) and several layers of ND filters that were fixed in front of the LED-ring to bring the UV intensity into a quasi-natural range (the behavioural rig is shown in Extended Data Fig. 9d).

Upward light intensity and spectrum measurements of the starry night sky were made underneath the projection screen in each rig with the probe located at the position of the moth (Extended Data Fig. 9b–d). The light metre used was a calibrated Ocean Optics QE65PRO Spectrometer (Ocean Optics).

**Stimulation with randomized starry skies.** To create randomized starry skies for experiments in autumn 2018 and spring 2018, the positions of all individual pixels of the natural night sky stimulus image were reassigned randomly to new positions, and the resulting randomized images were likewise saved as PNG files (Extended Data Fig. 9a). These featureless stellar conditions provided an identical stimulation intensity but provided no celestial spatial information. The randomized starry skies used during spring and autumn 2019 were improved by randomizing groups of pixels containing individual stars, therefore retaining the stars but removing spatial variations in the night sky (such as the Milky Way) that could be used for orientation (again maintaining identical intensity). Here, the stimulus image (of autumn 2019) was subdivided into squares with a size of 13 × 13 pixels, as the brightest star in the image had roughly these dimensions. The positions of these squares were now randomly reassigned and the resulting image was saved as a PNG file. A final improved randomized stimulus (used during spring 2019) was generated from the test stimulus by randomizing the positions of the individual visible stars. This was achieved by first detecting the position and size of each star in the test stimulus using a multiscale Laplacian of a Gaussian convolution of a greyscale version of the test stimulus, followed by local maxima detection. The resulting spatial information was then used to extract and save the image of each star from the natural night sky stimulus, before replacing them on a new background image, with a uniform colour and intensity equal to that of the mean of all pixels in the test stimulus that were not part of a star. This way, the location of the stars on the randomized image could be drawn from any desired distribution. In this case, a uniform distribution was used for the location of all but the brightest star, which was placed in the centre of the image.

**Stimulation with artificial Milky Way cues.** To test the selectivity of central brain visual neurons to various parts of the Milky Way during electrophysiological experiments, we stimulated cells with artificial compass cues that, like the starry sky stimulus, filled the projection screen above the moth (Fig. 4d). These cues mimicked aspects of the autumn Milky Way: its brightest region around the Carina nebula (revolving dot) and its stripe-like shape (rotating bar). The bar length and width were chosen to mimic the main stripe of our natural Milky Way stimulus. The intensities of these artificial stimuli were also adjusted to provide a good mimic of the intensities of the stars and the dark background sky in our natural stimulus (see above) and had the same mean grey level (61 ± 1 for values between 0 to 255, darkest to brightest).

### Apparatus for behavioural experiments under an open sky in a natural landscape

Two ferromagnetic-free Mouritsen–Frost flight simulators (of the same type used in the laboratory) were placed on a hilltop at Glenhare and used to record the heading directions of tethered migratory Bogong moths experiencing the full local surrounding landscape and the entire dome of the natural sky. To achieve this, each flight simulator arena consisted of a transparent UV-transmissive Perspex cylinder (again with diameter, 50 cm; height, 35 cm), placed vertically onto an aluminium table (Extended Data Fig. 1a,b). The table top was also made of transparent Perspex. The two flight simulators (and their tables) were placed around 15 m apart. A tethered moth was connected to an optical encoder suspended at the centre of the open top end of each arena by a thin horizontal UV-transmissive Perspex arm. The two arenas were controlled from computers that were operated from within a black light-tight cubical tent that was placed midway between the two arenas, approximately 7 m from each. The tent was therefore a landscape feature that the moths could potentially see during their tethered flights.

### Behavioural procedures

**Indoor behavioural procedures.** Most behavioural procedures used in this study have been previously described[4,5]. Before attachment of tethering stalks, moths were chilled in a freezer for 5–10 min to immobilize them. The scales on the moth's dorsal thorax were removed by suction using a micro-vacuum pump (custom built by B.F.). Afterwards a thin vertical tungsten stalk (which is ferromagnetic free), fashioned at its end to create a small circular footplate, was glued to the dorsal thorax using contact cement while being restrained by a weighted-down plastic mesh. Moths were tested on the day of stalk attachment.

Shortly before sunset, UV-transmissive Perspex boxes holding individual stalked moths were placed onto an elevated outdoor location and provided with a clear view of the western sky and the setting sun (and the skylight polarization pattern), in case these cues were important for calibrating compass mechanisms (as found for the magnetic compass of birds[44–46]). After sunset, the moths could also see the stars (and the celestial rotation).

For tethering within the behavioural arena (performed in dim red light to maintain a dark-adapted visual state), the arena lid holding the optical encoder was lifted and the tungsten stalk of a vigorously flying moth (held with medical forceps) was attached to the bottom end of the encoder shaft through a 1.5 cm length of thin rubber intravenous medical tubing that connected the stalk to the shaft (Extended Data Fig. 1e). Once the arena lid was returned to the arena, this coupling enabled the moth to rotate freely around its yaw axis and choose any flight direction. Once the moth was mounted in the arena, it was gently pointed manually towards geographical north. The heading direction count was then reset, the moth was released and the optical encoder was enabled to register the flight heading direction of the moth under a given night sky condition (projected on the arena lid) at a sampling rate of 5 Hz (and a horizontal resolution of 3°). Each moth was tested for exactly 5 min in each stimulus condition.

Moths chosen for analysis were required to fulfil three ante hoc criteria, two before the experiment and one during the experiment: (1) the tethering stalk was perfectly vertical; (2) wing flapping was vigorous and its amplitude was large and equal for both wings (indicating that the contact cement had not interfered with the wings); and (3) that the moth flew continuously for the full 5 min. For the last criterion, if a moth stopped flying, the arena was gently tapped in order to stimulate the moth to continue flight behaviour. A moth that stopped flying four times was rejected and the recording aborted. For the field seasons of spring 2018 and 2019, as well for Autumn 2019, the percentages of moths that were aborted due to failing to meet these criteria were 2 out of 59 or 3.4% (Spring 2018), 25 out of 137 or 18.3% (Spring 2019), and 35 out of 76 or 46.1% (Autumn 2019). The high rejection rate during Autumn 2019 may have been due to the unusually wet and cold weather that occurred during this season. Moths often behave erratically on rainy or stormy nights, and even on days before and after such nights.

**Outdoor behavioural procedures.** The same behavioural methods (and criteria) that were used indoors were also used outdoors. The goal of these experiments was to understand how migratory Bogong moths deal with natural skies (in particular the nightly movements of the stars and moon, as well as cloud cover) while experiencing a normal geomagnetic field and the local surrounding landscape under natural illumination. Experiments were performed under clear starry skies during autumn 2023 (over three nights during the last week of March), as well as at two times of night, to test whether the migratory orientations of moths were affected by the nightly movements of the stars and moon (which was approximately half-full): (1) between 20:32 and 21:06 (about 1.5 h after sunset), and (2) between 23:25 and 23:59. The same moths were used for orientation measurements at both times, and moths that were flown in one arena at the earlier time (and saw the black tent on their eastern side) were flown in the other arena at the later time (and now saw the black tent on their western side) and vice versa. Moreover, there was a stand of trees close to the arena on the eastern side of the tent, and single trees close to the arena on the western side of the tent. Thus, other panoramic landmarks (apart from the tent) differed markedly in their spatial positions from within the two arenas. Between earlier and later experiments, tethered moths were kept isolated and in the dark in a suitcase that was warmed with hot water bottles. Experiments were also carried out on a fourth completely overcast night that totally covered the stars and moon (between 21:12 and 21:48).

**Indoor electrophysiological procedures**
To maximize the success rate of the demanding intracellular recordings during the short migratory season, these experiments were performed both in the afternoon and during the natural nocturnal flight time of the moths. For afternoon experiments, we removed the two 1.2 log unit ND filters in front of the projector lens (as described above) to generate a starry sky projection around 250 times brighter than the one used at night (to account for the circadian-rhythm-induced light-adapted state of the moths). For night experiments, the ND filters were reinserted. No obvious differences were found in results obtained in the two situations. The moths were mounted onto a custom-made 3D-printed animal holder and immobilized using wax. The antennae were fixed to the front of the head with a drop of wax, and a square piece of cuticle was removed from the head capsule to expose the brain. The neural sheath was digested with Pronase (Sigma-Aldrich) for about 30 s and then carefully washed. It was then removed using a pair of fine forceps. A second small hole was cut into the cuticle above the proboscis muscle and a chlorinated silver wire was inserted into this muscle to serve as reference electrode.

Glass electrodes were pulled from borosilicate glass on a P-97 Flaming-Brown micropipette puller (Sutter Instrument), and had a typical resistance of 50–100 MΩ. The electrode tip was filled with Neurobiotin solution (4% Neurobiotin in 1 M KCl, Vector Laboratories), and the remainder of the electrode was filled with 1 M KCl. Electrodes were moved into position with a non-magnetic Sensapex micromanipulator. Signals were amplified using a BA-03X intracellular amplifier and headstage (NPI Electronic), and were then digitized using a CED Micro 1401-3 (Cambridge Electronic Design) and recorded with Spike2 software (v.8.03, Cambridge Electronic Design). Stimulus control signals from MATLAB (v.2019a and 2022b, MathWorks) were simultaneously recorded in Spike2. During the recording, the brain was kept hydrated through regular application of moth ringer solution[47].

Moths were mounted under starry skies that were naturally oriented before sky rotation (that is, had the same orientation as the stars outside the laboratory). In earlier experiments, the initial heading orientation of the mounted moth was north relative to the stars and, in later experiments, the initial heading orientation south; however, greatest ease of access to the brain was eventually found for an eastward heading. The action potential spike trains obtained for a 360° rotation of the sky were corrected according to the initial heading orientation of the moth so that spike trains obtained across all experiments were comparable (with a correction angle of 0° applied for an initial northward orientation, 90° for an eastward orientation and 180° for southward orientation). Impaled cells were stimulated with dorsally projected images of the natural starry night sky, or control images of randomized stars (see above). Cells were also stimulated with bars and dots that mimicked different parts of the Milky Way (see above). All of these projected images (natural and randomized stars, artificial Milky Way stimuli) were rotated 360° at 30° s$^{-1}$ to 45° s$^{-1}$, with a 2 s break between clockwise and anticlockwise rotations, using custom-written MATLAB code (v.2019a and 2022b, MathWorks). Between rotations, a neutral grey background image was presented that had the same average grey value as the starry sky/control images.

We targeted an area of the central brain in which we expected to find both CX and lateral complex neurons, as well as optic lobe neurons traversing the brain in the posterior optic tract. Neurons that clearly did not respond to an initial sky rotation were immediately discarded and no recording was saved. 79 neurons were assessed as potentially responding to the stimulus and, of these, 28 (35%) met the inclusion criteria of a unimodal or bimodal response profile. The remaining 51 neurons were classified as uniform in their response to stellar rotation (that is, showed no obvious response) and were therefore excluded from the analysis.

After recording from a suitable cell, a positive current (range: 1–3 nA for 3 min) was applied to the electrode to inject Neurobiotin into the cell. The electrode was then removed and the brain was dissected out of the head capsule. Brains were fixed in 4 °C overnight in paraformaldehyde solution (4% PFA in phosphate buffer) and then washed in 0.1 M PBS (4 times for 15 min). During washing, the retinas were removed. Brains were then incubated with streptavidin–Cy5 (Jackson Immuno Research, 1:1,000 in PBS with 0.3% Triton X-100) at 4 °C for 3 days and kept in the dark from this point onwards. After incubation, the brains were washed in PBS-Triton X-100 (6 times for 20 min) and PBS (2 times for 20 min) and then dehydrated in an increasing ethanol series (50%, 70%, 90%, 95% and twice at 100%). Brains were then transferred to a fresh mixture of methylsalicylate and ethanol (1:1) and, after 15 min, were left to clear in 100% methylsalicylate for 75 min. The cleared brains were mounted in Permount (Thermo Fisher Scientific) mounting media between two coverslips and left to dry for at least 2 days.

Brain samples were scanned with the 633 nm laser of a Leica SP8 confocal microscope and viewed with a ×20 oil-immersion objective (Leica Microsystems). For optimal resolution, the scan settings were set to 1,024 × 1,024 pixels, 12-bit pixel depth, 3 times line accumulation and 400 lines per s in the photon-counting mode of the hybrid detector. Neurons and relevant neuropils were then reconstructed in Amira v.5.3 (Thermo Fisher Scientific) and registered into the Bogong moth standard brain[22,23].

## Data analysis

The data used for the analyses described below are available online[48]. As there were no statistical differences in results obtained from male and female moths, the results were pooled.

**Indoor laboratory behaviour.** The behavioural analysis used in this study has been previously described[4,5]. As mentioned above, the encoder software (USB1 Digital Explorer v.1.07, US Digital) recorded the instantaneous heading directions of a tethered flying moth every 200 ms (5 Hz) and saved these values in a text file. We used custom-written MATLAB code (v.2019a and 2022b, MathWorks) to visualize the virtual flight paths of all tested moths and calculated a mean orientation vector based on each virtual flight path. Each vector for each moth in the circular plots (Fig. 3c–f) encodes the mean orientation direction of a moth's individual recorded flight path as well as its $r$ value (that is, length, or directedness, of the flight path vector). To take advantage of the extra information in our data arising from the fact that the flight trajectories of moths not only had a mean direction (as used for a classic Rayleigh test[49]) but also a mean directedness (vector length), we used the circular statistics software Oriana (v.4 (2011), KCS) and Excel (Microsoft Office 2019, Microsoft) to apply a one-sided Moore's modified Rayleigh test[4,50,51] with Bonferroni correction for multiple comparisons (Supplementary Table 1). The $R*$ value encodes the directedness of a population of tested moths and reveals the likelihood that the combined flight direction of these moths—each with its own direction and directedness—differs significantly from random.

To confirm that the two distributions of moth flight directions were significantly different between naturally oriented and 180°-rotated night sky conditions, as well as significantly different between spring and autumn for any single sky condition (Fig. 3c–f), a Mardia–Watson–Wheeler test[49] (Oriana) was used (Extended Data Fig. 2g).

**Outdoor behaviour.** The statistical procedures used for data collected outdoors were the same as those used for data collected indoors, with the exception that a likelihood-ratio test[52] was used to test whether mean orientation directions for moth populations flown under two different clear natural sky conditions (earlier and later in the evening) were significantly different from each other (Fig. 2a,b). We did this by fitting maximum-likelihood distributions to the mean orientation directions for each trial, using the circular statistics package circular[53] in R v.3.6.1 (www.r-project.org). In a similar manner, we determined the likelihood that the moth population flown under overcast conditions (Fig. 2c) had the same mean direction as either of the two populations flown under clear skies, in this case testing whether the means of the two different populations differed significantly.

Moreover, for the cohort of 95 moths flown under clear skies earlier and later in the evening (Fig. 2a,b), we used the mean orientation direction of each moth flown earlier (Fig. 2a), subtracted from the mean direction for the same moth flown later (Fig. 2b), and tested whether the mean angular differences were significantly different from zero (no change in heading). As the moth cohort was clearly less oriented later in the evening (possibly due to fatigue), we made this analysis by separating moths into three groups (Extended Data Fig. 2): (1) those that were well-oriented throughout the night ($r > 0.8$, both earlier and later in the evening, 35 out of 95 moths); (2) those that were well-oriented in only one trial (typically $r > 0.8$ earlier and $r < 0.8$ later, 42 out of 95 moths), and those that were less well-oriented throughout the evening ($0.2 < r < 0.8$ in both trials, 18 out of 95 moths). This analysis revealed that moths that were well oriented throughout the night (group 1) showed no change in heading, while those that were less-well oriented (groups 2 and 3) showed a westward drift later in the evening (Extended Data Fig. 2). However, this drift was not sufficient (in either direction or directedness) to significantly alter the mean direction of the population as a whole from earlier to later in the evening (Fig. 2a,b).

**Electrophysiology.** Spike train data were analysed using custom-written code in MATLAB (v.2019a and 2022b, MathWorks). Circular statistical analysis was performed in R (v.3.6.1; www.r-project.org) using the circular maximum-likelihood estimation package (CircMLE[54]). Responses were classified as unimodal (models M2A, M2B or M2C in the R-library CircMLE) or bimodal (models M4A or M4B) based on the Akaike information criterion. Only responses that were classified as unimodal (models M2A, M2B or M2C in the R-library CircMLE, based on the Akaike information criterion) were analysed further with respect to the half-width of the rotation tuning curve, the SNR and the variability of the response.

For cells that responded to night sky rotation (rotation angle $\varphi$) with an increase in spiking activity (which became maximal at the sky rotation angle $\varphi_{max}$), the bootstrapped interquartile range of the unimodal part of the response distribution was taken to be an estimator of the half-width of the rotation tuning curve ($\varphi$ range for 50% maximal response or greater). For cells that responded with a decrease in spiking activity, von Mises and mixed-von Mises distributions did not appropriately reflect this inhibition. Instead, the responses of these neurons were binned at 1° intervals and low-pass filtered, and the width at half-maximum was then determined in MATLAB (v.2019a and 2022b, MathWorks). The signal-to-noise ratio SNR was calculated in MATLAB as the ratio between the maximum response during night sky rotation (which occurs at a rotation angle $\varphi = \varphi_{max}$), and the s.e.m. before the night sky rotation started. This was calculated for each stimulus rotation separately. To obtain an estimate of the reliability of a given cell's response to repeated rotations of the night sky, the mean preferred stimulus rotation angle ($\varphi_{max}$) was calculated as the circular mean for each individual night sky rotation. We then used the circular s.d.[55] of the mean angle across all rotations as an estimate for the variability in the response.

**Statistics and reproducibility.** Behavioural data under natural clear starry skies (Fig. 2a,b) were obtained in independent experiments over three separate nights (with new moths used each night). Behavioural data under natural overcast skies (Fig. 2c) were obtained during only a single night. Behavioural data under projected starry skies generated using Stellarium (Fig. 3c–f) were obtained in independent experiments run over several nights with each night's cohort of moths used only once. These experiments were repeated with the same results over two separate spring and autumn seasons (2018 and 2019). Thus, all of these experiments deal with biological replicates. Sample sizes were based on the availability of moths, with at least 40 moths (and as many as 70) being used for each experiment. Experiments were conducted without blinding and without a specific randomization protocol (apart from the random selection of moths used as replicates). Intracellular electrophysiological recordings (Fig. 4) and dye injections (Fig. 5) were obtained from single visual cells in the brain that were subjected to rotations of a projected starry sky (generated using Stellarium). Since each recording (and subsequent dye injection) was a unique observation, replication was not possible. This is due to the stochastic nature of intracellular recordings—single neurons were penetrated randomly from target brain regions that contain many thousands of cells.

## Reporting summary

Further information on research design is available in the Nature Portfolio Reporting Summary linked to this article.

## Data availability

The raw data supporting the findings of Figs. 2 and 3 are available at Figshare[48] (https://doi.org/10.6084/m9.figshare.25780197.v2). The raw and experimental data supporting the findings of Figs. 4 and 5 are available at the above Figshare repository and in Extended Data Figs. 4, 5, 6 and 7, respectively. Source data are provided with this paper.

## Code availability

The code used to generate stimuli and to collect and analyse data in this study are available at GitHub (https://github.com/stanleyheinze/Starry_sky_code).

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

**Acknowledgements** This paper is dedicated to the memory of our friend and collaborator Barrie Frost who passed away in 2018. We thank the workshop staff at the University of Oldenburg and L. Fredriksson for manufacturing much of the experimental apparatus; G. Cohen, T. Reber, A. Lefevre, W. Souter, C. Sullivan, D. Gutierrez and J. Zeil for help in the field; A. Narendra for the photograph of the Bogong moth shown in Fig. 1; A. Hastie for logistical support in Canberra; T. Greville, T. Khan and the staff of the New South Wales National Parks and Wildlife offices in Jindabyne and Tumut for their assistance in the Kosciuszko National Park; and the members of the Lund Vision Group and M. Dacke, J. Zeil and B. el Jundi for discussions and suggestions and for reading the manuscript. This research made use of the Stellarium planetarium[43]. We acknowledge the European Research Council (advanced grant number 741298 to E.W., starting grant number 714599 to S.H., synergy grant 810002 to H.M.), US Air Force Office of Scientific Research (grant number FA9550-14-1-0242 to E.W.), the Australian Department of Defence (Defence Science and Technology Group to E.W.), the Royal Physiographic Society of Lund (to E.W.), the Swedish Research Council (Vetenskapsrådet, grant number 621-2012-2205 to E.W.), the German Research Council (Deutsche Forschungsgemeinschaft, grant number 395940726, SFB 1372 'Magnetoreception and navigation in vertebrates' to H.M.) and the Natural Sciences and Engineering Research Council of Canada (NSERC, grant number 353-2009 to B.F.) for their ongoing support. The research was performed under a Scientific Licence issued by the New South Wales National Parks and Wildlife Service (no. SL100806) and took place on the traditional lands of the Walgalu and Ngarigo peoples, to whose elders—past, present and future—the authors pay their respect.

**Author contributions** D.D., B.F. and E.W. conceived the project. K.G. contributed detailed knowledge of Bogong moth ecology and aestivation sites and provided critical logistical support in the Australian Alps. M.W., G.H. and J.C. provided crucial logistical, experimental and intellectual input. E.W., D.D., A.A., B.F., H.C., S.H. and H.M designed the experiments, and designed and manufactured the laboratory and experimental apparatus. D.D., A.A., H.C., J.X., J.W. and E.W. performed the experiments. D.D. and J.W. calibrated the magnetic stimulation apparatus and wrote the MATLAB scripts to control it. D.D., A.A., H.C., S.H., J.F., E.W. and B.F. analysed the results. A.A., S.H., D.D. and E.W. made the figures. E.W., A.A., S.H and D.D. wrote the initial version of the manuscript and all of the authors made substantial contributions to the final version.

**Funding** Open access funding provided by Lund University.

**Competing interests** The authors declare no competing interests.

**Additional information**
**Correspondence and requests for materials** should be addressed to David Dreyer, Andrea Adden or Eric Warrant.

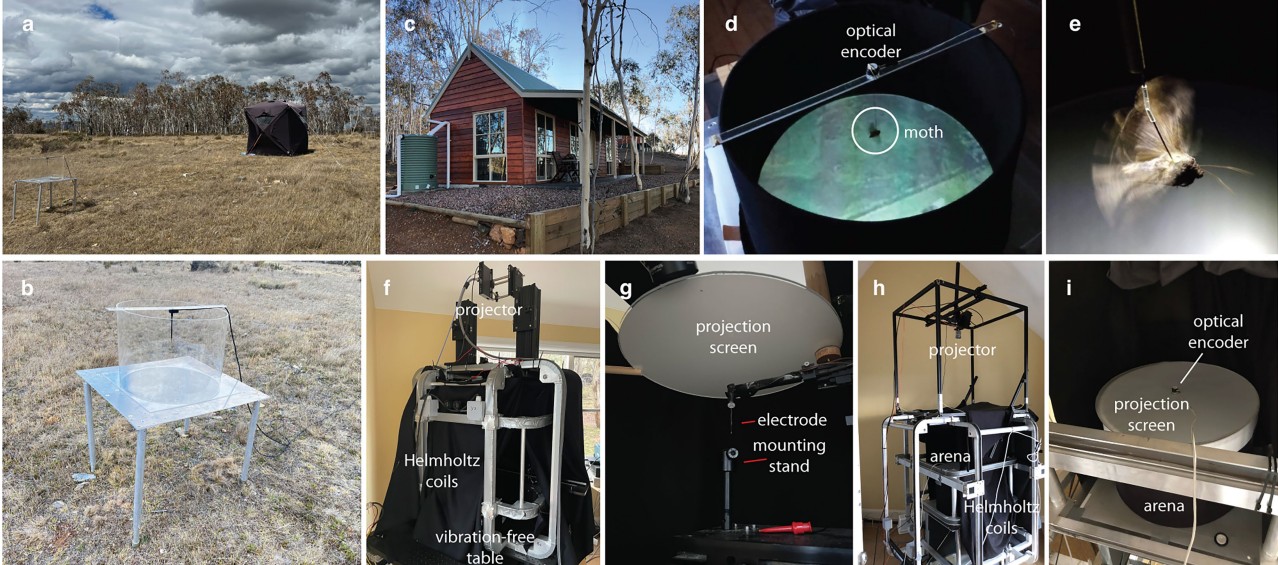

**Extended Data Fig. 1 | Methods for studying long-range navigation in migratory Bogong moths. a**, The two transparent hilltop arenas (sitting on their aluminium tables) seen on either side of the black tent enclosing the experimenters and the computers. **b**, A close-up of one of the arenas seen in **a**. Photographs Hui Chen. **c**, Our new Australian lab, built of entirely non-magnetic materials (to eliminate magnetic artefacts), at Adaminaby. The laboratory occupies 50 m², has two separate rooms (one for the electrophysiology rig, and one for the behavioural rig) and sits on a concrete slab. Each rig has its own earth separated from the mains earth (provided by a 6 mm thick, 30 mm wide and 12 m long copper strap dug into the ground below the concrete slab). **d**, The flight arena, with a flying moth (*circled*) tethered to an optical encoder that tracks changes in the moth's heading direction (relative to North). The flying moth is free to turn in any direction, and the encoder thus records its virtual flight trajectory over time. The floor of the arena displays an optic flow pattern of starlight-intensity (here brightly shown in order to see it) which always moves from head to tail as the moth turns, inducing it to fly. **e**, The tethering arrangement. **f**,**g**, The non-magnetic electrophysiology rig for recording the responses of compass cells in the moth's brain in response to precise magnetic and visual stimulation. The rig is enclosed within Helmholtz coils (**f**) to precisely control the direction and inclination of the Earth's magnetic field (or to null the magnetic field, as in the experiments described here), and a projector and screen (**g**) create natural images of the starry night sky above the moth. The moth is mounted horizontally on a stand beneath the projection screen and an electrode is inserted into its central brain. **h**,**i**, The non-magnetic behavioural rig for monitoring the trajectories of tethered flying moths. The flight arena, again enclosed within Helmholtz coils, receives an image of the starry night sky on a projection screen installed above the moth (**i**), from a projector mounted above the coils (**h**). Photographs Eric Warrant.

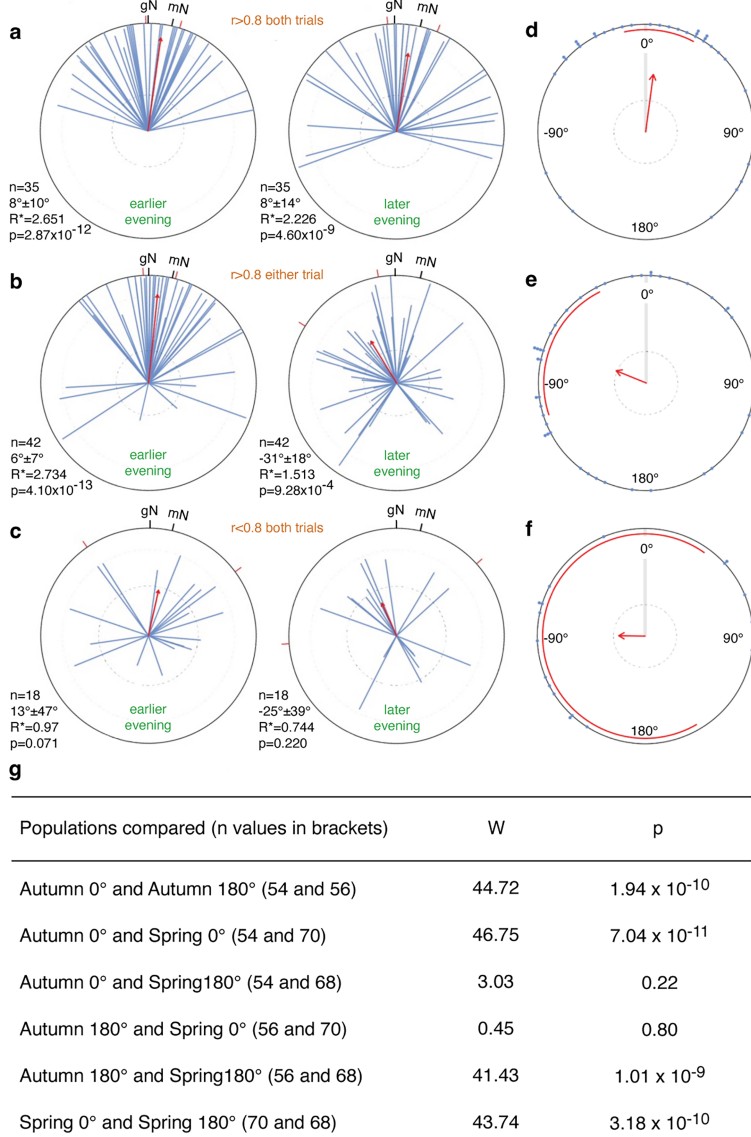

**Extended Data Fig. 2 | Bogong moth orientation under natural clear night skies, split by heading stability (a-f) and under naturalistic starry skies in the lab (g). a–c**, Orientation directions under a cloudless autumn night sky earlier and later in the evening (data from Fig. 2a,b). **d**–**f**, Change in mean population heading (Δα) from earlier to later in the evening. **a**,**d**, Individual moths that were well-oriented in both trials (r > 0.8 in both trials, n = 35) retained the same heading (**a**), exhibiting no change in direction between trials. At a population level, the pattern of changes (**d**) was not significantly different from 0° ($\chi^2$ = 0.428, d.f. = 1, p = 0.5131, Δα = 7.64°E). **b**,**e**, Moths that were well-oriented in only one of the two trials (r > 0.8 in either trial, n = 42), typically earlier in the evening, started the evening with a population mean heading similar to those that were well-oriented throughout the night (left panels in **a** and **b**), but drifted to the West later in the evening (**b**, right panel), a drift (**e**) that was significantly different from 0° ($\chi^2$ = 7.454, d.f. = 1, p = 0.0063, Δα = 67.77°W). **c**,**f**, For moths less well-oriented throughout the night (r < 0.8 in both trials, n = 18) there was no significant population level orientation (**c**), and no significant population-level difference (**f**) from a 0° heading change ($\chi^2$ = 1.769, d.f. = 1, p = 0.1835, Δα = 89.17°W). Nonetheless, the mean headings, despite lacking

significance, were similar to those of the moths in **b**. **a**–**c**, Direction and directedness of MV given by α and $R^*$ value, respectively. *Inner dashed circle*: required $R^*$ value for statistical significance: $p$ < 0.05. *Red radial dashes*: 95% confidence intervals (CI$_{95}$). mN = magnetic North, gN = geographic North. One-sided Moore's modified Rayleigh tests with Bonferroni correction for multiple comparisons – corrected p-values: $1.72 \times 10^{-11}$ (earlier) and $2.76 \times 10^{-8}$ (later) in **a**; $2.46 \times 10^{-12}$ (earlier) and $5.57 \times 10^{-3}$ (later) in **b**; 0.426 (earlier) and 1.000 (later) in **c**. **d**–**f**, *Red arrow*: the bias-corrected mean vector for the maximum-likelihood distribution of heading changes; *Dashed circle*: required $r$ value for statistical significance: $p$ < 0.05; *Red arc*: 95% confidence intervals (CI$_{95}$) of the mean angle across heading changes (numerical simulation). One-sided likelihood-ratio tests[52] adjusted within each dataset (from **a**, **b** and **c**) using the Benjamini-Hochberg method[56]. **g**, Mardia-Watson-Wheeler tests for determining whether populations of moth directions (n) in different seasons and under different skies (as presented in Fig. 3c,d) were significantly different from each other (if so, *p* is small). 0° is the natural starry sky orientation and 180° is the starry sky rotated by 180°.

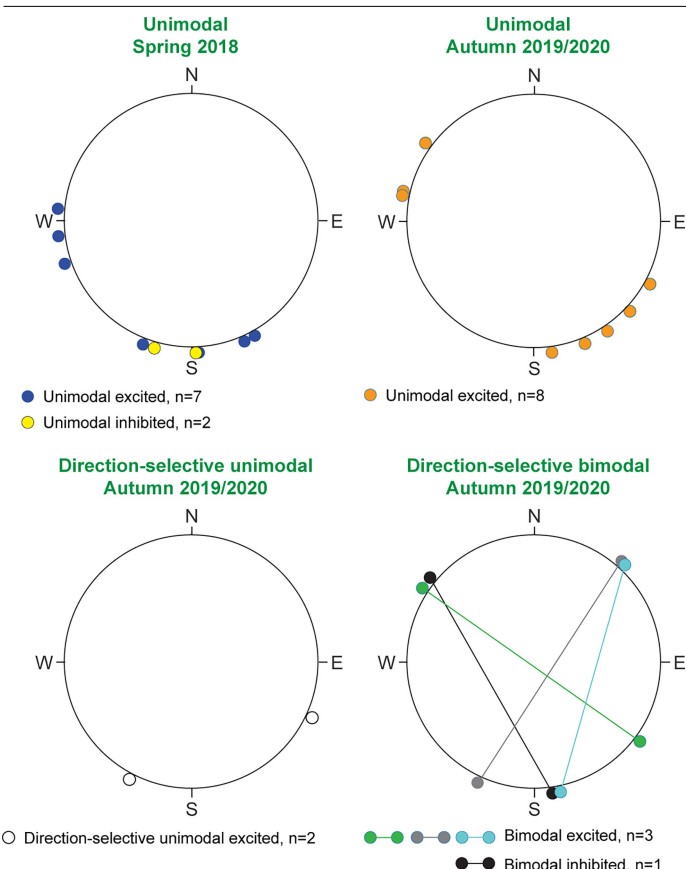

**Extended Data Fig. 3 | $\varphi_{max}$ values in different categories of visual neurons in the Bogong moth central brain.** $\varphi_{max}$ values were derived from neural responses to a rotating starry sky stimulus, subdivided into response type and seasons.

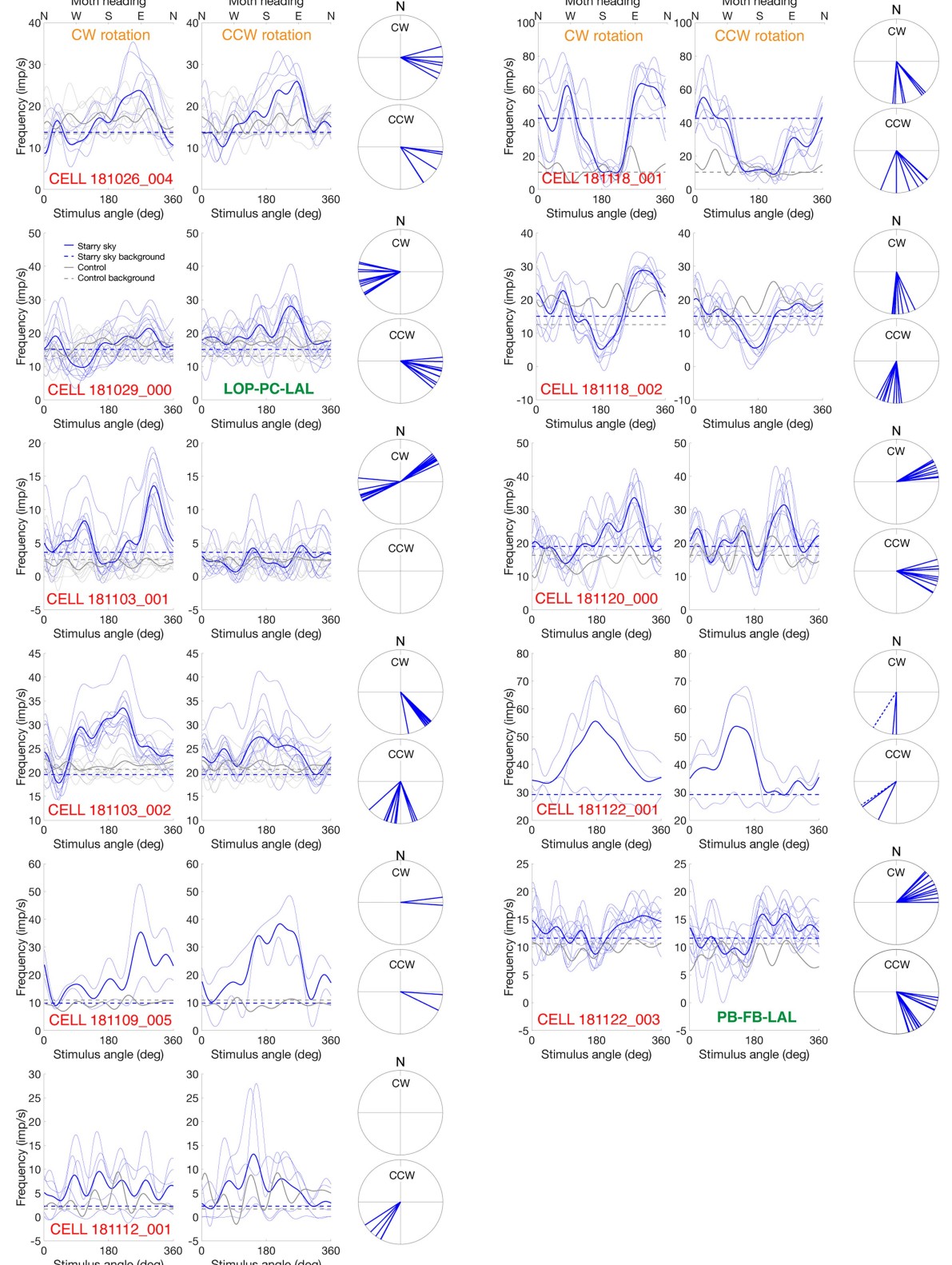

**Extended Data Fig. 4 | Visual cells recorded during the spring migration (October and November 2018).** Frequency plots and circular $\varphi_{max}$ diagrams, separated into clockwise (CW, *left panels*) and counter-clockwise (CCW, *right panels*) stimulus rotations, for every cell recorded in spring 2018 (cell ID, red text). Responses to the starry sky stimulus are shown in *blue*, responses to the control stimulus are shown in *grey*. Where the cell's anatomy was identified, branching regions are also given (shown as *green text*). For detailed anatomies, see Extended Data Fig. 7.

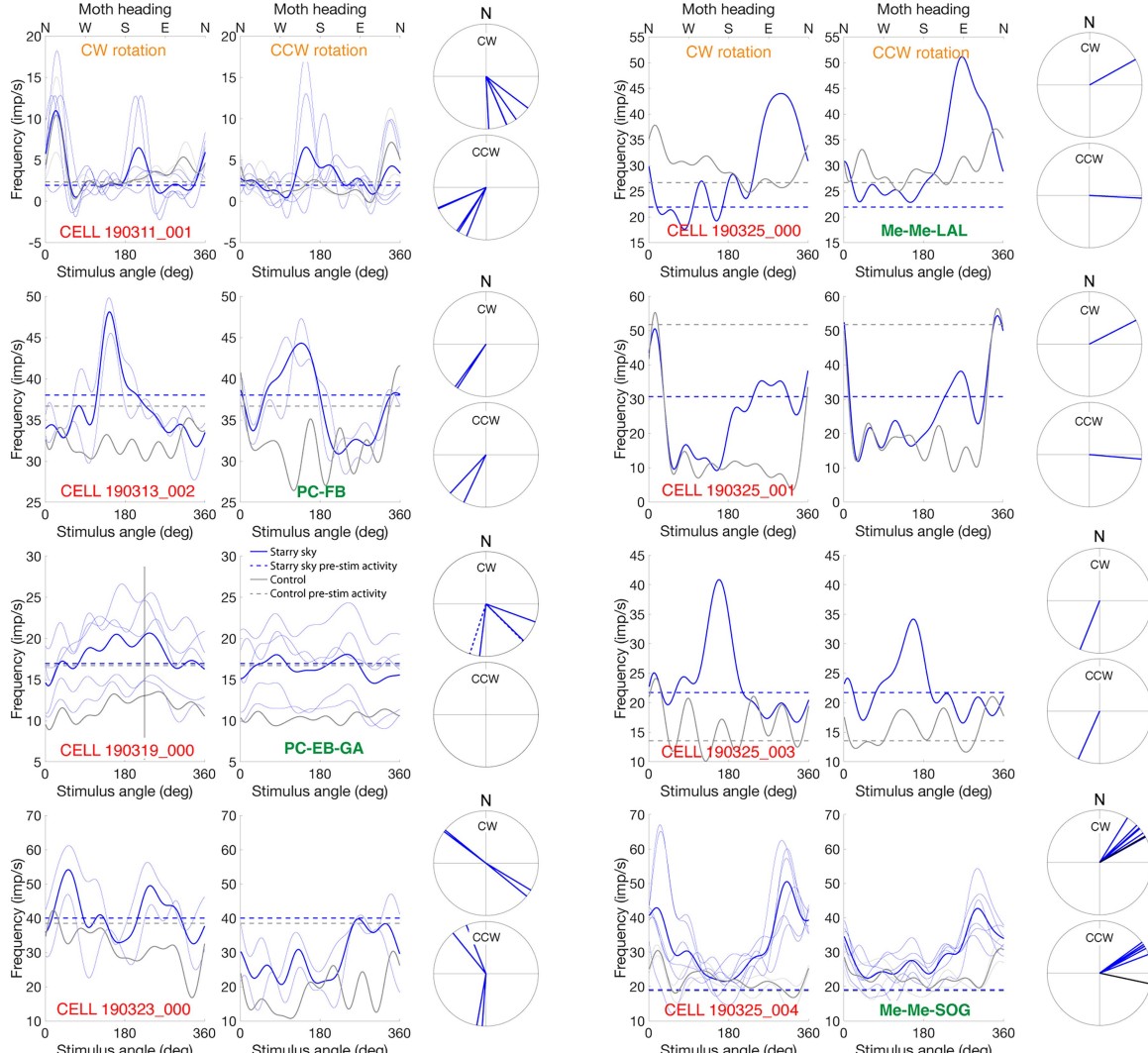

**Extended Data Fig. 5 | Visual cells recorded during the autumn migration (March 2019).** Frequency plots and circular $\varphi_{max}$ diagrams, separated into clockwise (CW, *left panels*) and counter-clockwise (CCW, *right panels*) stimulus rotations, for every cell recorded in autumn 2019 (cell ID, *red text*). Responses to the starry sky stimulus are shown in *blue*, responses to the control stimulus are shown in *grey*. Where the cell's anatomy was identified, branching regions are also given (shown as *green text*). For detailed anatomies, see Extended Data Fig. 7.

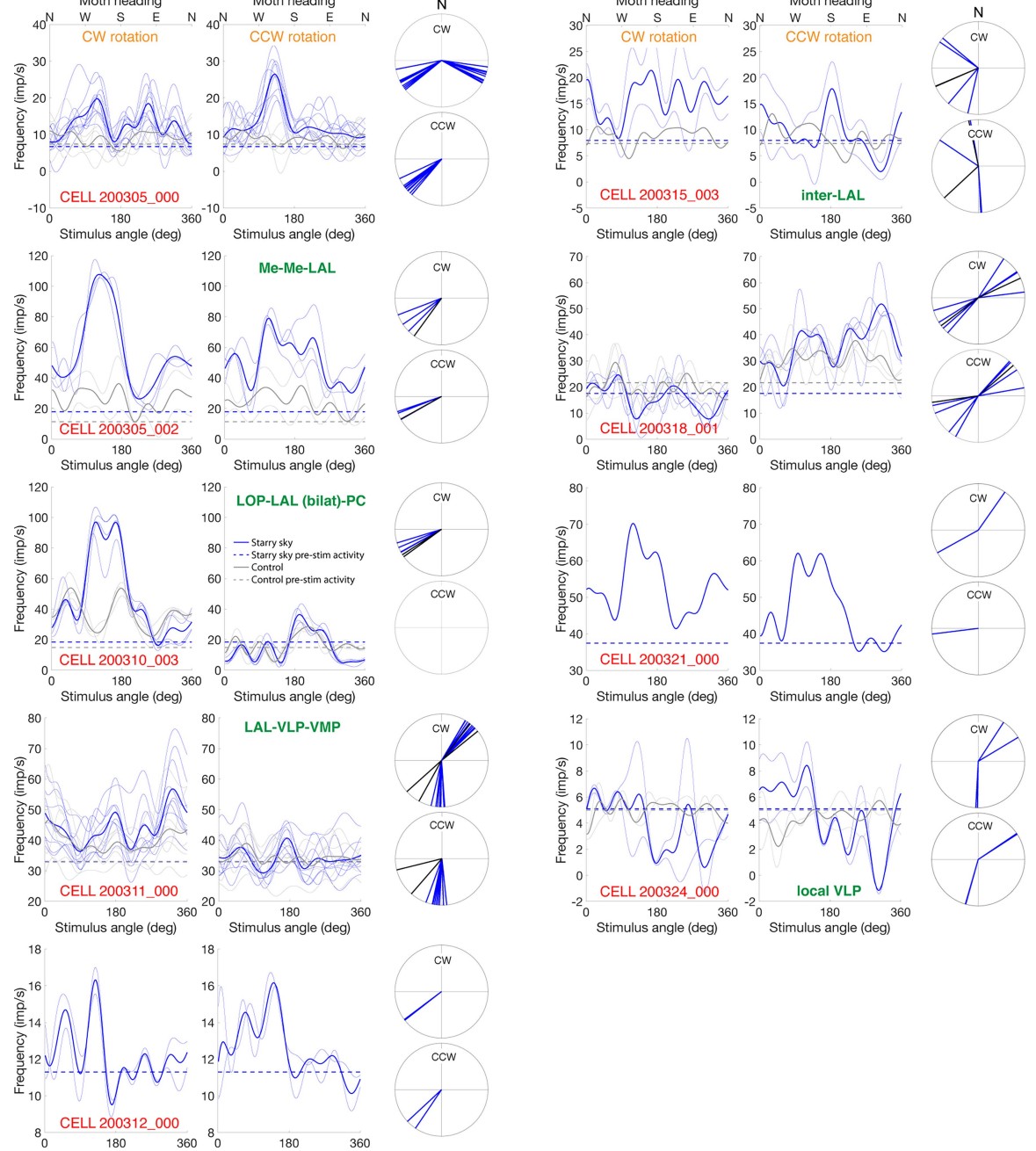

**Extended Data Fig. 6 | Visual cells recorded during the autumn migration (March 2020).** Frequency plots and circular $\varphi_{max}$ diagrams, separated into clockwise (CW, *left panels*) and counter-clockwise (CCW, *right panels*) stimulus rotations, for every cell recorded in autumn 2020 (cell ID, *red text*).

Responses to the starry sky stimulus are shown in *blue*, responses to the control stimulus are shown in *grey*. Where the cell's anatomy was identified, branching regions are also given (shown as *green text*). For detailed anatomies, see Extended Data Fig. 7.

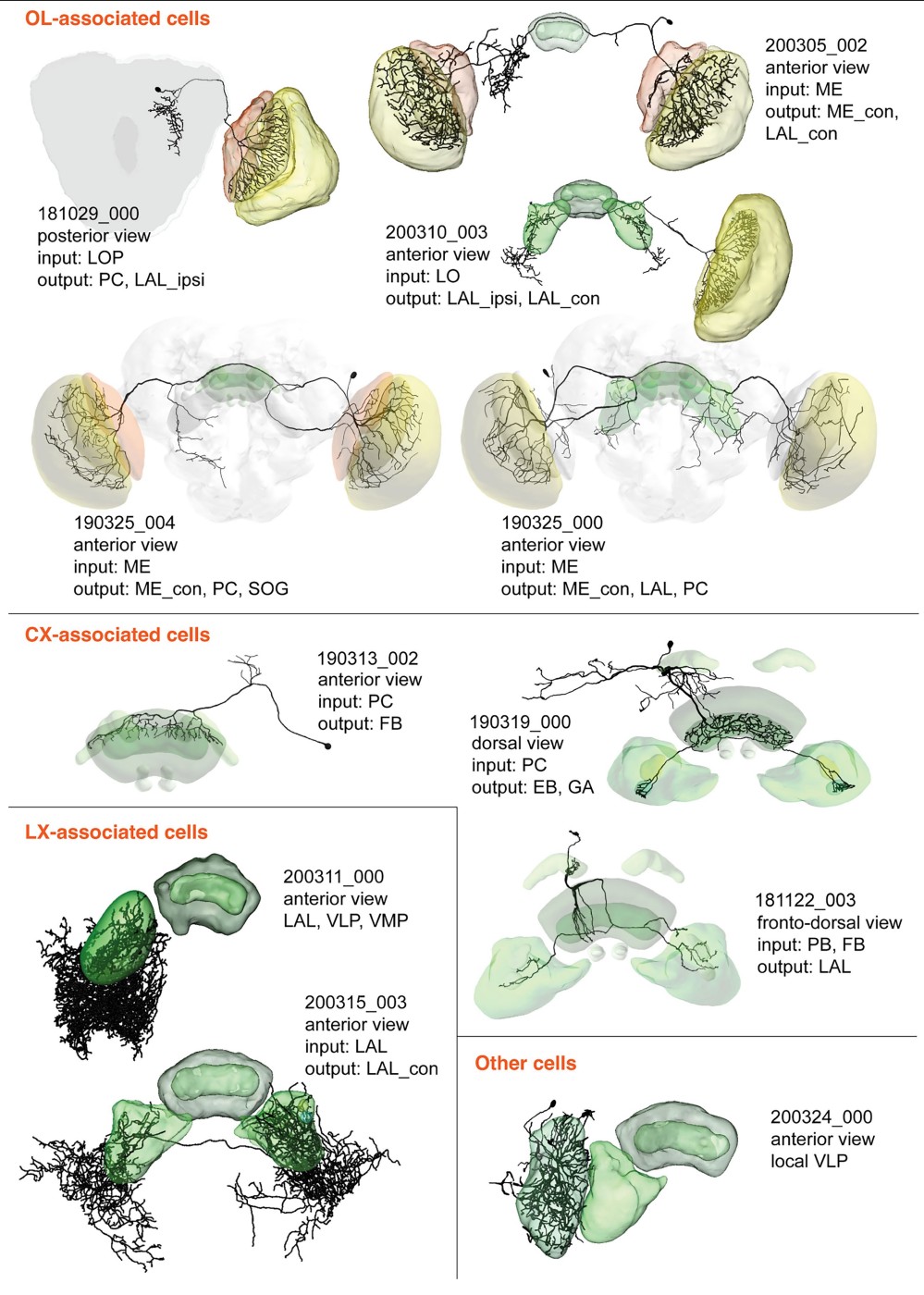

**OL-associated cells**

200305_002
anterior view
input: ME
output: ME_con,
LAL_con

181029_000
posterior view
input: LOP
output: PC, LAL_ipsi

200310_003
anterior view
input: LO
output: LAL_ipsi, LAL_con

190325_004
anterior view
input: ME
output: ME_con, PC, SOG

190325_000
anterior view
input: ME
output: ME_con, LAL, PC

**CX-associated cells**

190313_002
anterior view
input: PC
output: FB

190319_000
dorsal view
input: PC
output: EB, GA

**LX-associated cells**

200311_000
anterior view
LAL, VLP, VMP

200315_003
anterior view
input: LAL
output: LAL_con

181122_003
fronto-dorsal view
input: PB, FB
output: LAL

**Other cells**

200324_000
anterior view
local VLP

**Extended Data Fig. 7 | Morphologies and branching regions of identified visual cells.** Abbreviations: OL = optic lobe, CX = central complex, LX = lateral complex, ME = medulla, LO = lobula, LOP = lobula plate, PC = protocerebrum, LAL = lateral accessory lobes, FB = fan-shaped body, EB = ellipsoid body, PB = protocerebral bridge, GA = gall, VLP = ventrolateral protocerebrum, VMP = ventromedial protocerebrum. All cells will be published on insectbraindb.org, preliminary links to a subset of these cells are given below. Cell 181022_003: https://hdl.handle.net/20.500.12158/NIN-0000291.2 Cell 190325_000: https://hdl.handle.net/20.500.12158/NIN-0000320.1 Cell 190325_004: https://hdl.handle.net/20.500.12158/NIN-0000318.2 Cell 200313_002: https://hdl.handle.net/20.500.12158/NIN-0000293.2 Cell 200319_000: https://hdl.handle.net/20.500.12158/NIN-0000292.2.

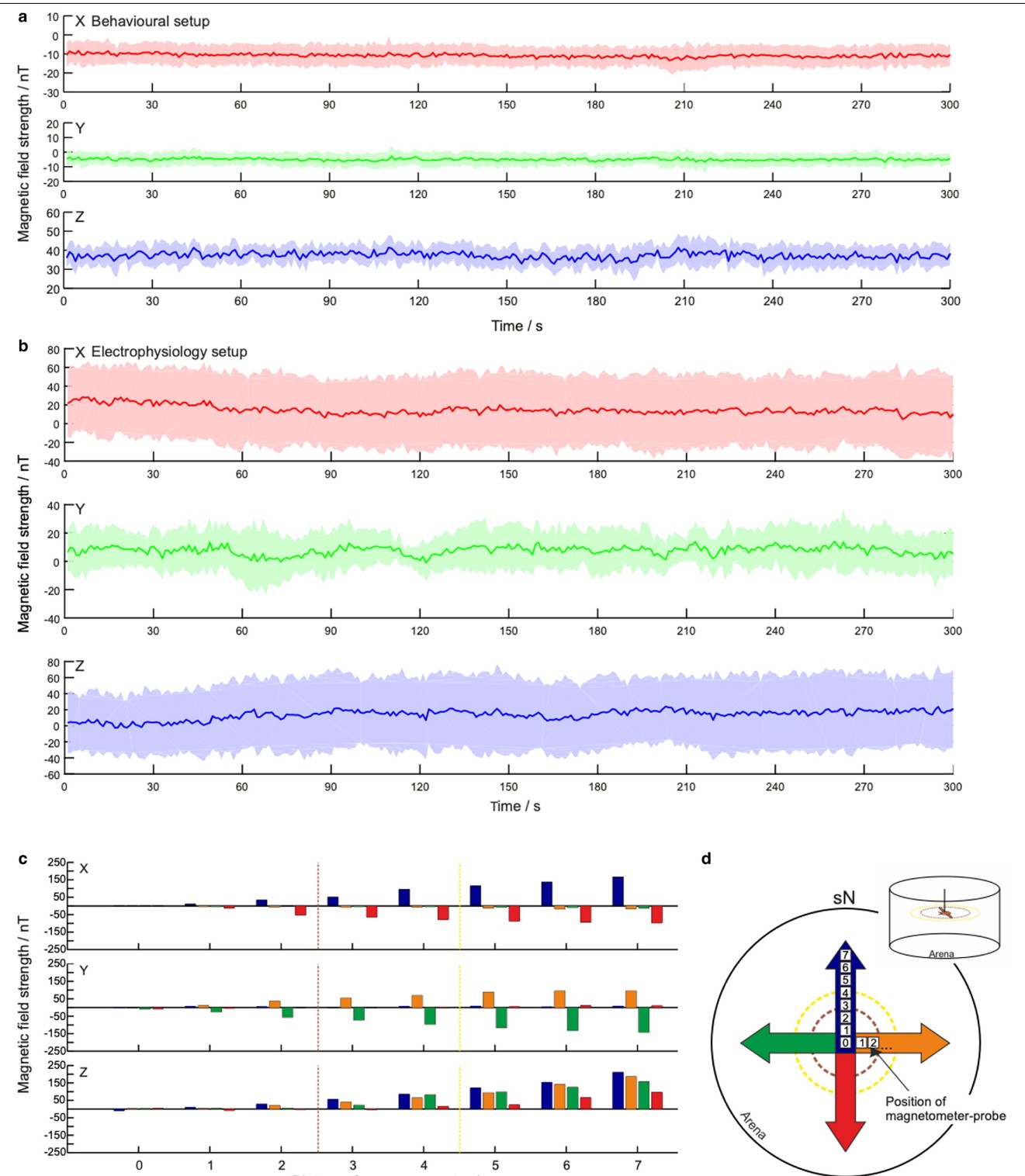

**Extended Data Fig. 8 | The magnetic conditions used in the electrophysiological (b) and behavioural rigs (a,c,d). a,b,** The Helmholtz coil system was used to compensate the local geomagnetic field and create a "nulled" or near-zero magnetic field (NZMF) within the two experimental rigs. The typical variation of the magnetic field strength, shown as nanotesla (nT) over time (s), for the duration of a typical behavioural experiment (300 s). This measurement was made at the position of the moth (position 0 in **d**). The three-component vector (X, Y, Z: *red, green* and *purple lines* respectively) of the field was measured 6 times over 300 s to obtain X, Y and Z traces. These six traces were averaged to give the mean magnetic field strength for each component (X, Y and Z) and the standard deviation (*shaded areas* around the mean). **c,d,** To control that the central area of the arena (where tethered moths flew) had a nulled magnetic field, the magnetometer probe was positioned at the arena centre (position 0 in **c**) and moved outwards in 1 cm steps along 4 directions (*coloured arrows* in **d**) relative to geographic North (gN) and the strengths of the X, Y, and Z field components were measured at each position. These field strengths were less than 100 nT a radius of 4 cm from the coil centre (*dashed yellow line* and *circle* in **c** and **d**) in all four directions (bar colours in **c** correspond to direction colours in **d**). Since the wingspan radius of a Bogong moth is about 2.5 cm (*dashed brown line* and *circle* in **c** and **d**), this indicates that the entire moth experienced a magnetic field strength substantially less than 100 nT, effectively a "nulled field" (which can be compared to the external local magnetic field strength of around 58,500 nT).

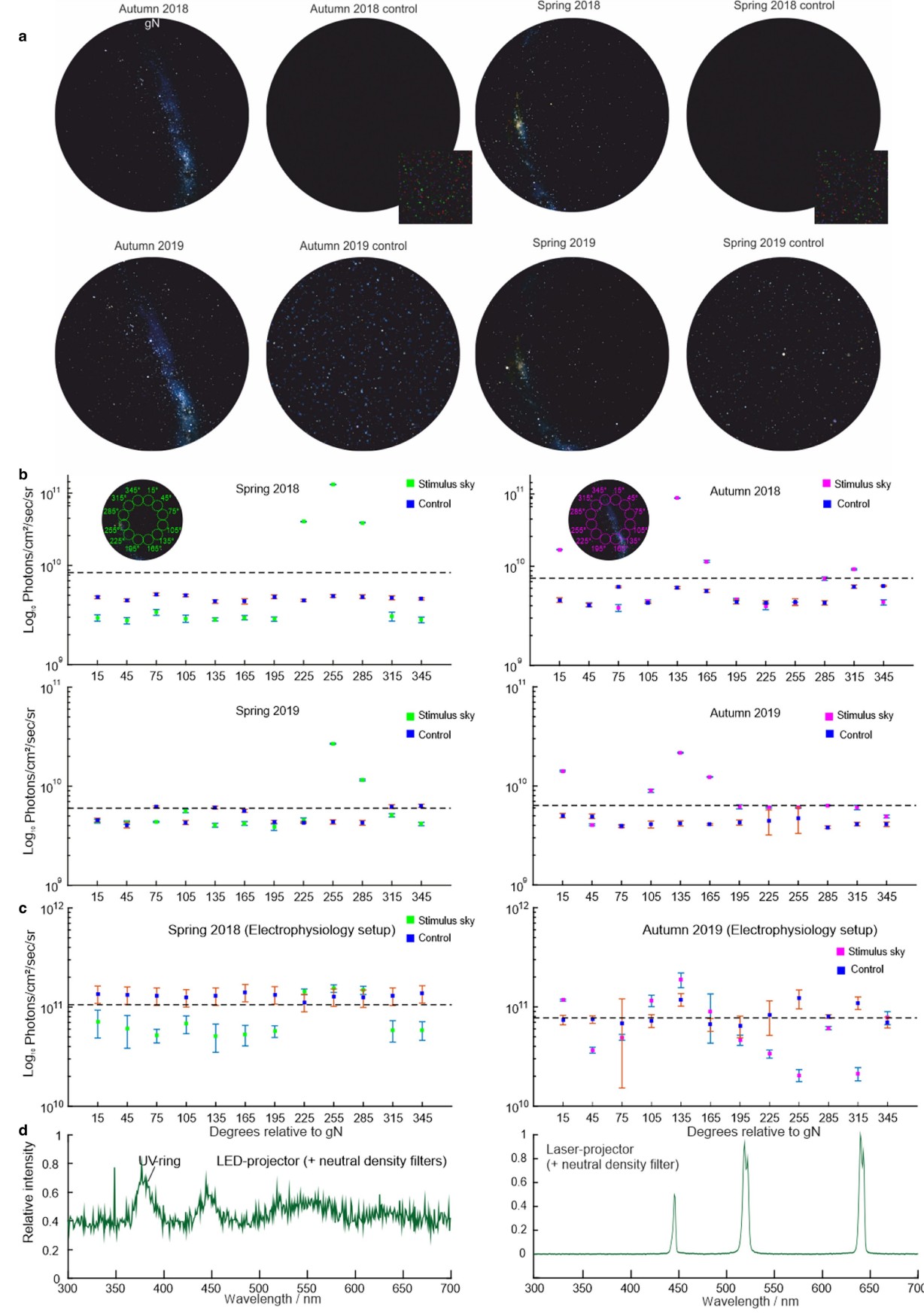

**Extended Data Fig. 9** | See next page for caption.

**Extended Data Fig. 9 | The visual stimulus conditions used in the electrophysiological and behavioural rigs. a**, Stellarium images of the austral starry night sky (and randomized sky) used during the spring and autumn migratory seasons in 2018 and 2019. Geographic North (gN) is upwards in each image. Randomized skies in 2018 were created by randomizing pixels (*insets*), whereas those in 2019 were improved by randomizing stars (see Methods). **b,c**, Brightness distributions of natural starry skies and randomized skies projected onto a screen above the moth in the behavioural (**b**) and electrophysiological rigs (**c**). The probe of a spectrometer (Ocean Optics QE65 PRO, see Methods), equipped with a collimating lens (25° field of view), was positioned at the same location as the moth and systematically pointed at the projected sky in different directions (0–360° in 30° steps) relative to geographic North. The radiance (photons/cm²/sec/sr) was measured from the screen in each direction in (5 measurements per position): mean radiance is shown as a function of measurement direction (*green squares* = spring projection, *magenta squares* = autumn projection, *blue squares* = randomized projection). Error bars give standard deviations. Note that the brightest areas of the autumn and spring projections were respectively located south-easterly and westerly. **d**. The spectra of the projectors used in the behavioural (*left*) and electrophysiological rigs (*right*), measured using the Ocean Optics QE65 PRO spectrometer. For the projector in the behavioural rig, a ring of UV LEDs was used to create a quasi-realistic spectrum (see Methods). Note that for this spectrum the noise level is relatively high since we used all the neutral density filters normally used during an experiment.

**Extended Data Table 1 | Comparison[1] of natural night skies with stimulus night skies**

|  | North | East | South | West |
|---|---|---|---|---|
| Natural Starry Sky | 19.68 | 21.10 | 20.21 | 19.82 |
| Stimulus Starry Sky | 19.75 | 19.57 | 19.62 | 19.78 |
| Stimulus Randomised Sky | 19.04 | 19.36 | 19.21 | 19.17 |

1. To compare the background brightness of the natural starry night sky with those of the starry sky stimuli presented in the behavioural rig (natural and randomized), we used a Sky Quality meter (SQM; Unihedron, Sky Quality meter with lens, Version L). All measurements were made in autumn 2019. The natural sky measurement was done at 21:30 on a hill behind the laboratory at *Glenhare* on a clear moonless night. Measurements were made by pointing the SQM (at 60° to vertical) towards the starry sky in the four cardinal directions. The same procedure was repeated within the behavioural arena, by pointing the SQM towards the projection of the starry sky on the screen. Higher values (magnitudes/arcsec$^2$) indicate lower background brightness and higher stellar contrast (i.e. a greater visibility of stars and lower light pollution). On the Bortle scale (en.wikipedia.org/wiki/Bortle_scale), the highest value on the scale – 22 – indicates the brightest stars seen against the darkest possible skies (e.g. a desert sky far from human civilisation). Values between 19.1–21.3 correspond to the brightness of the stars in suburban to rural/suburban transition areas (i.e. skies still very dark and stars highly visible, but with some background influence from human lighting).

# Reporting Summary

## Statistics

For all statistical analyses, confirm that the following items are present in the figure legend, table legend, main text, or Methods section.

| n/a | Confirmed | |
|---|---|---|
| ☐ | ☒ | The exact sample size (*n*) for each experimental group/condition, given as a discrete number and unit of measurement |
| ☐ | ☒ | A statement on whether measurements were taken from distinct samples or whether the same sample was measured repeatedly |
| ☐ | ☒ | The statistical test(s) used AND whether they are one- or two-sided *Only common tests should be described solely by name; describe more complex techniques in the Methods section.* |
| ☒ | ☐ | A description of all covariates tested |
| ☐ | ☒ | A description of any assumptions or corrections, such as tests of normality and adjustment for multiple comparisons |
| ☐ | ☒ | A full description of the statistical parameters including central tendency (e.g. means) or other basic estimates (e.g. regression coefficient) AND variation (e.g. standard deviation) or associated estimates of uncertainty (e.g. confidence intervals) |
| ☐ | ☒ | For null hypothesis testing, the test statistic (e.g. *F*, *t*, *r*) with confidence intervals, effect sizes, degrees of freedom and *P* value noted *Give P values as exact values whenever suitable.* |
| ☒ | ☐ | For Bayesian analysis, information on the choice of priors and Markov chain Monte Carlo settings |
| ☒ | ☐ | For hierarchical and complex designs, identification of the appropriate level for tests and full reporting of outcomes |
| ☒ | ☐ | Estimates of effect sizes (e.g. Cohen's *d*, Pearson's *r*), indicating how they were calculated |

*Our web collection on statistics for biologists contains articles on many of the points above.*

## Software and code

Policy information about availability of computer code

| Data collection | Custom-written MATLAB code (versions 2019a and 2022b) to control magnetic & stellar simulation, USB1 Explorer encoder software (version 1.07, US Digital) to track moths, Spike2 (version 8.03, Cambridge Electronic Design) to collect electrophysiology data. |
|---|---|
| Data analysis | Custom-written MATLAB code (versions 2019a and 2022b) to analyse moth trajectories and electrophysiology data, MS Excel (MSO 2019), R (version 3.6.1) and Oriana (version 4, 2011) to perform the statistics. Code available: github.com/stanleyheinze/Starry_sky_code |

For manuscripts utilizing custom algorithms or software that are central to the research but not yet described in published literature, software must be made available to editors and reviewers. We strongly encourage code deposition in a community repository (e.g. GitHub). See the Nature Portfolio guidelines for submitting code & software for further information.

## Data

Policy information about availability of data

All manuscripts must include a data availability statement. This statement should provide the following information, where applicable:

- Accession codes, unique identifiers, or web links for publicly available datasets
- A description of any restrictions on data availability
- For clinical datasets or third party data, please ensure that the statement adheres to our policy

The experimental data that support the findings of Figures 2, 3, 4 and 5 of this study are available in Figshare with the identifier https://doi.org/10.6084/m9.figshare.25780197.v2. The experimental data that support the findings of Figures 4 and 5 can also be found in the Extended Data.

# Research involving human participants, their data, or biological material

Policy information about studies with [human participants or human data](). See also policy information about [sex, gender (identity/presentation), and sexual orientation]() and [race, ethnicity and racism]().

| | |
|---|---|
| Reporting on sex and gender | Not applicable |
| Reporting on race, ethnicity, or other socially relevant groupings | Not applicable |
| Population characteristics | Not applicable |
| Recruitment | Not applicable |
| Ethics oversight | Not applicable |

Note that full information on the approval of the study protocol must also be provided in the manuscript.

# Field-specific reporting

Please select the one below that is the best fit for your research. If you are not sure, read the appropriate sections before making your selection.

☒ Life sciences    ☐ Behavioural & social sciences    ☐ Ecological, evolutionary & environmental sciences

For a reference copy of the document with all sections, see [nature.com/documents/nr-reporting-summary-flat.pdf](http://nature.com/documents/nr-reporting-summary-flat.pdf)

# Life sciences study design

All studies must disclose on these points even when the disclosure is negative.

| | |
|---|---|
| Sample size | As we were dealing with wild-caught migratory insects that were caught over periods of several weeks and were difficult to catch in large numbers, we simply used as many individuals as we could. In our behavioural work, we routinely obtained significant results even after a relatively low number of individuals were tested (around 15-20). However, since we wished to make sure we could replicate our results over at least four migratory seasons, our sample sizes were typically around 50-70 individuals. |
| Data exclusions | From our Methods section: "Moths chosen for analysis were required to fulfil three ante hoc criteria, two prior to the experiment and one during the experiment: (1) the tethering stalk was perfectly vertical, (2) wing flapping was vigorous and its amplitude was large and equal for both wings (indicating that the contact cement had not interfered with the wings), and (3) that the moth flew continuously for the full 5 min. For the last criterion, if a moth stopped flying, the arena was gently tapped in order to stimulate the moth to continue flight behaviour. A moth that stopped flying 4 times was rejected and the recording aborted." |
| Replication | We replicated our behavioural results over 2 spring migratory seasons and 2 autumn migratory seasons. The replication was so good that we pooled the data from the two spring migratory seasons into a single data set. The same was true for the two autumn migratory seasons. With regards to the electrophysiological data, since each recording was a unique observation, replication was not possible. This is due to the stochastic nature of intracellular recordings - single neurons were penetrated randomly from target brain regions that contain many thousands of cells. |
| Randomization | Allocation of moths to experiments was random. Each afternoon approximately 20-30 moths were collected randomly from our store of moths and a tether was glued to the back of each of them for behavioural experiments. These moths were then used for experiments on the same night. Afterwards their stalks were removed and they were released into the wild. In parallel, on the same afternoons, 3-5 moths were similarly collected (i.e. randomly) and prepared for electrophysiological experiments. Moths used for these experiments were sacrificed at the conclusion of the experiment by swift removal of the head using a razor blade. |
| Blinding | Our study was not blinded as a single individual carried out the experiment and it was impossible to hide the experimental condition applied. |

# Reporting for specific materials, systems and methods

We require information from authors about some types of materials, experimental systems and methods used in many studies. Here, indicate whether each material, system or method listed is relevant to your study. If you are not sure if a list item applies to your research, read the appropriate section before selecting a response.

## Materials & experimental systems

| n/a | Involved in the study |
|-----|----------------------|
| ☒ | Antibodies |
| ☒ | Eukaryotic cell lines |
| ☒ | Palaeontology and archaeology |
| ☐ | ☒ Animals and other organisms |
| ☒ | Clinical data |
| ☒ | Dual use research of concern |
| ☒ | Plants |

## Methods

| n/a | Involved in the study |
|-----|----------------------|
| ☒ | ChIP-seq |
| ☒ | Flow cytometry |
| ☒ | MRI-based neuroimaging |

## Animals and other research organisms

Policy information about studies involving animals; ARRIVE guidelines recommended for reporting animal research, and Sex and Gender in Research

| | |
|---|---|
| Laboratory animals | The study did not involve laboratory animals. |
| Wild animals | The Australian Bogong moth Agrotis infusa, wild caught during their migration to the Australian Alps in spring (age: ca. 5-6 months old) and from the Australian Alps in autumn (age: ca. 8-9 months old). Migrating moths were caught using light traps between 9 pm and 11 pm in alpine areas. They were placed in individual plastic containers (one moth per container) and transported back to the lab (70 km away) in a car. Following the single night of behavioural experiments that each moth was subjected to, moths were released back to the wild (following removal of the tethering stalk). This release was made near the lab at a location that is close to the natural migratory route. Moths used for electrophysiological experiments were sacrificed at the conclusion of the experiment by swift removal of the head using a razor blade. |
| Reporting on sex | We always noted the sex of each moth we tested, however as there was no statistical difference in the results obtained from males and females the data was pooled. |
| Field-collected samples | Our study involved lab experiments on wild-caught moths (see above). Prior to experiments moths were housed in small cylindrical plastic containers with screw-on lids  (ca. 5 cm diameter and 7 cm high) and kept in cool dim conditions with a natural but maximally dim light cycle. Moths were not held like this for more than 3 days and were fed daily with an earbud soaked in honey solution. |
| Ethics oversight | No ethical approval is required for working on insects. However, the work was performed under a Scientific License issued by the Australian Government. |

Note that full information on the approval of the study protocol must also be provided in the manuscript.

## Plants

| | |
|---|---|
| Seed stocks | Not applicable |
| Novel plant genotypes | Not applicable |
| Authentication | Not applicable |

