## [Peer Review File · Nature]

Bogong moths use a stellar compass for long-distance navigation at night

Corresponding Author: Professor Eric Warrant

Version 0:

Reviewer comments:

Referee #1

(Remarks to the Author)

In this manuscript, Dreyer, Adden and colleagues explore the navigational mechanisms used by bogong moths to migrate seasonally. Specifically, they test whether bogong moths rely on celestial cues at night, in coordination with magnetic field cues, for long-distance navigation. They begin by demonstrating through convincing behavioral experiments that, in absence of Earth's magnetic field, moths can rely on the austral starry night sky (notably the Milky Way) to orient in the appropriate migratory direction, and that reversing these celestial cues by 180° reverse orientation by the same amount, supporting a role for stars as a compass. Importantly, they show that moths become disoriented in absence of both magnetic and stellar cues. This evidence is consistent with work in dung beetles and birds which demonstrates use of the Milky way or star patterns respectively, as directional cues. The authors further report the existence of neurons in the moth brain that respond to 360° rotations of the austral night sky by increasing firing when the moth is flying south, but not when star patterns are randomized. They show that some of these neurons are located in the optic lobes, central complex and lateral accessory lobes (steering center). In a second set of experiments, the authors show that the same neurons respond to key aspects of the night sky that correlate with the straight pattern of the Milky Way and one of its bright spot. While these regions of the brain have previously been shown to be involved in compass navigation, the authors identified several new cell types that are likely important for a stellar compass.

In terms of the care with which the experiments appear to have been conducted, the quality of the data and the clarity of the manuscript, which was a pleasure to read, this is a very solid study that yields interesting results. However, conclusions are not always drawn with caution and the novelty is somewhat diminished by the fact that many similar conclusions have been previously drawn in other animals. Indeed, the use of a stellar compass for long-distance navigation has been shown in birds and the use of the Milky Way for navigational purposes was previously demonstrated in dung beetles, an ability referred to as a compass. Likewise, the neural basis of insect compass orientation and steering has been well described in insects, including monarchs, dung beetles and bees, and occurs in the same brain regions to those described here. The discovery of neurons that respond to 360° rotations of the austral night sky/Milky Way thus seems to be the major conceptual novelty. Therefore, while this study represents the first demonstration of the use of a stellar compass in a migrating insect, paired with neural recordings, authors should be cautious of the claim that it is the first demonstration of a stellar compass in a long-distance migrant or in an insect. While dung beetles do not migrate long distances, their use of the Milky Way for orientation is described as a compass system and has been well studied. Authors describe some of the differences in dung beetle and bogong moth stellar navigation, but nonetheless, there are still many similarities between the two and these should be acknowledged.

I would add the following further comments:

1. Previously, Dreyer et al, 2018 demonstrated that moths require a visual cue in coordination with Earth's magnetic field to orient properly. However, when experiments were conducted in Figure 2 under cloudy sky, and no other celestial visual cues were provided, moths remained significantly oriented. Authors concluded it was because the moths were using Earth's magnetic field. However, because the arenas were transparent, the formal possibility that moths could have been relying on a terrestrial visual cue has not been excluded. Given that moths oriented in the same direction in all three experimental treatments, ruling out potential bias in the set-up would have been important to reach the author's conclusion. In absence of this control, the statement in line 100-102 that "moths rely on the geomagnetic field as their main compass for navigation" is an overstatement as moths have not been shown to rely solely on magnetic cues in the absence of all other stimuli. I

acknowledge that the focus of the study is on the stellar compass and not the magnetic compass, but a more robust demonstration that Earth's magnetic field is used in the absence of any other cues would have been to reverse the horizontal component of the field, or invert the vertical component to determine if moths reverse their orientation under cloudy sky. While I do not think repetition of the natural setting or laboratory experiments would be a reasonable request given the seasonal constraints associated with performing such work, the authors should try to reconcile the results from these experiments with previous work, which suggested moths rely on both terrestrial and magnetic cues. Additionally, authors should be cautious with the conclusions drawn about the magnetic compass of moths throughout the manuscript.

2. It is surprising that the authors did not address or discuss the possibility that polarized light could be used as an additional navigational tool in bogong moths. Polarized light cues are present even at night, and contribute to the dung beetle compass system (in addition to numerous other insects). Would polarized light be a potential navigational cue for moths? As noted above, the use of the Milky Way for short distance navigation has been well-described in dung beetles, but this work is only minimally addressed by the authors. Addressing some key similarities and differences between the mechanisms underlying dung beetle use of the stars, and bogong moth use of the stars in light of the different navigational types of these insects would strengthen the current manuscript and its importance to the field.

3. Results indicate that a stellar compass and a geomagnetic compass (pending that the result under cloudy sky are not due to other visual cues) may each be sufficient to guide Bogong moths in their migratory direction. However, in natural conditions, both are likely to operate together. Can the authors speculate/discuss on the possibility of a hierarchy between cues when they co-occur?

4. The idea that the central complex may be the site of integration for starry sky cues (lines 228-230) is supported only by physiological data from a single neuron. I think this needs to be acknowledged.

5. In the method sections, many custom-written codes are mentioned. However, these codes are only available upon reasonable request. I see no justifiable reasons for not sharing them publicly.

Minor comments:

1. Throughout the figures, the purple color being used is barely distinguishable from the blue one. More contrasted colors would help the readers.
2. Figure 4: Panels d and e should be switched in the figure as e is described first in the text. What kind of plots are being shown in panel e? These are not typical plots and an explanation of what they are, which is currently missing in the legend, is needed.
3. Extended data Figure 1: in panel d, the flying moth is not circled. Either circle it or remove "(circled)" from the legend.

(Remarks on code availability)

The codes were not available.

Referee #2

(Remarks to the Author)

This study investigates migratory bogong moths and whether the moths can use star patterns to hold a consistent migratory heading. In arenas outside, where the moths had access to natural cues, tethered moths flew in their migratory direction under both clear and cloudy skies. In the lab, moths exposed to a projected view of star patterns in a zero magnetic field (to prevent use of their magnetic compass) moved in the migratory direction. When the star pattern was reversed the moths flew the other way. Electrophysiological recordings were made from 28 neurons in the optic lobe while star patterns were rotated. The neurons responded to these rotations and also to rotations of bars and circles of light simulating some aspects of the Milky Way. A subset of neurons was traced to the central complex and to the lateral accessory lobe, two areas of insect brain involved with orientation and steering. It is concluded that the moths have a star compass that is used in their migration and that neurons in the visual system respond to star patterns or at least some element of it, most likely the brightness of the Milky Way.

This is an interesting, high-quality study. The orientation mechanism is not entirely novel inasmuch as use of the Milky Way as a compass for orientation has been shown before in dung beetles (e.g., Dacke et al., 2013, *Curr. Biol.*) and some limited evidence for use of stars by migratory moths has been reported (e.g., Sothibandhu, 1979). At the same time, the study elegantly extends the dung beetle findings to a long-distance migrant for the first time. Also, although there has been much work on the neural basis of celestial orientation during daytime (butterflies, desert ants, locusts, etc.) this is, to my knowledge, the first neural recordings in insects to suggest responses to star patterns.

Although there is much to admire in the study I was confused by several issues and unable to fully follow several aspects of it. Some conclusions might need to be modified and some additional information is needed.

1. The data in Fig. 3 make a strong case that moths can use some aspect of the night sky for orientation, but it does not seem clear yet whether the moths have a true compass or a different kind of response. Having a compass means that an animal can hold headings in any direction relative to a cue, just as a hiker with a standard compass can travel N, NE, SW, etc. The moths just travel north and south, so it doesn't seem clear that the moths can hold a path at any angle to the Milky Way, as

opposed to just flying toward it or away from it. Line 240 describes the Milky Way as a bright extended stripe of light brightest in the southern half of the sky. If moths just fly in the direction of the brightest area of sky when going south and away from it when going north, this would not require a compass, just a seasonal reversal of preference for traveling toward/away from bright sky. Are there additional data that might support or rule out this interpretation? Similarly, an earlier study by Dreyer (Curr. Biol.) showed the moths changed direction when the magnetic field and a visual cue were moved together. Could the Milky Way be the natural visual cue that moths use? If so, moths might use the Milky Way as a sky landmark rather than as a stellar compass.

2. I was confused by statements that the moths use the starry night sky to distinguish N,E,S,W cardinal directions (e.g., lines 32, 126, 238). Maybe the moths can do this but there seems no evidence presented here that they can do this or need to do this. They only fly north or south and might do that just based on brightness of the southern sky without reference to cardinal directions. More broadly, an animal does not need to know cardinal directions to maintain a heading, for example a fish can move straight upstream toward a food odor. I believe these statements are in error or, if they are not, need to be clarified.

3. I could not fully follow Fig. 4b. There are data points with two different shades of gray but the darker shade does not seem to be defined. Also, are unimodal inhibited cells included on this graph, so that peaks of inhibition of some neurons are plotted in the same area as peaks of excitation for others? I would suggest plotting in separate panels in Fig. 4, or in supplemental, the unimodal for spring, the unimodal for fall, and the direction-sensitive unimodal and bimodal. On the unimodal data it would help to indicate which are inhibition peaks and which are excitatory peaks. At present it is difficult to tell from this figure what is happening. Also, although it is true that all unimodal when grouped together show maximal responses with the average in the south, it also appears that the spring (light blue) and autumn (dark blue) might be different, with the spring bimodal (peaks in SE/NW) and the autumn unimodal SW. Is a seasonal difference possible, perhaps reflecting changes from moving south to moving north?

4. In the context of moths responding to the brightest area of sky illuminated by the Milky Way, it would be worthwhile to discuss the role of the moon, as this illumination in Fig. 2A, 2B seems opposite that of the Milky Way. There was no moon in the indoor experiments, which is an important way that the tests deviate from a natural setting. It would be helpful to discuss this. Would the use of the stars be diminished during full moon? Are the moths able to filter out the moon somehow?

5. I did not understand the distinction drawn between the dung beetle compass and the moth compass (lines 248-252). Many others have referred to the dung beetle's use of the Milky Way as a compass (e.g., reviews by Dacke, el Jundi, etc.) so I was surprised by the statement that dung beetles do not have a star compass (line 248). Similarly, on line 57, I don't think it is correct to say that 'no invertebrate is known to use the stars to discern compass direction,' as dung beetles do exactly that (Dacke et al., 2013). Really there seems better evidence for a true compass in the beetles because they use the Milky Way to move in any direction, not just north or south. The beetle and moth systems seem likely identical, just used over different distances (short for beetles, long for moths). It would improve clarity to state this if it is true.

6. There is room for improvement in discussing other possible cues besides stars and magnetic fields. Most animals use multiple cues so there seems little reason to assume that star cues and magnetic cues are the only ones for moths. Dung beetles use polarised moonlight, so why not moths? The moon itself might be a cue (Sotthibandhu, 1979), maybe odors when they get near the end of the migration, and so on.

Minor comments:

Image in 3A seems to have Milky Way more in west than in south. Is that the true situation? If so, please reconcile with statement about the Milky Way constantly remaining in the southern part of the sky (lines 244-246).

It would be helpful to clarify that the bird star compass and the moth star compass (if it is a compass) operate on different principles, with birds in the northern hemisphere using the non-rotating part of the sky as a reference.

For the electrophysiology, it would be helpful to provide more information about how the four types of neuronal responses were identified and recognised. Not very much information is given. With some of the directional bimodal data, it does not always seem obvious that there are two peaks instead of one.

Please specify the rate at which the projected sky pattern was rotated. That was listed for the dot and bar but not the star pattern unless I missed it.

Line 90: Are the moths really unaffected by the nightly movement of the stars and moon or do they have a time-compensated star or moon compass? Isn't either possible?

Line 148: Does 28 neurons refer to the total number impaled or the number that responded? Were there neurons that did not respond? Please clarify.

(Remarks to the Author)

This is a very interesting and clearly presented paper describing the use of stellar cues in the night sky for navigation of the Bogong moth. It is the only case that I am aware of, in insect species where night sky cues are used in this way. It, thus, should be of interest to the wide audience that reads *Nature*. The work relies on excellent behavioral data and is backed up with intracellular recordings in 3 brain regions that are known to monitor and process visual cues used in navigation in a wide range of insect species. The paper clearly shows that either the geomagnetic or visual cues in the starry night are used by this migratory species and raises numerous interesting questions for future research such as how the visual and geomagnetic compasses interact and how navigational information is stored for the return trip and passed on to the next generation for their first migratory trip.

I must say, I was a bit confused when I first read about fig 2. My conclusion was that the moths were simply not using visual cues. Then I read the final paragraph of the section and found that the authors were also coming to that conclusion and then in the next section tested this with a very clever experiment. That's fine. I came away convinced after discussion of figure 3 that the stellar compass is important by itself. However, it left me with questions about the relative importance of the two compasses under normal conditions and how they might interact. I will get to that below.

The behavioral data certainly make a convincing argument that the moths can use either the stellar compass or the geomagnetic compass when only it is available and the neural data clearly show that visual information is available in the various brain regions. The neural data are not surprising given the wealth of visual data taken from these brain regions in various insect species. Nevertheless, they are important to the story. However, it seems to me that the more crucial question for this species is how the two compasses normally interact. These authors have previously presented data showing that the Earth's magnetic field can be used by these moths to direct their flight and here they focus on the cues found in the night sky. Normally both compasses are available on a clear night. I understand why you would want to have the geomagnetic compass when the sky is obscured, but why do you need the stellar compass at all? The geomagnetic compass is always there under natural conditions when it is not being blocked in the simulator. Is there an advantage to adding the stellar compass over using the geomagnetic compass alone? Is the stellar compass more accurate on its own than the geomagnetic compass or do the two working together provide a better navigational solution?

I wonder if the setup the authors used could be modified to examine how the moths are using both compasses simultaneously. What if you placed the moths in the simulator with the magnetic field left alone and intact and then manipulated the sky image? What if the sky were rotated not so much as in figure 3 but more than the few days used in figure 1. Perhaps, you could use intervals matching say 1 or 2 month time segments. Would the moths still orient according to the geomagnetic seasonal cues or would they change their orientation according to the altered visual time sequence preferring that compass if it is available? If the sky is a major cue, one would expect the direction to rotate with the sky rotation (even though the moths may never experience this kind of rotation in their normal life span). If not, then one would conclude that the stellar compass, while present, may be irrelevant. Another way of testing this would be to rotate the sky image in the same increments, but in random directions. Would the moths ignore this confusion and rely solely on the geomagnetic compass or orient according to the situation at any particular month's sky? Or another alternative is that they would orient to a direction between the two compasses suggesting that both are normally used in tandem. Yet another possibility is that one of the compasses is normally used to calibrate the other. I am reminded here of how the visual system of owls has been shown to calibrate the auditory map as shown by Brainard and Knudsen's classical studies using prisms to shift the owl's visual field. Perhaps the geomagnetic compass is refined by the visual cues to accomplish something like the directional switch that occurs between seasons.

I understand that the kinds of experiments that I suggest above may be beyond the scope of the present study. Nevertheless, I think it would be useful to have the authors mention the question of interactions between the two compasses and how they think they may interact, especially, since the last section of the paper points out the existence of these two compass mechanisms.

I found the neural data presentation in figure 4 a bit confusing. The text seemed to bounce around among the various sets of data. First the data in 4a are briefly mentioned and then the text switches to the graphs in 4e before an in-depth discussion of 4c and then going back to the details in 4a then finally mentioning 4d. I would suggest discussing the types of cells described in 4a first and then moving to the data in 4b and the more detailed discussion of 4c. I also found the discussion of 4c difficult to follow. I think this needs some clarification about what the dots and bars represent as well as the two rows of graphs.

Referee #4

(Remarks to the Author)

In their manuscript entitled "Bogong moths use a stellar compass for long distance navigation at night", Dreyer et al provide an exciting study demonstrating how Australian bogong moths use a combination of both the Earth's magnetic field and visual cues (i.e. the starry night) as orienting cues during their long-range migration. These findings appear to show striking similarities for what has been demonstrated for navigating birds, thereby demonstrating that these rather sophisticated navigational skills also exist in the invertebrate world.

The manuscript combines behavior experiments, electrophysiology, and neuroanatomy, using a non-model organism which is not exactly easy to work with. Overall, the data are very convincing and demonstrate the amazing abilities of invertebrate

navigators, adding a beautiful nocturnal facet for comparison with the Monarch butterflies. I have a couple of major points that the authors should address, since I believe this is an exciting manuscript for a general audience, and addressing these points should clarify the messages for the reader considerably. If these points were to be addressed, then I think this manuscript would be a great addition to Nature.

Major points:

- Orientation of the milky way: It is not clear to me how the orientation of the stripe formed by the milky way changes over time (days, weeks, years). This needs to be clarified for the general audience. Figure 3 insinuates a general North-to-South orientation, yet Figure 2 shows an almost West-to-East orientation. Related to the behavior experiments in Figure 3: Have the authors ever projected skies rotated 90 degrees, i.e. with a milky way oriented in West-to-East direction? Two possible outcomes are conceivable: a) a 90 degree switch in heading, or b) undirected behavior (since the orientation of the milky way is 'off'. Has this experiment been done? Note: I understand that the authors come to the conclusion that the milky way may in fact not be perceived as a stripe, but instead leads to a luminance (?) bias in the South. However, their experiments with a rotating stripe suggest that cells in the animal's brain indeed respond to stripes.
- The authors mention that the stellar pattern changes over time, yet they do not mention whether the cells they record from in Figure 4 manifest any interesting changes over time. How long did individual recordings last? Were they too short to show any temporal effects? (I would assume so). More interestingly, could the large spread of data points in Figure 4c also be the result of temporal effects? Have the authors plotted these dots as a function of the date when these data were recorded? I am curious whether there is a systematic shift over time.
- Alternatively, most temporal changes of the stellar pattern could be averaged out, by generating some 'Southern blob' of brightness, I assume. The final sentences of the manuscript seem to allude to such a possibility. But aren't the dot/bar stimuli then a bit ill-chosen? Have the authors ever tried to destroy the resolution of these stimuli, i.e. generated more diffuse patterns of brightness, to test whether this is what drives wither behavior and/or the activity of the recorded cells?
- As a direct follow up question to this, one wonders whether the observed behaviors are really in response to a stellar pattern or whether they could also be simpler, like phototaxis, or rather menotaxis? Could the authors elaborate on how stellar navigation is different from menotaxis in response to a celestial body? Directly related to that: Could the authors elaborate on why they think that the moon does not influence the behavior of the animals? Or does it? One would think that the moon adds to the overall brightness patterns. Can this be calculated? Is the rest much brighter than the moon? This would be exciting.
- It seems plausible that the unimodal cell types would be useful for defining the heading direction, whereas the bimodal ones could influence steering decisions/course correction, since they respond to CW vs SSW turns, if I understand well. Is such a hypothesis supported by the brain regions where these cells are found (more peripheral vs more central), or not?
- The heading of moths in Fig 5 (as indicated by the tuning of their cells) appears to be in the SW direction and not directed South, as in most other Figures. Is that an effect of the time of the year when the experiments were done? Please elaborate.
- The authors make a convincing argument that both magnetic and stellar compasses provide alternative sources of directional information. One can be prioritized when the other becomes unreliable. Can the authors please elaborate under which conditions the magnetic compass were to become unreliable? This is not intuitively obvious. Alternatively, it seems entirely plausible that the two compass systems act together, one reinforcing the other. Have the authors tried to reduce the quality of the stimulus (by blurring, or by reducing intensity) when the 2nd cue was not present? One could expect a deteriorated/less robust response in this case. The way I understand the current magnetoreception literature, it all seems to happen in the photoreceptors....hence, that's where the two systems could directly interact, no? I feel like this part needs more discussion, since the 'choice between to systems' is a bit simplistic and not supported by data.
- The authors say that their work is the 1st demonstration of an invertebrate using the stellar pattern for navigation. In the past, they have demonstrated dung beetles using the milky way for orientation. Such orientation tasks usually include many of the same visual cues as navigational tasks do, they even rely on (partly) the same circuits. Can the authors make it more clear how the navigational challenges of bogong moths are fundamentally different from the orientation tasks that a dung beetle faces?

Minor:

- Please assign the four cell types in Figure 4a the numbers that they have been given in the text
- How has the stellar pattern changed over the last 10.000's of years? I think this question is important when considering how the underlying circuits have evolved/were selected and may be modified in the future.
- From what I know bogong moths have become extremely endangered & the numbers have gone down rather dramatically. Can the authors please mention that in the text? Can give an outlook as to what kind of population development can be expected? One should seize the opportunity to raise awareness here.

Version 1:

Reviewer comments:

Referee #1

(Remarks to the Author)

We feel that the authors have done a thorough job addressing all our prior comments and concerns. Based on this new draft, we have a couple additional comments and thoughts below.

1. This point is minor, but in response to our original major comment #1, the authors state they place the results in the context of the overcasts conditions and softened the text by removing “main” in lines 116-118. This is most likely an oversight, but “main” has not been removed.

2. Lines 284-288: This new sentence is very long with several moving parts, which makes it difficult to follow. The authors should consider breaking it down in two.

3. Bogong moths were shown to continue to maintain a northward orientation irrespective of whether they were tested during the early or late evening (when the Milky Way and star patterns had significantly shifted). This suggests two possibilities: either moths rely, like birds, on the axis of rotation to determine north/south, or their use of celestial cues might be time-compensated. It might be worth expanding on this idea following the mention that birds use the rotation of the sky for their stellar compass (lines 148-150) – e.g. if bogong moths use a time-compensated stellar compass, this would suggest the interesting possibility of a conserved role for the circadian clock in time-compensation of stellar and sun compasses used by night-time and day-time migrants, respectively.

(Remarks on code availability)

I verified the code were made publicly available but do not have the expertise to evaluate them.

Referee #2

(Remarks to the Author)

The manuscript has been improved and a number of aspects clarified. The authors have persuaded me that the moths cannot be using a simple phototaxis. I agree now that it is more akin to some kind of stellar or celestial compass, although I am mystified about how this can work with the Milky Way moving across the sky. If I understand correctly, the bar of light moves and rotates (e.g. Fig. 2), the brightest area of light changes position, there is no obvious stationary part of the sky (like the north star in the northern hemisphere) but moths tested 3 h apart (Fig. 2a, 2b) did not change headings. Can the authors provide any additional thoughts on how this is possible? I know that the hierarchy/weighting of stellar vs magnetic cues cannot be determined in the experiments done outside (Fig. 2a, 2b), but if moths are using stars, is there any possible explanation besides a time-compensated stellar compass? I think it would be worthwhile to comment briefly on this in the manuscript, because time-compensated sun compasses are well known, but a time-compensated star compass would be completely new.

I continue to be enthusiastic about the study, but I remain confused about the difference between the stellar compass of dung beetles and the moths, despite the explanation provided in the rebuttal. To me the two compasses are similar, if not identical, and the attempts to distinguish between them seem premature and unneeded. All we know is that rotating the night sky causes a change in orientation direction in both insects. Functionally, the beetles use stars to travel on a bearing, with each individual selecting a different direction. The moths use stars to travel on a bearing, with all individuals programmed to migrate the same way. It is possible the compasses are different, but right now it seems too soon to know. The ecology of the two insects is different so it makes sense that what the compass is used for is different. It seems analogous to a group of hikers in a forest, each using a compass to move in a different direction (dung beetles), vs. hikers using the same compasses to move toward the same faraway goal (moths).

Something similar is known for sun compasses and magnetic compasses, both are found in animals that use them differently, e.g. by moving short or long distances, or toward a migratory goal vs variable local directions. Also the goal vs non-goal argument does not resonate with me because the dung beetle has a goal (a suitable area away from a crowd of rivals), it is just that suitable areas can be reached by traveling in many directions, whilst the suitable areas for moths all lie in directionally similar areas.

One question of terminology. Although navigation means different things to different people, few readers will agree with the definition of navigation provided in the manuscript and on pp. 3 and 8 of the rebuttal, which is that navigation is holding a bearing in ‘a specific geographical direction’. That is usually considered orientation, not navigation. Most places in the paper, including in the title, the word navigation should probably be replaced with orientation to avoid confusion, as many readers will assume navigation involves a familiar goal, a learned route/cognitive map like a rodent, true navigation (see rebuttal footnote), etc. The abstract (line 34) can avoid confusion by saying something like “moths can determine direction using the starry night sky and therefore adopt a course that reflects their inherited migratory direction”. And so on.

Line 104: “Implying that if polarised light is used as a compass cue, it is not essential.” True enough, but I am not quite understanding the point because the same is true for the magnetic compass and star compass. Neither is essential if other cues are present. The sentence after also does not seem relevant, as it probably just means the moths have a magnetic compass, but this seems unrelated to using polarised light. Revise slightly for better clarity?

Line 114 maybe say the only known remaining compass cue, as others might be discovered later.

Line 234 section about how long-distance navigation might be controlled, some minor changes might be helpful. As I understand things, the moth does not have any way to correct if it goes off course. I am trying to reconcile what seems to be great variation in flight directions (Fig. 2) with what would seem to be the need for a fairly precise compass. In other words, if flight directions represent what moths do, won't many of them miss the target area, given the large dispersion around the mean? Is the area they go to large enough to allow moths to reach the area with such variable headings? Some clarification would be useful.

Line 238 I think most magnetic anomalies are pretty weak, not enough to change a compass heading by much, but a few are strong enough to cause navigation problems for hikers. Are there any strong magnetic anomalies along the path of these moths? If the moths cross such anomalies this would bolster the argument that they sometimes cannot use a magnetic compass.

Lines 286-290 do not seem to me to make a convincing distinction between the beetle and moth systems except on the basis of distance travelled. I don't think the term "true compass" is helpful as it raises unanswered questions of when a compass isn't a true compass. In general, I continue to think that the beetle and moth star compasses are more alike than different, but the moth work is a valuable and elegant contribution on its own, even if the compasses are similar.

Methods: For purposes of future replication, it would be helpful to provide some general indicator of the number of responsive neurons vs the number investigated. I know that little can be concluded from negative results but having some idea (about 2% of cells were responsive vs 20%) would help future researchers. Similarly, can some indication be given about the approximate percent of moths that met the testing criteria, recognising that this probably changes with weather, date, etc.?

Referee #3

(Remarks to the Author)

I believe that the authors have addressed the concerns I raised very well. I am pleased with the final manuscript at this point. I will leave it to the other reviewers to discuss in detail the responses to the points they raised, but in general, I also think the authors did a good and thorough job addressing all concerns. I will also say that the degree of points raised by all the reviewers both for this study and future work speaks well to the notion that the paper should indeed be published in Nature.

Referee #4

(Remarks to the Author)

Thank you very much to the authors for this very detailed discussion of all comments from all four reviewers. It took me a while to read through all of it & think it was a very enlightening discussion. I am happy that the manuscript has improved even more now. All my points have been addressed, so I recommend publication. Happy New Year!

(Remarks on code availability)

N/A

Version 2:

Reviewer comments:

Referee #1

(Remarks to the Author)

The authors have appropriately addressed our comments and we feel that the manuscript is now in a good shape for publication in Nature.

(Remarks on code availability)

Same as before.

Referee #2

(Remarks to the Author)

The authors have addressed all of my concerns adequately. I am satisfied with the responses and consider the manuscript to be ready for publication.

Response to Reviewers

On behalf of all the authors, we wish to sincerely thank the four reviewers for their extensive efforts to provide excellent feedback that has improved our manuscript considerably. We really appreciate it. We will address each reviewer in turn and use **brown text** to indicate our responses. All line numbers below refer to the line numbers following revision (these are the line numbers shown in the revised version where **red text** is used to indicate changes, *not* in the track-changed version where for some reason the line numbers became irreversibly corrupted). However, things that we have changed in the manuscript to accommodate our responses will be seen as track-changes in the track-changed version (even if the line numbers disagree).

Important new addition to the text not previously seen by the reviewers: After the first version of the manuscript was submitted, our close collaborator Basil el Jundi (Trondheim), who works on the neural basis of long-distance navigation in Monarch butterflies, had important opinions about our interpretation of the tightly clustered tuning directions of unimodal central brain cells (red arrow, Figure 4b). He suggested that this clustering could have arisen due to the quiescent (fixed) state of our moths in electrophysiology experiments – i.e. this tuning might be dependent on behavioural state (as he found in Monarchs). We cannot exclude this possibility, and thus updated our text to include this caveat (lines 202-206).

Referee #1 (Remarks to the Author):

In this manuscript, Dreyer, Adden and colleagues explore the navigational mechanisms used by bogong moths to migrate seasonally. Specifically, they test whether bogong moths rely on celestial cues at night, in coordination with magnetic field cues, for long-distance navigation. They begin by demonstrating through convincing behavioral experiments that, in absence of Earth's magnetic field, moths can rely on the austral starry night sky (notably the Milky Way) to orient in the appropriate migratory direction, and that reversing these celestial cues by 180° reverse orientation by the same amount, supporting a role for stars as a compass. Importantly, they show that moths become disoriented in absence of both magnetic and stellar cues. This evidence is consistent with work in dung beetles and birds which demonstrates use of the Milky way or star patterns respectively, as directional cues. The authors further report the existence of neurons in the moth brain that respond to 360° rotations of the austral night sky by increasing firing when the moth is flying south, but not when star patterns are randomized. They show that some of these neurons are located in the optic lobes, central complex and lateral accessory lobes (steering center). In a second set of experiments, the authors show that the same neurons respond to key aspects of the night sky that correlate with the straight pattern of the Milky Way and one of its bright

spots. While these regions of the brain have previously been shown to be involved in compass navigation, the authors identified several new cell types that are likely important for a stellar compass.

In terms of the care with which the experiments appear to have been conducted, the quality of the data and the clarity of the manuscript, which was a pleasure to read, this is a very solid study that yields interesting results. However, conclusions are not always drawn with caution and the novelty is somewhat diminished by the fact that many similar conclusions have been previously drawn in other animals. **Indeed, the use of a stellar compass for long-distance navigation has been shown in birds** and the use of the Milky Way for navigational purposes was previously demonstrated in dung beetles, an ability referred to as a compass. Likewise, the neural basis of insect compass orientation and steering has been well described in insects, including monarchs, dung beetles and bees, and occurs in the same brain regions to those described here. The discovery of neurons that respond to 360° rotations of the austral night sky/Milky Way thus seems to be the major conceptual novelty. Therefore, while this study represents the first demonstration of the use of a stellar compass in a migrating insect, paired with neural recordings, authors **should be cautious of the claim that it is the first demonstration of a stellar compass in a long-distance migrant or in an insect**. While dung beetles do not migrate long distances, their use of the Milky Way for orientation is described as a compass system and has been well studied. Authors describe some of the differences in dung beetle and bogong moth stellar navigation, but nonetheless, there are still many similarities between the two and these should be acknowledged.

Many thanks for your positive view of our manuscript and for your thoughtful comments.

As 3 out of 4 referees discuss whether what we found in Bogong moths is the same or very similar to what has previously been found in dung beetles, it has become clear to us that we must address this point much more clearly in the revised manuscript, which we have now done, both in the introductory paragraph and in the main manuscript text.

Please allow us to clarify the claim we *are* making, and what we are *not* claiming. We are **not claiming** that Bogong moths are the first invertebrate to use stellar cues as a directional reference (as suggested by the reviewer in **green text** above) – clearly some dung beetles do use the stars as a basic directional compass (as we implied in more than one place in the manuscript). However, these beetles use the stars to *orient* in any random and goalless direction for a short distance. This is not navigation in a specific geographical direction, which is what is mostly implied when the term “star compass” is used in (say) bird migration studies. Some researchers do admittedly refer to the orientation behaviours of dung beetles as “compass responses” as they are directional and can be altered when global celestial cues are manipulated. Even though we never used the word “compass” in our original dung beetle studies (see below), we concede

that the use of the word “compass” is acceptable in the context of dung beetle orientation (and we have amended the manuscript text accordingly – see for example lines 58 and 284).

What we **are claiming** is that the Bogong moth is the first invertebrate to use the stars to *navigate*¹ – i.e. to find a specific, inherited, geographical bearing relative to North (i.e. a bearing in a specific geographic direction, not just fixed orientation in a random direction) and to use this bearing to travel towards a distant goal which they have never previously visited (and this bearing varies widely depending on where the journey commenced – see Figure 1b). In other words, Bogong moths need to translate stellar information into a specific geographical meaning in order to fly in their inherited migratory direction, whereas dung beetles do not need to associate stellar information with a specific geographic direction or a goal. As far as we know, only humans (e.g. with a sextant) and some species of night migratory songbirds are known to be able to do this. This is *very* different to the use of the stars by dung beetles, that use the Milky Way for randomly oriented locomotion over metres and minutes. The referee also suggests that we do not acknowledge that birds have this capacity (blue text above), but we did so in several places in the manuscript (e.g. lines 60, 147-149, 153 and 268) – we are not claiming that Bogong moths are the first *animal* with a stellar compass of this type, but the first *invertebrate*.

The senior author on the present manuscript was also the senior author on the original dung beetle study, where we wrote about the dung beetle’s use of the stars for the first time (Dacke et al., 2013). So, we are well aware of the similarities and differences in how the stars are used by the two species. In our 2013 paper the word “navigate” is never mentioned. Neither is the word “compass”. For dung beetles, the Milky Way is more like a “celestial landmark” that they use as a directional reference to hold a straight rolling

¹ Please note, we are **not** suggesting that Bogong moths are “true navigators” as defined by Able (*J. Avian Biol.* 32: 174-183, 2001). True navigators (like migratory birds) not only know (and follow) the direction to their distant goal (by using a “compass”), *but in addition* know their own position relative to that goal (by using a “map”). We have no idea if the magnetic sense of Bogong moths (for instance) endows them with a map sense (we have experiments planned to test this), but we sincerely doubt it. Monarch butterflies, which like Bogong moths also migrate to a specific destination, have been shown to lack such a map. Instead, Bogong moths (and possibly Monarchs) probably rely on recognition of a set of genetically (or epigenetically) encoded cues that differ during each of the three stages of long-distance migration and which allow them to reach their goal. Here we describe one of them used during the first “long distance” phase (their inherited migratory bearing). We now have compelling evidence (still unpublished), that a specific odour compound emitted from their aestivation caves acts as an olfactory beacon that guides them to their final destination during the third and final “pin-point the goal” phase of migration (when they are possibly less than a couple of kilometres from the cave). There are likely to be further sensory cues that alert the moths to their arrival in the broad vicinity of their destination (during the second “narrowing-in” phase of migration), possibly changes in air pressure and temperature, certain plant odours etc. Thus, since the Bogong moth only makes this journey once, it must be the case that moths rely on an *innate* understanding of cues encountered along the route. This is different to a migratory bird, that by making yearly journeys over many years, has the chance to *learn* these cues to create a true sensory map (thus claiming the title of a “true navigator”).

direction. We concede that you could (and should) legitimately call this “celestial landmark” a “compass” (since the “landmark” is at infinity with a location and orientation that are independent of the animal’s position), and Marie Dacke together with others of our colleagues have admittedly referred to the dung beetles as having a stellar compass in later publications. However, Dacke and her colleagues have never referred to dung beetles as “navigating” (they consistently refer to them as “orienting”). Thus, dung beetles are definitely not using the stars to navigate in a specific geographical direction. We thought we had adequately emphasised this important distinction in the original manuscript (e.g. in the final paragraph), but since three of the four reviewers have a similar criticism, we have now addressed this important point up front and much more clearly.

We hope the changes made in the revised version of the manuscript – which has encompassed an entire re-write of its first and last paragraphs – will avoid this misunderstanding (lines 57-64, 284-298). If, however, the reviewers still feel (for whatever reason) that our interpretation of our results is unwarranted, we will reconsider our conclusions further.

Interestingly, this comment (from 3 of 4 reviewers) has led us to go back to our old data of dung beetle rolling directions under starry skies. We specifically looked for data from experiments where individuals were forced to roll several times in their chosen rolling directions over time intervals long enough that the starry night sky had noticeably rotated. Even though this data is very limited, this reexamination showed that individual beetles changed their their rolling directions in a direction consistent with the change in the direction of the Milky Way during the same time interval (from first to last roll). This suggests that the beetles may indeed be using the broad stripe of the Milky Way as a “celestial landmark”, i.e. as a temporary directional reference to hold a straight rolling direction. As this reference direction shifted, so too did the beetle’s rolling direction. In contrast, Bogong moths remained oriented in the same (inherited migratory) direction irrespective of the time of night and the rotational position of the stars (Figure 2). This difference suggests a very different manner in which the stars are used as a compass cue in the two species.

However, despite this reexamination of our earlier results, the number of individuals for whom we have data, together with the rather short time intervals involved, do not allow us to be absolutely sure of these conclusions. This has now inspired us to plan a new series of experiments where we will specifically address this question – individual nocturnal dung beetles will be forced to roll many times over a 2-3 hour period with their rolling directions being measured as the stars rotate.

Major comments:

I would add the following further comments:

1. Previously, Dreyer et al, 2018 demonstrated that moths require a visual cue in coordination with Earth's magnetic field to orient properly. However, when experiments were conducted in Figure 2 under cloudy sky, and no other celestial visual cues were provided, moths remained significantly oriented. Authors concluded it was because the moths were using Earth's magnetic field. However, because the arenas were transparent, the formal possibility that moths could have been relying on a terrestrial visual cue has not been excluded. Given that moths oriented in the same direction in all three experimental treatments, ruling out potential bias in the set-up would have been important to reach the author's conclusion. In absence of this control, the statement in line 100-102 that "moths rely on the geomagnetic field as their main compass for navigation" is an overstatement as moths have not been shown to rely solely on magnetic cues in the absence of all other stimuli. I acknowledge that the focus of the study is on the stellar compass and not the magnetic compass, but a more robust demonstration that Earth's magnetic field is used in the absence of any other cues would have been to reverse the horizontal component of the field, or invert the vertical component to determine if moths reverse their orientation under cloudy sky. While I do not think repetition of the natural setting or laboratory experiments would be a reasonable request given the seasonal constraints associated with performing such work, the authors should try to reconcile the results from these experiments with previous work, which suggested moths rely on both terrestrial and magnetic cues. Additionally, authors should be cautious with the conclusions drawn about the magnetic compass of moths throughout the manuscript.

Good point, thank you. In these experiments, done in a transparent arena on a hilltop without magnetic coils, the moths did indeed have access to a wide panorama of possible landmarks in every direction (which was quite deliberate as we tried to make the most natural experiment possible). In our previous 2018 experiments (done on the same hilltop, but with coils), the arena was very different – it was opaque with a white interior and all dorsal visual cues (e.g. the starry sky, the coils etc) were blocked above the upper edge of the circular wall of the arena. Into this white arena we placed a black horizon around the bottom third of the inside arena wall, and above this horizon was superimposed a moveable black triangular "mountain top", a single highly dominant visual landmark in an otherwise featureless arena. Moths were highly attracted to this landmark as the only thing to fixate on while flying (except when we disrupted the learned relationship between the magnetic field direction and the direction of the mountain top – in this cue-conflict situation they became disoriented and no longer "found" the mountain top). But in the transparent arena no landmark dominated in any direction, apart from the black cubical tent (7 m distant) that housed the experimenter (see Extended Figure 1a). Thus, if a single landmark (e.g. the black tent) played a dominant role in navigation, one might possibly expect all moths to fly towards it (as we found moths doing with the dominant "mountain top" in our 2018 paper). If instead, within the

wide panorama of landmarks, no particular landmark dominated (not even the tent), then one might expect that each individual moth would choose a different random landmark and fly towards or relative to it, thus resulting in the population as a whole being disoriented. Yet neither flight scenario (landmark-directed or disoriented) arose – the moths all flew in roughly the same direction, but one that aligned with their inherited migratory direction.

Nonetheless, we did try to eliminate the possibility that landmarks inadvertently affected the flight directions of our moths (as desired by the reviewer). As we explain in the Methods (lines 746-749): “The same moths were used for orientation measurements at both times, and moths that were flown in one arena at the earlier time (and saw the black tent on their eastern side) were flown in the other arena at the later time (and now saw the black tent on their western side), and *vice versa*.” Also, there was a stand of trees close to the arena on the eastern side of the tent, and single trees close to the arena on the western side of the tent, so even other panoramic landmarks (apart from the tent) differed markedly in their spatial positions in the two arenas (which I have now added to the methods – lines 749-751). We found no significant difference in the flight orientations of the same moths flown in the two arenas. We have added words to the main text where we point out our methods to eliminate landmarks as a cue (lines 92-93).

Thus, we are not sure how we can reconcile our 2018 results with our current results as the reviewer suggests we do, as the two arena experiments were so different. In the 2018 paper we concluded that Bogong moths may use their magnetic sense to determine their inherited migratory bearing and then fixate on a distant visual landmark in the same direction that they then fly towards (like we might do when we are hiking with a magnetic compass). Even though we have not ruled out that possibility here, our swapping of moths from one arena to the other without noticeable effect on flight direction suggests that landmarks in those particular experiments played a negligible role in determining moth flight directions (possibly since the landmarks were present in all directions, unlike in 2018). Certainly, if the moths were *not* relying on a magnetic compass, but instead relied on a random landmark to orient, then they would likely be disoriented as a population since there is no *a priori* reason why the moths would all choose the same landmark. Moreover, there is no such thing as a terrestrial landmark that inherently signals (say) north and which can then be used to find an inherited migratory bearing. This last fact alone leads one to conclude that the magnetic field, and not visual landmarks, must be responsible for their observed oriented flight behaviour since it is the only cue available that can allow moths to determine their migratory bearing. But relying on a magnetic compass to find their migratory bearing does not rule out the possibility that the moths (in either arena) afterwards looked for a visual landmark in the same direction to fly towards. We have no way of knowing whether or not they did this. But our 2018 findings certainly suggest they could.

Thus, in summary, our feeling is that the conclusion as it is stated still holds and should remain unchanged (lines 116-118): “The most parsimonious explanation for these results is that Bogong moths rely on the geomagnetic field as their main compass for navigation”. The conclusion becomes less parsimonious if we attempt to invoke an influence of landmarks that we have not observed. However, we have softened the text to remove “main” here, and place the results in the context of the overcasts conditions: “The most parsimonious explanation for these results is that in overcast conditions Bogong moths rely on the geomagnetic field as their main compass for navigation”.

On a final note: The reviewer writes: “In absence of this control, the statement in line 100-102 that “moths rely on the geomagnetic field as their main compass for navigation” is an overstatement as moths have not been shown to rely solely on magnetic cues in the absence of all other stimuli”. We now have recent preliminary (and now more recently replicated) evidence from the lab that moths can indeed rely solely on magnetic cues in the absence of all other stimuli. In our new experiments, under randomised stars, tethered moths flying in a natural geomagnetic field oriented in their inherited migratory direction – when the field was turned by 180°, the moths turned and flew in the opposite direction.

2. It is surprising that the authors did not address or discuss the possibility that polarized light could be used as an additional navigational tool in bogong moths. Polarized light cues are present even at night, and contribute to the dung beetle compass system (in addition to numerous other insects). Would polarized light be a potential navigational cue for moths? As noted above, the use of the Milky Way for short distance navigation has been well-described in dung beetles, but this work is only minimally addressed by the authors. Addressing some key similarities and differences between the mechanisms underlying dung beetle use of the stars, and bogong moth use of the stars in light of the different navigational types of these insects would strengthen the current manuscript and its importance to the field.

Thanks, these are very good points, and we agree that they do need to be addressed as they do represent a glaring omission. Like basically all insects, Bogong moths have a dorsal rim area and can see and analyse polarised light. It is certainly possible that they use it for navigation, although we have seen no evidence of it in our experiments. Our lab behavioural experiments reveal a stellar compass even though the set-up is devoid of polarised light. There is a polarised light pattern formed around the moon, but in our outdoor experiments we saw no effect of the movement of the moon (and its polarisation pattern) on the flight directions of moths from earlier to later in the evening. On moonless nights (when the sun and moon are greater than 18° below the horizon), there is no polarised light in the sky (the stars each give off a weak pattern, but due to their multitudes, these patterns interfere with each other to provide no useful directional signal). But moths happily migrate on moonless nights (we have kept very accurate

records of moth catches relative to moon phase over the last 15 years in our field logs). Interestingly though, during their summer aestivation moths emerge from their caves exactly at sunset and fly in huge circles around the mountain for about an hour before re-entering the caves (Wallace et al. (2023) Camera-based automated monitoring of flying insects in the wild (Camfi). II. Flight behaviour and long-term population monitoring of migratory Bogong moths in Alpine Australia. *Frontiers in Insect Science* 3:1230501). At sunset, the polarised light pattern around the sun is oriented N-S across the entire dome of the sky. This, together with the glow of the setting sun in the west, the N-S geomagnetic field and the slowly appearing stars would possibly allow all 4 of these directional cues to be calibrated against each other – as seems to be done by night-migratory birds. It could well be the case that the wide circular flights of Bogong moths at dusk are for compass calibration where polarised light plays a crucial part.

We have added a paragraph to specifically address the possible role of lunar polarised light as a compass cue, and have placed this in the context of what we understand from dung beetles. We hope that this important addition to the text (lines 100-106) satisfies this very legitimate criticism.

Finally, just to reiterate, dung beetles do not use the stars to navigate (see reviewer's text in blue above). They use them for short-term orientation. In stark contrast, Bogong moths *are* using the stars to navigate in the sense that they determine a specific geographical heading. We felt that we had already clearly made this distinction between dung beetles and Bogong moths in the original manuscript, but in light of your comments, we have now made the distinction much clearer in the revised manuscript (lines 57-65, 284-298).

3. Results indicate that a stellar compass and a geomagnetic compass (pending that the result under cloudy sky are not due to other visual cues) may each be sufficient to guide Bogong moths in their migratory direction. However, in natural conditions, both are likely to operate together. Can the authors speculate/discuss on the possibility of a hierarchy between cues when they co-occur?

This is an excellent point and will be the subject of already planned new experiments. The two compass certainly act together (it would be crazy if they didn't). At present we have no indication from our current results which of the two cues – magnetic or stellar – is the dominant of the two cues. However, this topic has been studied extensively over 2-3 decades in night-migratory songbirds, where it has become clear that no fixed hierarchy exists. Which compass/cue dominates depends on many factors including the ecological context, the availability of the cues, and probably the reliability of the cues. It even seems to vary within the same species/individual over a single migratory season, so this not something we can solve in the present study. In light of your comment, we have added a very short text that discusses a couple of conditions under which one compass is likely to dominate over the other, but we feel that to say much more would be a bit too speculative (lines 237-239).

4. The idea that the central complex may be the site of integration for starry sky cues (lines 228-230) is supported only by physiological data from a single neuron. I think this needs to be acknowledged.

Thanks for picking this up. We dye-filled three central complex cells (see Figure 5a and b middle panels for one cell, and Extended Figure 7 for the other two cells). However, we held the cells only long enough to physiologically characterise two of them (the cell in Figure 5, and Cell 181022_003 in Extended Figure 4). So, you are quite right – the idea that the CX is the site of stellar integration is supported physiologically by only two cells (although not by a single cell as you thought). We have amended the text to acknowledge this (lines 216-219).

5. In the method sections, many custom-written codes are mentioned. However, these codes are only available upon reasonable request. I see no justifiable reasons for not sharing them publicly.

We quite agree. However, they were provided publicly:

https://github.com/stanleyheinze/Starry_sky_code

We are unsure why you were not informed of this. Our apologies.

Minor comments:

1. Throughout the figures, the purple color being used is barely distinguishable from the blue one. More contrasted colors would help the readers.

Thanks so much for noticing this. As far as we can tell, only one figure contains panels where blue and purple are used together (Figure 4), and we agree this is not optimal. We have fixed the colour scheme as suggested.

2. Figure 4: Panels d and e should be switched in the figure as e is described first in the text. What kind of plots are being shown in panel e? These are not typical plots and an explanation of what they are, which is currently missing in the legend, is needed.

Indeed, this is true. Reviewer 3 also complained about this, and the entire section has been re-written in a more logical order of concepts and the panels in the figure moved accordingly (lines 168-192).

The plots in panel e have now been changed to more conventional box-and-whisker plots (i.e. boxplots).

3. Extended data Figure 1: in panel d, the flying moth is not circled. Either circle it or remove “(circled)” from the legend.

Whoops, well spotted. Thanks. This is now fixed with a circle around the moth.

Referee #1 (Remarks on code availability):

The codes were not available.

See comment above.

Referee #2 (Remarks to the Author):

This study investigates migratory bogong moths and whether the moths can use star patterns to hold a consistent migratory heading. In arenas outside, where the moths had access to natural cues, tethered moths flew in their migratory direction under both clear and cloudy skies. In the lab, moths exposed to a projected view of star patterns in a zero magnetic field (to prevent use of their magnetic compass) moved in the migratory direction. When the star pattern was reversed the moths flew the other way. Electrophysiological recordings were made from 28 neurons in the optic lobe while star patterns were rotated. The neurons responded to these rotations and also to rotations of bars and circles of light simulating some aspects of the Milky Way. A subset of neurons was traced to the central complex and to the lateral accessory lobe, two areas of insect brain involved with orientation and steering. It is concluded that the moths have a star compass that is used in their migration and that neurons in the visual system respond to star patterns or at least some element of it, most likely the brightness of the Milky Way.

This is an interesting, high-quality study. The orientation mechanism is not entirely novel inasmuch as use of the Milky Way as a compass for orientation has been shown before in dung beetles (e.g., Dacke et al., 2013, *Curr. Biol.*) and some limited evidence for use of stars by migratory moths has been reported (e.g., Sothibandhu, 1979). At the same time, the study elegantly extends the dung beetle findings to a long-distance migrant for the first time. Also, although there has been much work on the neural basis of celestial orientation during daytime (butterflies, desert ants, locusts, etc.) this is, to my knowledge, the first neural recordings in insects to suggest responses to star patterns.

Although there is much to admire in the study I was confused by several issues and unable to fully follow several aspects of it. Some conclusions might need to be modified and some additional information is needed.

Many thanks for your positive comments about our study. We will try and resolve any confusions as best we can. Thanks for bringing them up.

Major comments:

1. The data in Fig. 3 make a strong case that moths can use some aspect of the night sky for orientation, but it does not seem clear yet whether the moths have a true compass or a different kind of response. Having a compass means that an animal can hold headings in any direction relative to a cue, just as a hiker with a standard compass can travel N, NE, SW, etc. The moths just travel north and south, so it doesn't seem clear that the moths can hold a path at any angle to the Milky Way, as opposed to just flying toward it or away from it. Line 240 describes the Milky Way as a bright extended stripe of light brightest in the southern half of the sky. If moths just fly in the direction of the brightest area of sky when going south and away from it when going north, this would not require a compass, just a seasonal reversal of preference for traveling toward/away from bright sky. Are there additional data that might support or rule out this interpretation? Similarly, an earlier study by Dreyer (Curr. Biol.) showed the moths changed direction when the magnetic field and a visual cue were moved together. Could the Milky Way be the natural visual cue that moths use? If so, moths might use the Milky Way as a sky landmark rather than as a stellar compass.

These are very important points, thank you. Indeed, the moths we used appear to travel roughly north in autumn and south in spring (as seen in Figs. 2 and 3). Almost all the moths used in these experiments were captured at the same place – near Mt. Selwyn in the northern part of the Kosciuszko National Park, a linear distance of approximately 68 km NNE from the nearest aestivation cave. At the start of the study, a very small number of moths were also caught at Dead Horse Gap near Thredbo (just south of the nearest aestivation cave) but these were used for electrophysiology (the Methods have been updated to reflect this – lines 528-530). From early 2018, due to its closer proximity to our field lab, we confined our moth capture to Mt. Selwyn (for both behaviour and electrophysiology). The moths we caught at Mt. Selwyn were a very tiny subset of all moths making their migratory journey – approximately 2.2 billion moths are estimated to arrive in the Kosciuszko National Park from all over southeast Australia during the spring and the vast majority of them survive the summer to depart from the Park during the autumn – also across a huge range of return directions. We have no knowledge of where our tiny subset of moths was heading, but two things are certain: (1) moths travel from (spring) and to (autumn) all parts of NSW, southern, southwest and southeast Queensland, and western Victoria, so a northerly-southerly directional axis of travel would be highly disadvantageous to nearly all moths except those expressly heading to and from northern NSW and southern Queensland, and (2) the moths we caught at Mt Selwyn over the course of the three separate years of this study all flew roughly N to NNW in autumn and S to SSE in spring, suggesting that Mt. Selwyn lies under a flyway for moths heading to northern NSW and southern Queensland. Unfortunately, for logistical reasons we cannot easily repeat these experiments on moths caught (say) in the Victorian alps which might be expected to be heading to and from western Victoria (i.e.

along an east-west directional axis of travel, meaning they would become severely lost if they simply used the Milky Way for orientation in the manner suggested by the reviewer). We would dearly love to do this experiment, however. To make this clearer, we have underscored this important point in the main text (lines 70-72 and 122-124) and added text to the Methods (lines 522-528).

Another piece of evidence that suggests that Bogong moths are not simply flying north and south comes from our 2018 paper (mentioned by the reviewer above). In Figure 2 of that paper, moths caught during the spring at Mt Kaputar (near Narrabri) in northern NSW, were clearly heading southwest, in a direction that is highly favourable for reaching the Australian Alps from that location (and quite different to the directions of spring migrants caught at Mt. Selwyn). Moths caught during the autumn at Dead Horse Gap in the Kosciuszko National Park, were heading northwest, in a direction that might be expected at that time of year (again we do not know the destination of this cohort of moths). Admittedly the directions of these moths are similar to those we caught at Mt. Selwyn.

Finally, while the autumn night sky shows the brightest part of the Milky Way in the south (earlier in the evening), this is not the case in the early evening spring night sky where the brightest part is arguably in the west (Figure 3a), which you also noted below in another comment. Moreover, in both seasons the stars dramatically shift their positions over a single night, moving the brightest parts of the sky to radically new positions as time progresses (and often well away from due south). For these reasons too, it is highly unlikely that the moths simply use some simple brightness feature in the south to find their direction. Moreover, if simple photaxis is used then the presence of the moon would cause major problems for orientation as it is by far the brightest object in the sky, even when it is less than half its full size.

Thus, our assertion in the manuscript that the southerly biased distribution of light in the Milky Way might be useful as a navigational cue for Bogong moths (e.g. in line 240 of the submitted manuscript) is probably too simplistic and thus misleading and needs to be modified. This light distribution has been suggested as useful for the short-duration ball-rolling orientation of dung beetles (references 7 & 10), and indeed for dung beetles it would be perfect, but for an insect like the Bogong moth that navigates for many hours each night (when the Milky Way dramatically changes its position and orientation in the sky), as well as in different seasons when the night sky differs significantly, this is too simplistic. “Southerly biased” is not intended to imply “located exactly south” (which would indeed be useful for the simple positive or negative phototactic orientation mechanism suggested by the reviewer). We have modified the text accordingly to imply that this is not the case (lines 267-282).

2. I was confused by statements that the moths use the starry night sky to distinguish N,E,S,W cardinal directions (e.g., lines 32, 126, 238). Maybe the moths can do this but

there seems no evidence presented here that they can do this or need to do this. They only fly north or south and might do that just based on brightness of the southern sky without reference to cardinal directions. More broadly, an animal does not need to know cardinal directions to maintain a heading, for example a fish can move straight upstream toward a food odor. I believe these statements are in error or, if they are not, need to be clarified.

Thanks for pointing this out. We agree that cardinal directions is likely to be misunderstood here. We have removed “cardinal directions” everywhere, except for one place in the methods section where we actually mean N, S, E, and W specifically. We now state that Bogong moths use the stellar cues to distinguish and move in “specific geographic directions”. Regarding moths simply “flying north or south”, please see our response to the previous comment.

3. I could not fully follow Fig. 4b. There are data points with two different shades of gray but the darker shade does not seem to be defined. Also, are unimodal inhibited cells included on this graph, so that peaks of inhibition of some neurons are plotted in the same area as peaks of excitation for others? I would suggest plotting in separate panels in Fig. 4, or in supplemental, the unimodal for spring, the unimodal for fall, and the direction-sensitive unimodal and bimodal. On the unimodal data it would help to indicate which are inhibition peaks and which are excitatory peaks. At present it is difficult to tell from this figure what is happening. Also, although it is true that all unimodal when grouped together show maximal responses with the average in the south, it also appears that the spring (light blue) and autumn (dark blue) might be different, with the spring bimodal (peaks in SE/NW) and the autumn unimodal SW. Is a seasonal difference possible, perhaps reflecting changes from moving south to moving north?

Thank you for this very helpful suggestion concerning Figure 4b, which we agree is confusing and data intense. We have done two things to address the problems: (1) we have coloured the data points for the bimodal cells so that their pairs are much more obvious in the figure, and (2) we have made a new extended data figure (Extended Data Fig. 3) which separates the data from the different cell types exactly in the manner you have suggested. We agree that this is a major improvement (thank you!).

Regarding the second question, the overall median φ_{\max} , calculated over all cells recorded in spring 2018, was 220.8° (SW). The overall median φ_{\max} for autumn 2019 and 2020 was 145° (SE). So there is a seasonal difference, shifting from SW in spring to SE in autumn. However, the overall φ_{\max} remains in the southern hemisphere, throughout the year. Moreover, when we performed a Mardia-Watson-Wheeler test to determine whether the spring and autumn data are significantly different, we found that they were not ($p > 0.2$). This has been added to the text (lines 197-199).

4. In the context of moths responding to the brightest area of sky illuminated by the Milky Way, it would be worthwhile to discuss the role of the moon, as this illumination in Fig. 2A, 2B seems opposite that of the Milky Way. There was no moon in the indoor experiments, which is an important way that the tests deviate from a natural setting. It would be helpful to discuss this. Would the use of the stars be diminished during full moon? Are the moths able to filter out the moon somehow?

Thank you for this very interesting point. As mentioned above, the fact that Bogong moths are oriented in the same, season-specific migratory direction both with and without the moon present is a strong argument against any simple phototaxis-based orientation mechanism. While we have not specifically tested the idea that the moon can function as a cue for long-distance navigation, the moon (unlike the sun) is a terribly fickle cue for long-distance navigation, rising and setting at different times of night (and day) and being present in different places in the sky throughout each month, and constantly changing size and brightness. Indeed, the moon was only present in the outdoor experiments, but we found no evidence that it had any affect at all. This is similar to the results obtained in birds, where the moon can sometimes be an artefact in behavioural experiments with captive birds, but so far there is no convincing evidence that the moon provides geographically relevant cues for long-distance migratory birds.

Nonetheless we have added text about potential lunar cues and why we feel they play no role in the navigation of Bogong moths (lines 100-106, 278-282)

5. I did not understand the distinction drawn between the dung beetle compass and the moth compass (lines 248-252). Many others have referred to the dung beetle's use of the Milky Way as a compass (e.g., reviews by Dacke, el Jundi, etc.) so I was surprised by the statement that dung beetles do not have a star compass (line 248). Similarly, on line 57, I don't think it is correct to say that 'no invertebrate is known to use the stars to discern compass direction,' as dung beetles do exactly that (Dacke et al., 2013). Really there seems better evidence for a true compass in the beetles because they use the Milky Way to move in any direction, not just north or south. The beetle and moth systems seem likely identical, just used over different distances (short for beetles, long for moths). It would improve clarity to state this if it is true.

Since three of four reviewers take up this point, we have provided an answer to this query in a single response at the beginning of this document (as our first response to Reviewer 1) and have modified the manuscript text accordingly.

6. There is room for improvement in discussing other possible cues besides stars and magnetic fields. Most animals use multiple cues so there seems little reason to assume that star cues and magnetic cues are the only ones for moths. Dung beetles use

polarised moonlight, so why not moths? The moon itself might be a cue (Sotthibandhu, 1979), maybe odors when they get near the end of the migration, and so on.

It is not difficult to agree with you on these points (which are similar to your point made about the moon above), and we agree, such a discussion holds merit. Thus the new paragraph about the moon (lines 100-106, see our response to your Comment 4). Moreover, just to let you know, we now have very strong evidence that there is a specific odour coming from the caves that guides the moths at the very end of their long journey to their destination – but that is another story. Our manuscript deals with the so-called first phase of a long migratory journey – the “long-distance” phase which takes the migrant from its starting point to the broad vicinity of its goal (we have now stated this clearly in the manuscript (lines 51, 63), and added a new reference: Mouritsen, H. (2018) Long-distance navigation and magnetoreception in migratory animals. *Nature* 558, 50-59). The odour cue we have found represents the third and final phase of the journey – the “pin-pointing the goal” phase. This, however, is outside the scope of the present manuscript.

Minor comments:

1. Image in 3A seems to have Milky Way more in west than in south. Is that the true situation? If so, please reconcile with statement about the Milky Way constantly remaining in the southern part of the sky (lines 244-246).

Yes, you are quite right. The image is correct and we have now modified the text to better reflect the variation of the location of the brightest part of the sky (lines 267-282). This fact also has bearing on our interpretation of our results (almost certainly excluding the simple phototaxis alternative you suggested earlier), as explained in our response to your first comment above.

2. It would be helpful to clarify that the bird star compass and the moth star compass (if it is a compass) operate on different principles, with birds in the northern hemisphere using the non-rotating part of the sky as a reference.

Yes, excellent point. This has been added (lines 148-149).

3. For the electrophysiology, it would be helpful to provide more information about how the four types of neuronal responses were identified and recognised. Not very much information is given. With some of the directional bimodal data, it does not always seem obvious that there are two peaks instead of one.

Thanks for pointing this out. The following text has been added to the Methods (lines 877-879): “Responses were classified as unimodal (models M2A, M2B or M2C in the R-library CircMLE) or bimodal (models M4A or M4B) based on the Akaike information criterion, AIC.”

4. Please specify the rate at which the projected sky pattern was rotated. That was listed for the dot and bar but not the star pattern unless I missed it.

Very sorry: the wording in the Methods was ambiguous. The rotation rate was the same for both the stars and the dot-and-bar pattern. We have amended the text to make this clearer (lines 793-794). Thanks for noticing this.

5. Line 90: Are the moths really unaffected by the nightly movement of the stars and moon or do they have a time-compensated star or moon compass? Isn't either possible?

Good point. Time compensation is not impossible, but we have no proof of it. When Stephen Emlen asked the same question about the stellar compass of night migratory Indigo buntings, he found that they did not time compensate. In an earlier version of the manuscript, we included this as a possibility, but the fact that the moths could still orient when the stars were covered by cloud led us to the most parsimonious explanation for maintenance of the migratory heading on clear nights while the stars and moon were visible and moved – the geomagnetic field. This of course does not rule out time compensation, but it is not needed to explain the results. So, we decided to leave it out. Testing the idea of time compensation requires time shifting Bogong moths and exposing them to starry skies that do not match their biological clock. We have plans to do such experiments in the future.

6. Line 148: Does 28 neurons refer to the total number impaled or the number that responded? Were there neurons that did not respond? Please clarify.

28 neurons refers to the number of neurons that responded. We impaled and recorded from many more neurons, but found no responses to the stimuli we presented. In fact, the majority of neurons impaled did not respond to any shown stimulus. This is expected, as we impale cells randomly and only a small fraction of the hundreds of thousands of neurons in the moth brain would be involved in processing sky compass information. To clarify this point the following text was added to the sentence mentioning these 28 neurons (lines 208-210): “Of the 28 neurons *that responded to our stimuli*, we anatomically identified ten cells by intracellular tracer injections (Fig. 5a).”

As only one neuron per brain can be dye filled, only neurons we suspected of responding to our stimuli were filled and processed for histology. Recordings of non-responding neurons without histology are impossible to interpret in a meaningful way, as the vast majority of brain neurons would fall into this category. A subset of non-responding neurons was nevertheless dye-filled, as occasionally suspected responses were not significant after analysis. However, an absence of response, given a sample size of 1, does not necessarily mean that the neuron never responds to this stimulus, but might instead arise from issues such as the moth not being in the correct behavioural state, or a deteriorating state of the animal during the recording. A negative result is thus much

less informative than a positive one. We therefore refrained from drawing conclusions from these neurons, and did not include them in the dataset.

Referee #3 (Remarks to the Author):

This is a very interesting and clearly presented paper describing the use of stellar cues in the night sky for navigation of the Bogong moth. It is the only case that I am aware of, in insect species where night sky cues are used in this way. It, thus, should be of interest to the wide audience that reads Nature. The work relies on excellent behavioral data and is backed up with intracellular recordings in 3 brain regions that are known to monitor and process visual cues used in navigation in a wide range of insect species. The paper clearly shows that either the geomagnetic or visual cues in the starry night are used by this migratory species and raises numerous interesting questions for future research such as how the visual and geomagnetic compasses interact and how navigational information is stored for the return trip and passed on to the next generation for their first migratory trip.

I must say, I was a bit confused when I first read about fig 2. My conclusion was that the moths were simply not using visual cues. Then I read the final paragraph of the section and found that the authors were also coming to that conclusion and then in the next section tested this with a very clever experiment. That's fine. I came away convinced after discussion of figure 3 that the stellar compass is important by itself. However, it left me with questions about the relative importance of the two compasses under normal conditions and how they might interact. I will get to that below.

Many thanks for your positive comments about our manuscript. Hopefully we can address your question concerning the relative importance of the two compasses below.

Major comments:

1. The behavioral data certainly make a convincing argument that the moths can use either the stellar compass or the geomagnetic compass when only it is available and the neural data clearly show that visual information is available in the various brain regions. The neural data are not surprising given the wealth of visual data taken from these brain regions in various insect species. Nevertheless, they are important to the story. However, it seems to me that the more crucial question for this species is how the two compasses normally interact. These authors have previously presented data showing that the Earth's magnetic field can be used by these moths to direct their flight and here they focus on the cues found in the night sky. Normally both compasses are available on a clear night. I understand why you would want to have the geomagnetic compass when the sky is obscured, but why do you need the stellar compass at all? The geomagnetic compass is always there under natural conditions when it is not being blocked in the simulator. Is there an advantage to adding the stellar compass over using the

geomagnetic compass alone? Is the stellar compass more accurate on its own than the geomagnetic compass or do the two working together provide a better navigational solution?

Thank you – so many good and interesting questions. We want to know the answer to these too, and have many experiments planned to address them. Our feeling is that two compasses are simply better than one since if one of them becomes temporarily unavailable or diminished in saliency, the other can take over. In the case of stars, cloud cover of various degrees (up to totally overcast) can reduce or even eliminate their usefulness. The geomagnetic field is ubiquitous, this is true. But even this cue can be perturbed by local variations in ferrous materials in the ground (e.g. large iron-rich outcrops) which can strengthen, weaken or perturb the local magnetic field (known as magnetic anomalies). National geological surveys have produced and regularly update national maps of magnetic anomalies (the latest Magnetic Anomaly Map of Australia (Edition 7), produced by the Australian Government, Geoscience Australia, is from 2020). Moreover, geomagnetic storms have the potential to severely disrupt the geomagnetic field as well. There is also speculation, based on orientation experiments in birds that show that the directional choices of birds in experiments where only magnetic cues are available, seem to be more scattered than when celestial cues are available, either alone or in combination (a similar conclusion is also drawn for other species, as discussed in a commentary article on the topic by the senior author and others: Johnsen, Lohmann and Warrant (2020). *Animal navigation: A noisy magnetic compass? J. Exp. Biol.* 223: jeb164921). Thus, both compasses – together with other possible sensory cues – could in principle back each other up in times of trouble. We have added some text to clarify these issues (lines 237-239). We have experiments planned to test the role of the two compasses in the lab by comparing the orientation of tethered flying moths with stars alone, geomagnetic field alone and with both cues together, to see what (if any) differences there are. Then we plan to change the saliency of these cues (e.g. by simulating a change in the level of cloud cover in the case of stars) to test whether the cues are in an obvious hierarchy. We also plan to place the two cues in unnatural configurations relative to each other (i.e. perform cue conflict experiments) to see if one compass wins out over the other and decides the bearing, or whether the bearing is intermediate to the two bearings predicted by the two compasses.

However, as our co-author Henrik Mouritsen (who does a lot of bird experiments) points out: the topic of hierarchies and relative importance of the *three* different parallel compasses in night-migratory songbirds (sun, stars, magnetic field) has been studied extensively over 2-3 decades. The results are still unclear except that no fixed hierarchy exists. Which compass/cue dominates depends on many factors including the ecological context, the availability of the cues, and also probably the relative reliabilities of the cues. It even seems to vary within the same species/individual over a single

migratory season, so this is likely to be an extremely complicated subject and not something we can solve in the present study, or possibly not even in the next decade.

2. I wonder if the setup the authors used could be modified to examine how the moths are using both compasses simultaneously. What if you placed the moths in the simulator with the magnetic field left alone and intact and then manipulated the sky image? What if the sky were rotated not so much as in figure 3 but more than the few days used in figure 1. Perhaps, you could use intervals matching say 1 or 2 month time segments. Would the moths still orient according to the geomagnetic seasonal cues or would they change their orientation according to the altered visual time sequence preferring that compass if it is available? If the sky is a major cue, one would expect the direction to rotate with the sky rotation (even though the moths may never experience this kind of rotation in their normal life span). If not, then one would conclude that the stellar compass, while present, may be irrelevant. Another way of testing this would be to rotate the sky image in the same increments, but in random directions. Would the moths ignore this confusion and rely solely on the geomagnetic compass or orient according to the situation at any particular month's sky? Or another alternative is that they would orient to a direction between the two compasses suggesting that both are normally used in tandem. Yet another possibility is that one of the compasses is normally used to calibrate the other. I am reminded here of how the visual system of owls has been shown to calibrate the auditory map as shown by Brainard and Knudsen's classical studies using prisms to shift the owl's visual field. Perhaps the geomagnetic compass is refined by the visual cues to accomplish something like the directional switch that occurs between seasons.

I understand that the kinds of experiments that I suggest above may be beyond the scope of the present study. Nevertheless, I think it would be useful to have the authors mention the question of interactions between the two compasses and how they think they may interact, especially, since the last section of the paper points out the existence of these two compass mechanisms.

Many thanks for these extremely interesting thoughts, most of which we ourselves have had (and indeed plan to pursue in terms of future experiments). A major thrust of our future research is to sort out the relative roles of the two compasses as we outlined in the response to the previous comment. Presently, at the end of the manuscript we write: "While the Earth's magnetic field certainly acts as one such compass (Fig. 2c)⁴, the southern night sky clearly acts as another, with each of these compasses likely taking over from the other when the salience of either diminishes or fails (Figs. 2c, 3c,d). However, exactly which celestial features are used for the stellar compass, whether the common directional tuning of stellar compass neurons plays any role in the computation of desired heading, and how stellar, magnetic, and any other hitherto unknown sensory cues, are behaviourally and neurally integrated for robust navigation (including how and if they are calibrated against each other⁴⁰), all remain enticing topics for future research." Here we already mention the interaction between the two compasses in terms of one

taking over from the other if the other loses saliency, which we have data to back up. But to go beyond this (e.g. by postulating a hierarchy for these cues), would be pure speculation since we currently lack the data.

3. I found the neural data presentation in figure 4 a bit confusing. The text seemed to bounce around among the various sets of data. First the data in 4a are briefly mentioned and then the text switches to the graphs in 4e before an in-depth discussion of 4c and then going back to the details in 4a then finally mentioning 4d. I would suggest discussing the types of cells described in 4a first and then moving to the data in 4b and the more detailed discussion of 4c. I also found the discussion of 4c difficult to follow. I think this needs some clarification about what the dots and bars represent as well as the two rows of graphs.

Thank you, and sorry for the confusion (which is shared by Reviewer 1). Indeed, this could have been better written. We have re-arranged and modified the text to improve its logical flow as you have suggested and swapped panel labels in the figure accordingly (lines 168-192).

Referee #4 (Remarks to the Author):

In their manuscript entitled “Bogong moths use a stellar compass for long distance navigation at night”, Dreyer et al provide an exciting study demonstrating how Australian bogong moths use a combination of both the Earth’s magnetic field and visual cues (i.e. the starry night) as orienting cues during their long-range migration. These findings appear to show striking similarities for what has been demonstrated for navigating birds, thereby demonstrating that these rather sophisticated navigational skills also exist in the invertebrate world.

The manuscript combines behavior experiments, electrophysiology, and neuroanatomy, using a non-model organism which is not exactly easy to work with. Overall, the data are very convincing and demonstrate the amazing abilities of invertebrate navigators, adding a beautiful nocturnal facet for comparison with the Monarch butterflies. I have a couple of major points that the authors should address, since I believe this is an exciting manuscript for a general audience, and addressing these points should clarify the messages for the reader considerably. If these points were to be addressed, then I think this manuscript would be a great addition to Nature.

Many thanks for your positive comments on our manuscript. I hope we can address your comments to your satisfaction.

Major comments:

1. Orientation of the milky way: It is not clear to me how the orientation of the stripe formed by the milky way changes over time (days, weeks, years). This needs to be clarified for the general audience.

We have tried to expand this a bit more without adding too many words (*Nature* has strict word limits) – see lines 84-87 and 276-278.

Figure 3 insinuates a general North-to-South orientation, yet Figure 2 shows an almost West-to-East orientation. Related to the behavior experiments in Figure 3: Have the authors ever projected skies rotated 90 degrees, i.e. with a milky way oriented in West-to-East direction? Two possible outcomes are conceivable: a) a 90 degree switch in heading, or b) undirected behavior (since the orientation of the milky way is ‘off’. Has this experiment been done? Note: I understand that the authors come to the conclusion that the milky way may in fact not be perceived as a stripe, but instead leads to a luminance (?) bias in the South. However, their experiments with a rotating stripe suggest that cells in the animal’s brain indeed respond to stripes.

At this stage it is hard to say with certainty exactly which aspects of the Milky Way are used for navigation. Yes, its stripe-like length does rotate substantially over the course of the night, and its brightness gradient and its brightest part move to wildly different parts of the sky as the night progresses (although still to positions that are mostly within the southern half of the sky, but not consistently due south). We have so far always chosen a 9:30 pm sky as our lab stimulus because we usually begin our experiments around 8.30 pm and finish around 10:30 pm, meaning that our chosen starry sky is a proxy for the sky at 9:30 pm \pm 1 hrs. We have yet to look at the specific effect of changing the sky according to the local time over the entire night while recording moth flight directions to see what (if any) influence this has on flight orientation (in the absence of a geomagnetic field). But these experiments are planned.

2. The authors mention that the stellar pattern changes over time, yet they do not mention whether the cells they record from in Figure 4 manifest any interesting changes over time. How long did individual recordings last? Were they too short to show any temporal effects? (I would assume so). More interestingly, could the large spread of data points in Figure 4c also be the result of temporal effects? Have the authors plotted these dots as a function of the date when these data were recorded? I am curious whether there is a systematic shift over time.

Indeed, cells were held for very short periods of time – usually a minute or less, or in exceptional cases a few minutes. Thus, at a single cell level, temporal changes were impossible to detect. We did check however if there is any relationship between the recorded φ_{\max} and the time at night we did the experiment, and we found nothing (“Time difference” on the x-axis refers to the difference between the time of the sky snapshot (21:30) and the recording time, in minutes):

Figure: φ_{\max} values of neural responses plotted against recording time relative to the time of the starry sky stimulus image (21:30) – a cell recorded at 21:00 would have a time difference of -30 minutes. No pattern is apparent, and we could find no correlation between φ_{\max} and recording time, neither for autumn recordings nor spring recordings, nor for the dataset as a whole.

We tested for trends/correlations and found nothing - a regression analysis found nothing either. So it's probably fair to say that recording time does not have an effect.

3. Alternatively, most temporal changes of the stellar pattern could be averaged out, by generating some 'Southern blob' of brightness, I assume. The final sentences of the manuscript seem to allude to such a possibility. But aren't the dot/bar stimuli then a bit ill-chosen? Have the authors ever tried to destroy the resolution of these stimuli, i.e. generated more diffuse patterns of brightness, to test whether this is what drives wither behavior and/or the activity of the recorded cells?

I must confess that we haven't tried this, but it is an interesting idea. The bar and dot were our attempt to try and emulate the most obvious features of the Milky Way to a human observer. They are clearly very artificial but interestingly they resulted in similar responses in cells to those we obtained using the starry sky, which gave us hope that we had done something right. But we agree, taking natural starry sky images and systematically removing parts of them (e.g. the hazy background stripe of the Milky Way to leave only the visible stars, or vice versa) to test the effects on behaviour or cellular physiology is a great idea, and one we can pursue in the future.

4. As a direct follow up question to this, one wonders whether the observed behaviors are really in response to a stellar pattern or whether they could also be simpler, like phototaxis, or rather menotaxis? Could the authors elaborate on how stellar navigation

is different from menotaxis in response to a celestial body? Directly related to that: Could the authors elaborate on why they think that the moon does not influence the behavior of the animals? Or does it? One would think that the moon adds to the overall brightness patterns. Can this be calculated? Is the rest much brighter than the moon? This would be exciting.

Thank you, these are all excellent and thought-provoking questions which were also raised by Reviewer 2. Please see our responses to Reviewer 2's major comments 2 and 4.

5. It seems plausible that the unimodal cell types would be useful for defining the heading direction, whereas the bimodal ones could influence steering decisions/course correction, since they respond to CW vs SSW turns, if I understand well. Is such a hypothesis supported by the brain regions where these cells are found (more peripheral vs more central), or not?

This is an incredibly insightful comment, and something we did not notice ourselves. It turns out that there is indeed a correlation between function and brain region in the cells we identified. Of the 8 bimodal cells that we recorded from, we were able to anatomically identify 3, which all localise to the lateral accessory lobes (LAL) and the ventrolateral and ventromedial protocerebrum (VLP, VMP). This distribution exactly matches what the reviewer expressed as a possible function for these cells: The LAL (bimodal cells) is situated directly downstream of the CX and is known to be the steering centre of the insect brain (data from moths and flies), hence a region expected to encode motor steering commands. The other cells (unimodal) are located in the optic lobe and central complex (at least those we could anatomically identify), locations more concerned with early stage sensory processing (optic lobe) and heading encoding (central complex). This insightful point has been added to the manuscript text (lines 219-224). Thank you for this!

6. The heading of moths in Fig 5 (as indicated by the tuning of their cells) appears to be in the SW direction and not directed South, as in most other Figures. Is that an effect of the time of the year when the experiments were done? Please elaborate.

This I think might be a slight misunderstanding as the data in Figure 5 (and Figure 4) show the tuning directions of cells, not the headings of moths – these two are not the same and cannot be compared. The tuning directions of cells in Figure 5 broadly agree with the tuning directions of cells shown in Figure 4b (the red mean arrow). The orange and dark blue dots (cellular tuning directions) that led to the red arrow in Figure 4b were obtained in both seasons, i.e. in spring when moths headed roughly south, and in autumn when they headed roughly north. In other words, the mean tuning angle indicated by the red arrow is constant irrespective of season and thus moth flight heading (a fascinating

finding that we elaborate on in the third paragraph of the section dealing with electrophysiology – “The neural basis of the stellar compass”).

7. The authors make a convincing argument that both magnetic and stellar compasses provide alternative sources of directional information. One can be prioritized when the other becomes unreliable. Can the authors please elaborate under which conditions the magnetic compass were to become unreliable? This is not intuitively obvious. Alternatively, it seems entirely plausible that the two compass systems act together, one reinforcing the other. Have the authors tried to reduce the quality of the stimulus (by blurring, or by reducing intensity) when the 2nd cue was not present? One could expect a deteriorated/less robust response in this case. The way I understand the current magnetoreception literature, it all seems to happen in the photoreceptors....hence, that’s where the two systems could directly interact, no? I feel like this part needs more discussion, since the ‘choice between to systems’ is a bit simplistic and not supported by data.

Again, very interesting questions. And not surprisingly these are also questions raised by another reviewer (Reviewer 3, major comments 1 and 2). Please consult our answers to these comments for the answers to most of your questions above. Concerning a possible location of the magnetic sense in the eyes. This is the hypothesis for birds (with growing evidence that a cryptochrome (Cry)-based mechanism (based on Cry4) is located in the double cones of night-migratory songbirds). However, for insects the evidence so far points to the antennae as the sight of magnetoreception. In Monarch butterflies, for example, which is the only other insect we know of in any detail that migrates to a highly specific destination (like the Bogong moth), recent beautiful work by Christine Merlin’s group in Texas shows that it seems to be the antennae that house the magnetic sense. We have no idea as yet where the magnetic sense might be located in Bogong moths.

8. The authors say that their work is the 1st demonstration of an invertebrate using the stellar pattern for navigation. In the past, they have demonstrated dung beetles using the milky way for orientation. Such orientation tasks usually include many of the same visual cues as navigational tasks do, they even rely on (partly) the same circuits. Can the authors make it more clear how the navigational challenges of bogong moths are fundamentally different from the orientation tasks that a dung beetle faces?

Yes, we can – thanks for the question. Three of four reviewers raised exactly the same point. We have thus provided a detailed answer to this query in a single response at the beginning of this document (as our first response to Reviewer 1) and have modified the manuscript text accordingly. Please refer to this.

Minor comments:

1. Please assign the four cell types in Figure 4a the numbers that they have been given in the text.

Thanks! Done!

2. How has the stellar pattern changed over the last 10,000's of years? I think this question is important when considering how the underlying circuits have evolved/were selected and may be modified in the future.

This is a tremendously important question. The precession of the Earth's axis will produce changes in the apparent positions of stars. Over each 13,000 years, the polar axis moves through half its circle, which means that in 13,000 years from now today's spring stars will be the autumn stars, and vice versa. This means that you can swap the images in Figure 3a. But I think that the moths could slowly adapt to this change, or like birds, rely on the rotational centre of the stars rather than any feature of the stars, to navigate (although our lab experiments with stationary stars reveals that they are sufficient on their own to allow moths to travel in their inherited migratory directions). This is also the case in birds. Once they have learned where the rotational centre is, they can find it without seeing any rotation. The main difference between how moths interpret spring and autumn skies lies in their broad direction of travel, which reverses in autumn compared to spring. In night-migratory birds we know that other factors (rather than the appearance of the stars) are responsible for triggering this reversal – both internal physiological factors (like the release of specific hormones) and/or external environmental factors (like changing daylength or temperatures). But we agree, the precession of the Earth must require some level of robustness to change in the stellar compass system. But so too do the nightly movements of the stars, and this robustness is evident in Figure 2a and 2b.

3. From what I know bogong moths have become extremely endangered & the numbers have gone down rather dramatically. Can the authors please mention that in the text? Can give an outlook as to what kind of population development can be expected? One should seize the opportunity to raise awareness here.

This is indeed sadly true. The recent drought (2017-2020), the worst in Australia's recorded history, reduced the population of this previously very common moth by around 99.5%. Due partly to our efforts, the Bogong moth was listed on the International Red List of endangered species in 2021. Thankfully though, since the drought, we have experienced 3 years of wet La Nina conditions and the moth population has slowly climbed again. It is currently at about $\frac{3}{4}$ of its pre-drought level. We have added a new sentence to the beginning of the first paragraph that highlights the moth's conservation status (see lines 45-46).

Response to Reviewers

Once again all the authors wish to sincerely thank the four reviewers for their extensive efforts to provide excellent feedback on the revised version of our manuscript which has once more considerably improved it. As before, we really appreciate it. Thank you!

We will address each reviewer in turn and use **brown text** to indicate our responses. All line numbers below refer to the line numbers following re-revision (these are the line numbers shown in the (second) re-revised version where **red text** is used to indicate changes in the first revision (as seen already by the reviewers) and **blue text** is used to indicate changes resulting from the latest comments on the first revision. For simplicity, we have not submitted a track-changed version of the re-revised manuscript since the required changes were relatively modest.

Referee #1 (Remarks to the Author):

We feel that the authors have done a thorough job addressing all our prior comments and concerns. Based on this new draft, we have a couple additional comments and thoughts below.

Thank you so much!

1.This point is minor, but in response to our original major comment #1, the authors state they place the results in the context of the overcasts conditions and softened the text by removing “main” in lines 116-118. This is most likely an oversight, but “main” has not been removed.

Yes clearly that was an oversight (sorry about that). The word “main” has now been removed (see line 117).

2.Lines 284-288: This new sentence is very long with several moving parts, which makes it difficult to follow. The authors should consider breaking it down in two.

Agreed, this was clumsy and has been fixed. Please see line 292.

3.Bogong moths were shown to continue to maintain a northward orientation irrespective of whether they were tested during the early or late evening (when the Milky Way and star patterns had significantly shifted). This suggests two possibilities: either moths rely, like birds, on the axis of rotation to determine north/south, or their use of celestial cues might be time-compensated. It might be worth expanding on this idea following the mention that birds use the rotation of the sky for their stellar compass (lines 148-150) – e.g. if

bogong moths use a time-compensated stellar compass, this would suggest the interesting possibility of a conserved role for the circadian clock in time-compensation of stellar and sun compasses used by night-time and day-time migrants, respectively.

Thank you, we completely agree! A small section covering these issues can now be found between lines 149 and 153.

Referee #1 (Remarks on code availability):

I verified the code were made publicly available but do not have the expertise to evaluate them.

Referee #2 (Remarks to the Author):

The manuscript has been improved and a number of aspects clarified. The authors have persuaded me that the moths cannot be using a simple phototaxis. I agree now that it is more akin to some kind of stellar or celestial compass, although I am mystified about how this can work with the Milky Way moving across the sky. If I understand correctly, the bar of light moves and rotates (e.g, Fig. 2), the brightest area of light changes position, there is no obvious stationary part of the sky (like the north star in the northern hemisphere) but moths tested 3 h apart (Fig. 2a, 2b) did not change headings. Can the authors provide any additional thoughts on how this is possible? I know that the hierarchy/weighting of stellar vs magnetic cues cannot be determined in the experiments done outside (Fig. 2a, 2b), but if moths are using stars, is there any possible explanation besides a time-compensated stellar compass? I think it would be worthwhile to comment briefly on this in the manuscript, because time-compensated sun compasses are well known, but a time-compensated star compass would be completely new.

First, many thanks for your many carefully considered comments on our revised manuscript. We really appreciate the efforts you have made to help get our manuscript into better shape. Thank you!

The points above are all very good points, and these were also raised by Reviewer 1 who specifically asked for extra text to briefly speculate on the mechanisms that moths might be using to hold a straight course despite the movements of the stars. Even though we currently have no proof of a time-compensated star compass, we cannot rule it out. Moths may also use the southern centre of stellar rotation to hold a straight course, but again we have no proof. Nonetheless we have now flagged these possibilities in the manuscript – please see lines 149-153 in the re-revised manuscript. Of course, the geomagnetic field, which does not change as the stars move, may also somehow be involved in maintaining a straight course (which our results under cloudy skies do seem to imply: Fig. 2c).

I continue to be enthusiastic about the study, but I remain confused about the difference between the stellar compass of dung beetles and the moths, despite the explanation provided in the rebuttal. To me the two compasses are similar, if not identical, and the attempts to distinguish between them seem premature and unneeded. All we know is that rotating the night sky causes a change in orientation direction in both insects. Functionally, the beetles use stars to travel on a bearing, with each individual selecting a different direction. The moths use stars to travel on a bearing, with all individuals programmed to migrate the same way. It is possible the compasses are different, but right now it seems too soon to know. The ecology of the two insects is different so it makes sense that what the compass is used for is different. It seems analogous to a group of hikers in a forest, each using a compass to move in a different direction (dung beetles), vs. hikers using the same compasses to move toward the same faraway goal (moths).

We understand the issues raised by the reviewer, and we think that in the end it really comes down to semantics. Neurally, the two compasses may indeed be rather similar in terms of their circuitry. Both result in a straight-line trajectory, and both probably require the integration of cues in order to do so. In addition to stellar and geomagnetic signals, Bogong moths require input from three spatiotemporal signals to fly in their inherited migratory heading: their geographic origin, the season and the time of night (as mentioned on lines 252-257). All these signals must be integrated in the central complex to produce the appropriate steering commands. Whether this level and sophistication of signal integration is required for the short random trajectories of nocturnal dung beetles is unknown, but it isn't impossible. The work of Marie Dacke has shown that day-active dung beetles integrate directional information from the sun and the wind to hold a straight trajectory (particularly when the directional information from the sun is weakest around midday). However, it is unknown whether wind information would ever be required under the stars at night, since the stars do not suffer the same degradation in their directional information as the sun does as it nears zenith.

Thus, we are not claiming that the neural circuitry used for the stellar compass necessarily differs between the two species. But what we are claiming is that the compasses of each are used for two very different tasks, one of which is significantly more sophisticated than the other and guides the animal to a distant goal¹ that it has never previously visited (analogous to how the stars are used by night-migratory birds or to seafarers using a sextant).

Something similar is known for sun compasses and magnetic compasses, both are found in animals that use them differently, e.g. by moving short or long distances, or toward a

¹ In terms of body length, this journey is roughly equivalent to a human circumnavigating the Earth twice with only their senses to guide them. During their short lives Bogong moths do this journey twice.

migratory goal vs variable local directions. Also the goal vs non-goal argument does not resonate with me because the dung beetle has a goal (a suitable area away from a crowd of rivals), it is just that suitable areas can be reached by traveling in many directions, whilst the suitable areas for moths all lie in directionally similar areas.

Again we think we are dealing with semantics here. Having worked with dung beetles almost his entire life, it is not the senior author's impression that they have a goal in the sense of the word we are using to describe the distant goal of Bogong moths. They are indeed heading to a suitable area to bury their dung ball, but these areas are literally everywhere around the dung heap and thus no goal (apart from rolling in a straight line) is necessary. Indeed many species of dung beetle have abandoned ball rolling behaviour altogether and simply bury their dung directly beneath the dung pile since this is as good a place as any other in the surroundings.

We suspect that this difference in semantics has arisen due to a difference in how the word "goal" had been interpreted by the reviewer and ourselves. What we mean by "goal" is a specific geographic goal, while the reviewer means the goal of a specific action, i.e. locating a suitable place to perform a specific action (i.e. bury dung). Both are goals, but one is a distant geographic destination reached by travelling in a specific direction while the other is an aim of an action.

One question of terminology. Although navigation means different things to different people, few readers will agree with the definition of navigation provided in the manuscript and on pp. 3 and 8 of the rebuttal, which is that navigation is holding a bearing in 'a specific geographical direction'. That is usually considered orientation, not navigation. Most places in the paper, including in the title, the word navigation should probably be replaced with orientation to avoid confusion, as many readers will assume navigation involves a familiar goal, a learned route/cognitive map like a rodent, true navigation (see rebuttal footnote), etc. The abstract (line 34) can avoid confusion by saying something like "moths can determine direction using the starry night sky and therefore adopt a course that reflects their inherited migratory direction". And so on.

This criticism could be levelled at multiple studies performed on many other species, including the Monarch butterfly, which demonstrably has the same behaviour as the Bogong moth, but makes its journey during the day (i.e. it uses celestial and geomagnetic compass cues to travel in a specific geographic direction to a very distant destination it has never previously visited). Even though it is reasonably well established that the Monarch butterfly is not a true navigator (i.e. it possesses a compass sense but lacks a

map sense), its extraordinary feat of migration to a specific destination in central Mexico is described throughout the literature as an act of navigation².

We completely agree that the uses of the word “navigation” in the literature are not consistent, and we also agree that you could probably argue endlessly about whether Bogong moths navigate or orient (or both). But because the migratory behaviour of Bogong moths is no different to that of Monarch butterflies, and since the behaviour of Monarch butterflies is consistently referred to in the literature as “navigation”, we wish to leave the use of the word “navigation” in this manuscript. Not to do so would imply that the migratory behaviours of Monarch butterflies and Bogong moths are somehow different from each other, which may cause more confusion than less.

Line 104: “Implying that if polarised light is used as a compass cue, it is not essential.” True enough, but I am not quite understanding the point because the same is true for the magnetic compass and star compass. Neither is essential if other cues are present. The sentence after also does not seem relevant, as it probably just means the moths have a magnetic compass, but this seems unrelated to using polarised light. Revise slightly for better clarity?

Thanks for this! Indeed, neither of these cues is essential, but at least one of them must be present for the moths to be able to travel in their inherited migratory direction. The final sentence of that paragraph was intended to lead the reader on to the next paragraph about our findings in overcast conditions, but we agree it may cause confusion. Thus we removed the sentence.

Line 114 maybe say the only known remaining compass cue, as others might be discovered later.

Indeed, thanks. We have now added “known” as suggested (see line 113).

Line 234 section about how long-distance navigation might be controlled, some minor changes might be helpful. As I understand things, the moth does not have any way to correct if it goes off course. I am trying to reconcile what seems to be great variation in

² Just a small sample of papers referring to Monarch butterflies as navigating include: Merlin, Gegear and Reppert (2009) Antennal circadian clocks coordinate sun compass orientation in migratory monarch butterflies. *Science* **325**: 1700-1704; Reppert, Zhu and White (2004) Polarized light helps Monarch butterflies navigate. *Curr. Biol.* **14**: 155-158; Merlin, Heinze and Reppert (2012) Unraveling navigational strategies in migratory insects. *Curr. Opin. Neurobiol.* **22**: 353-361; Reppert, Guerra and Merlin (2016) Neurobiology of Monarch butterfly migration. *Ann. Rev. Entomol.* **61**: 25-42; Reppert and de Roode (2018) Demystifying Monarch butterfly migration. *Curr. Biol.* **28**: R1009-R1022; Beetz and el Jundi (2023). The neurobiology of the Monarch butterfly compass. *Curr. Opin. Insect Sci.* **60**: 101109; Beetz and el Jundi (2023). The influence of stimulus history on directional coding in the Monarch butterfly brain. *J. Comp. Physiol. A* **209**: 663-677.

flight directions (Fig. 2) with what would seem to be the need for a fairly precise compass. In other words, if flight directions represent what moths do, won't many of them miss the target area, given the large dispersion around the mean? Is the area they go to large enough to allow moths to reach the area with such variable headings? Some clarification would be useful.

Yes you are right – in each experiment there is a large spread of moth directions around the mean direction for the population tested. It is difficult to know how much of this spread is due to the nature of the experiment itself (it must be distressing to be forced to fly in an arena with a large metal bar stuck to your back), and how much is natural spread. You are absolutely right that some of the directions the moths are flying in do not seem optimal. The only conclusion one can draw is that either these moths are going to end up in entirely the wrong place (and possibly die, since the destination is quite restricted in area), or that over large distances the average mean direction of a single individual moth is one that does in the end take them to their destination (even though over short distances and times the moth's instantaneous direction from one moment to the next may not always be “correct”). It may be a little like the stock market – sometimes it goes up and sometimes it goes down, but averaged over time the tendency is upwards.

But to answer more specifically, we think that a moth *does* have the ability to correct if it goes off course. That is the role of the central complex – to monitor compass (and other) cues and to generate a steering correction if the current heading does not coincide with the desired heading. We know it seems like the central complex is not doing its job if you look at the spread of moth directions in our 5 minute-long experiments, but again over longer periods of time (hours each night) in untethered conditions, the central complex will be active to correct heading frequently enough to ensure that the average heading over the entire night was the desired one.

Moreover, some spread is probably needed to make the migration stable and robust over evolutionary timescales. Since global patterns of (say) wind directions, magnetic field parameters etc. can change over millennia, a too precise fixation of the goal direction could make the migration vulnerable to changing conditions. Hence, having a spread in headings is likely to be advantageous for giving evolution something to work with and adapt to (if the variation we see is biological).

Line 238 I think most magnetic anomalies are pretty weak, not enough to change a compass heading by much, but a few are strong enough to cause navigation problems for hikers. Are there any strong magnetic anomalies along the path of these moths? If the moths cross such anomalies this would bolster the argument that they sometimes cannot use a magnetic compass.

In the regions of southeast Australia where Bogong moths migrate, the total geomagnetic field strength is between 55,000 nT and 60,000 nT. Significant magnetic anomalies, with increases and decreases in field strength around the average level, exist in many parts of these regions³, particularly in the far west and the eastern third of New South Wales, southeast Queensland and in western Victoria. Swings as high as +1800 nT and as low as -900 nT are not uncommon (i.e. up to around a $\pm 1-3\%$ change in field strength). It's hard to know whether a 1-3% change is enough to cause problems for a magnetic compass, but it is not inconceivable. We have added the reference in footnote 3 below to the main manuscript (new reference 31).

Lines 286-290 do not seem to me to make a convincing distinction between the beetle and moth systems except on the basis of distance travelled. I don't think the term "true compass" is helpful as it raises unanswered questions of when a compass isn't a true compass. In general, I continue to think that the beetle and moth star compasses are more alike than different, but the moth work is a valuable and elegant contribution on its own, even if the compasses are similar.

We have discussed the issue about the difference between dung beetles and Bogong moths above, but we agree that the use of the term "true compass" is probably not helpful given that discussion. We have removed the word "true" and replaced it with "global" (line 295). Certainly we do not mean to imply that Bogong moths undergo "true" navigation and the use of the word "true" here may inadvertently suggest that. Thanks very much for pointing this out!

Methods: For purposes of future replication, it would be helpful to provide some general indicator of the number of responsive neurons vs the number investigated. I know that little can be concluded from negative results but having some idea (about 2% of cells were responsive vs 20%) would help future researchers.

Thanks again for these interesting suggestions. Your request, however, may be difficult to fulfill in terms of the percentage of neurons impaled, as neurons that clearly did not respond to an initial stimulus rotation were immediately discarded and no recording was saved. However, 79 neurons were assessed as potentially responding to the stimulus, and these recordings were therefore saved. Of these 79 recordings, 28 (approx. 35%) met the inclusion criteria of a unimodal or bimodal response profile, while 51 were classified as uniform in their responses to stellar rotations (i.e. showed no obvious response) and were therefore excluded from the analysis. Thus, of the neurons that – at first glance – seemed to respond to the stimulus, 35% had a true response. We can also add that we targeted an area of the central brain in which we expected to find both central complex and lateral complex neurons, as well as optic lobe neurons traversing the brain

³ <https://www.ga.gov.au/bigobj/GA18004.pdf>

in the posterior optic tract. In other brain areas, the percentage of responsive neurons would potentially be different.

We have included a new paragraph in the Methods to summarise these statistics (lines 813-819).

Similarly, can some indication be given about the approximate percent of moths that met the testing criteria, recognising that this probably changes with weather, date, etc.?

We have carefully checked our lab journals for the field seasons of Spring 2018 and 2019, as well for Autumn 2019. The percentages of moths that were aborted due to failing to meet our testing criteria were 2 of 59 or 3.4% (Spring 2018), 25 of 137 or 18.3% (Spring 2019), and 35 of 76 or 46.1% (Autumn 2019). Thus, precisely as you suspected, the percentage of moths that met the testing criteria were quite variable. We checked our journals regarding the weather in Autumn 2019 (when the numbers of moths that met the criteria were least) and we found that this season was unusually wet and cold. For instance, our lab journals reveal that we conducted experiments on 4 nights that followed particularly rainy days (and when the number of moths meeting our criteria was low). We have consistently found that moths behave erratically on rainy or stormy nights, and even on days preceding and following such nights. They almost seem to “know” that this is not good weather to migrate in (even though they are indoors), possibly because they can sense changes in barometric pressure. The rainy weather might thus explain the high rate of aborted experiments during this particular Spring. We have since avoided running experiments in such weather.

We have included new text in the Methods to summarise these statistics (lines 743-748).

Referee #3 (Remarks to the Author):

I believe that the authors have addressed the concerns I raised very well. I am pleased with the final manuscript at this point. I will leave it to the other reviewers to discuss in detail the responses to the points they raised, but in general, I also think the authors did a good and thorough job addressing all concerns. I will also say that the degree of points raised by all the reviewers both for this study and future work speaks well to the notion that the paper should indeed be published in Nature.

Roy E. Ritzmann

Thank you, Roy! We are really grateful for your expert input, and we are very glad that you are happy with the manuscript following revision. Many thanks again for your big efforts to improve our manuscript! We really appreciate your wisdom and assistance.

Referee #4 (Remarks to the Author):

Thank you very much to the authors for this very detailed discussion of all comments from all four reviewers. It took me a while to read through all of it & think it was a very enlightening discussion. I am happy that the manuscript has improved even more now. All my points have been addressed, so I recommend publication. Happy New Year!

Referee #4 (Remarks on code availability):

N/A

Thank you so much! Again, we really are very grateful for your input which has dramatically improved our manuscript. Much appreciated! A very happy new year to you as well!